# Importance Corrected Neural JKO Sampling

## Abstract

In order to sample from an unnormalized probability density function, we propose to combine continuous normalizing flows (CNFs) with rejection-resampling steps based on importance weights. We relate the iterative training of CNFs with regularized velocity fields to a JKO scheme and prove convergence of the involved velocity fields to the velocity field of the Wasserstein gradient flow (WGF). The alternation of local flow steps and non-local rejection-resampling steps allows to overcome local minima or slow convergence of the WGF for multimodal distributions. Since the proposal of the rejection step is generated by the model itself, they do not suffer from common drawbacks of classical rejection schemes. The arising model can be trained iteratively, reduces the reverse Kullback-Leibler (KL) loss function in each step, allows to generate *iid* samples and moreover allows for evaluations of the generated underlying density. Numerical examples show that our method yields accurate results on various test distributions including high-dimensional multimodal targets and outperforms the state of the art in almost all cases significantly.

## 1 Introduction

We consider the problem of sampling from an unnormalized probability density function. That is, we are given an integrable function $g \colon \mathbb{R}^d \to \mathbb{R}_{>0}$ and we aim to generate samples from the probability distribution $\nu$ given by the density $q(x) = g(x)/Z_g$, where the normalizing constant $Z_g = \int_{\mathbb{R}^d} g(x)\mathrm{d}x$ is unknown. Many classical sampling methods are based on Markov chain Monte Carlo (MCMC) methods like the overdamped Langevin sampling, see, e.g., Welling & Teh (2011). The generated probability path of the underlying stochastic differential equation follows the Wasserstein-2 gradient flow of the reverse KL divergence $\mathcal{F}(\mu) = \mathrm{KL}(\mu, \nu)$. Over the last years, generative models like normalizing flows (Rezende & Mohamed, 2015) or diffusion models (Ho et al., 2020; Song et al., 2021) became more popular for sampling, see, e.g., Phillips et al. (2024); Vargas et al. (2023a). Also these methods are based on the reverse KL divergence as a loss function. While generative models have successfully been applied in data-driven setups, their application to the problem of sampling from arbitrary unnormalized densities is not straightforward. This difficulty arises from the significantly harder nature of the problem, even in moderate dimensions, particularly when dealing with target distributions that exhibit phenomena such as concentration effects, multimodalities, heavy tails, or other issues related to the curse of dimensionality.

In particular, the reverse KL is non-convex in the Wasserstein space as soon as the target density $\nu$ is not log-concave which is for example the case when $\nu$ consists of multiple modes. In this case generative models often collapse to one or a small number of modes. We observe that for continuous normalizing flows (CNFs, Chen et al., 2018; Grathwohl et al., 2019) this can be prevented by regularizing the $L^2$-norm of the velocity field as proposed under the name OT-flow by Onken et al. (2021). In particular, this regularization converts the objective functional into a convex one. However, the minimizer of the regularized loss function is no longer given by the target measure $\nu$ but by the Wasserstein proximal mapping of the objective function applied onto the latent distribution. Considering that the Jordan-Kinderlehrer-Otto (JKO) scheme (Jordan et al., 1998) iteratively applies the Wasserstein proximal mapping and converges to the Wasserstein gradient flow, several papers proposed to approximate the steps of the scheme by generative models, see Altekrüger et al. (2023); Alvarez-Melis et al. (2022); Fan et al. (2022); Lambert et al. (2022); Mokrov et al. (2021); Vidal et al. (2023); Xu et al. (2024). We will refer to this class of method by the name *neural JKO*. Even though this approximates the same gradient flow of the Langevin dynamics these approaches have usually the advantage of faster inference (once they are trained) and additional possibly allow for density evaluations.

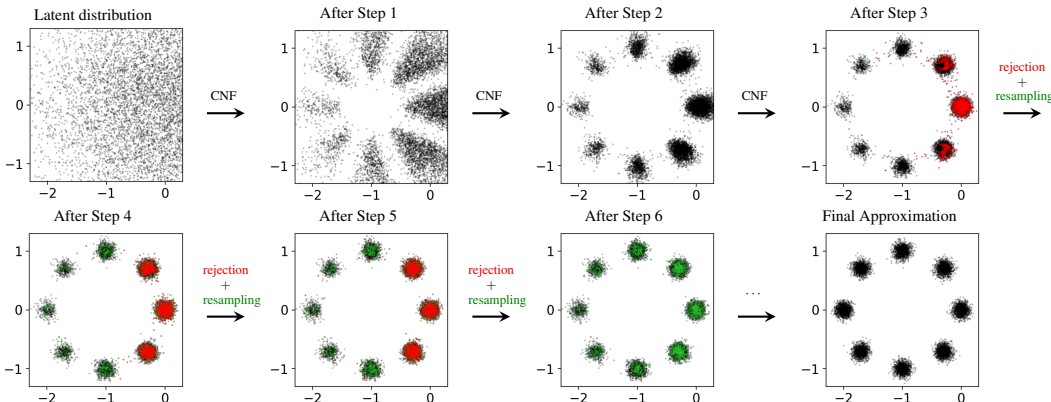

Figure 1: Iterative application of neural JKO steps and rejection steps for a shifted mixture target distribution. The red samples are rejected in the next following rejection step and the green samples are the resampled points. The latter approach enables for the correction of wrong mode weights introduced by the underlying WGF. See Figure 2 for more steps.

However, already the time-continuous Wasserstein gradient flow suffers from the non-convexity of the reverse KL loss function by getting stuck in local minima or by very slow convergence times. In particular, it is well-known that Langevin-based sampling methods often do not distribute the mass correctly onto multimodal target distributions. As a remedy, Neal (2001) proposed importance sampling, i.e., to reweight the sample based on quotients of the target distribution and its current approximation which leads to an unbiased estimator of the Monte Carlo integral. However, this estimator may lead to highly imbalanced weights between these samples and the resulting estimator might have a large variance. Moreover, the density of the current approximation has to be known up to a possible multiplicative constant, which is not the case for many MCMC methods like Langevin sampling or Hamiltonian Monte Carlo. In Del Moral et al. (2006) the authors propose a scheme for alternating importance sampling steps with local Monte Carlo steps. However, as a downside this strategy leads to samples which might not be *iid*.

**Contributions**   In this paper, we propose a sampling method which combines neural JKO steps based on CNFs with importance based rejection steps. While the CNFs adjust the position of the generated samples *locally*, the rejection steps readjusts the inferred distribution *non-locally* based on the quotient of generated and target distribution. Then, in each rejection step we resample the rejected points based on the current constructed generative model. We illustrate this procedure in Figure 1.

Our methods generates independent samples and allows to evaluate the density of the generated distribution. In our numerical examples, we apply our method to common test distributions up to the dimension $d = 1600$ which are partially highly multimodal. We show that our *importance corrected neural JKO* sampling (neural JKO IC) achieves significantly better results than the comparisons[1].

From a theoretical side, we prove that the velocity fields from a sequence of neural JKO steps strongly converges to the velocity field of the corresponding Wasserstein gradient flow and that the reverse KL loss function decreases throughout the importance-based rejection steps.

**Outline**   The paper is organized as follows. In Section 2, we recall the fundamental concepts which will be required. Afterwards, we consider neural JKO schemes more detailed in Section 3. We introduce our importance-based rejection steps in Section 4. Finally, we evaluate our model numerically and compare it to existing methods in Section 5. Conclusions are drawn in Section 6. Additionally, proofs, further numerical and technical details are presented in Appendix A- E.

RELATED WORK

**MCMC Algorithms**   Common methods for sampling of unnormalized densities are often based on Markov Chain Monte Carlo (MCMC) methods, see e.g. (Gilks et al., 1995). In particular first

---

[1]The code is available in the supplementary material.

order based variants, such as the Hamiltonian Monte Carlo (HMC) (Betancourt, 2017; Hoffman & Gelman, 2014) and the Metropolis Adjusted Langevin Algorithm (MALA Girolami & Calderhead, 2011; Rossky et al., 1978; Roberts & Tweedie, 1996) are heavily used in practice. The viewpoint of these samplers as sample space description of gradient flows defined in a metricized probability space then allows for extensions such as interacting particle systems (Chen et al., 2023; Eigel et al., 2024; Garbuno-Inigo et al., 2020; Wang & Li, 2022). However, since these algorithms are based on local transformations of the samples, they are unable to distribute the mass correctly among different modes, which can partially be corrected by importance sampling (Neal, 2001) and sequential Monte Carlo samplers (SMC) (Del Moral et al., 2006) as described above. In contrast to our model, SMC approximates the density of the approximation by assigning "inverse Markov kernels" to certain MCMC kernels, which might lead to propagating errors. Furthermore, the generation of additional samples requires to rerun the whole procedure which can be very costly.

**Generative Models**  In the last years, generative models became very popular, including VAEs (Kingma & Welling, 2014), normalizing flows (Rezende & Mohamed, 2015), diffusion models (Ho et al., 2020) or flow-matching (Lipman et al., 2022) which is also known as rectified flow (Liu et al., 2022). In contrast to our setting, they initially consider the *modeling* task, i.e., they assume that they are given samples from the target measure instead of an unnormalized density. However, there are several papers, which adapt these algorithms for the *sampling* task. For normalizing flows, this mostly amounts to changing the loss function (Hoffman et al., 2019; Marzouk et al., 2016; Qiu & Wang, 2024). Very recently, there appeared also a flow-matching variant for the sampling task (Woo & Ahn, 2024). For diffusion (and stochastic control) models this was done based on variational approaches (Blessing et al., 2024; Phillips et al., 2024; Vargas et al., 2023a;b; Zhang & Chen, 2021) or by computing the score by solving a PDE (Albergo & Vanden-Eijnden, 2024; Richter & Berner, 2024; Sommer et al., 2024). These methods usually provide much faster sampling times than MCMC methods and are often used in combination with some conditioning parameter for inverse problems, where a (generative) prior is combined with a known likelihood term (Ardizzone et al., 2019; Altekrüger & Hertrich, 2023; Andrle et al., 2021; Denker et al., 2024). Combinations of generative models with stochastic sampling steps were considered in the literature for generative modeling under the name stochastic normalizing flows (Hagemann et al., 2023; 2022; Noé et al., 2019; Wu et al., 2020) and for sampling under the name annealed flow transport Monte Carlo (Arbel et al., 2021; Matthews et al., 2022). Gabrié et al. (2022) use normalizing flows to learn proposal distributions in an Metropolis-Hastings algorithm. Additionally, flow-based generative sampling algorithms can be combined with one final importance sampling step, which is done in many of these references.

**Generative Models with Wasserstein Gradient Flows**  These generative models can be adapted to follow a Wasserstein gradient flow, by mimicking a JKO scheme with generative models or directly follow the velocity field of a kernel-based functional. Such approaches were proposed for generative modeling (Fan et al., 2022; Hagemann et al., 2024; Hertrich et al., 2024; Liutkus et al., 2019; Vidal et al., 2023; Xu et al., 2024), sampling (Fan et al., 2022; Lambert et al., 2022; Liu & Wang, 2016; Mokrov et al., 2021) or other tasks (Altekrüger et al., 2023; Arbel et al., 2019; Alvarez-Melis et al., 2022).

## 2 PRELIMINARIES

In this section, we provide a rough overview of the required concepts for this paper. To this end, we first revisit the basic definitions of Wasserstein gradient flows, e.g., based on Ambrosio et al. (2005). Afterwards we recall continuous normalizing flows with OT-regularizations.

### 2.1 WASSERSTEIN SPACES AND ABSOLUTELY CONTINUOUS CURVES

**Wasserstein Distance**  Let $\mathcal{P}(\mathbb{R}^d)$ be the space of probability measures on $\mathbb{R}^d$ and denote by $\mathcal{P}_2(\mathbb{R}^d) := \{\mu \in \mathcal{P}(\mathbb{R}^d) : \int_{\mathbb{R}^d} \|x\|^2 \mathrm{d}x < \infty\}$ the subspace of probability measures with finite second moment. Let $\mathcal{P}_2^{\mathrm{ac}}(\mathbb{R}^d)$ be the subspace of absolutely continuous measures from $\mathcal{P}_2(\mathbb{R}^d)$. Moreover, we denote for $\mu, \nu \in \mathcal{P}_2(\mathbb{R}^d)$ by $\Gamma(\mu, \nu) := \{\boldsymbol{\pi} \in \mathcal{P}_2(\mathbb{R}^d \times \mathbb{R}^d) : \pi_{1\#}\boldsymbol{\pi} = \mu, \pi_{2\#}\boldsymbol{\pi} = \nu\}$ the set of all transport plans with marginals $\mu$ and $\nu$, where $\pi_i \colon \mathbb{R}^d \times \mathbb{R}^d \to \mathbb{R}^d$ defined by $\pi_i(x_1, x_2) = x_i$ is the projection onto the $i$-th component for $i = 1, 2$. Then, we equip $\mathcal{P}_2(\mathbb{R}^d)$ with the Wasserstein-2

metric defined by

$$W_2^2(\mu, \nu) = \inf_{\boldsymbol{\pi} \in \Gamma(\mu,\nu)} \int_{\mathbb{R}^d \times \mathbb{R}^d} \|x - y\|^2 \mathrm{d}\boldsymbol{\pi}(x,y).$$

The minimum is attained, but not always unique. We denote the subset of minimizers by $\Gamma^{\mathrm{opt}}(\mu, \nu)$. If $\mu \in \mathcal{P}_2^{\mathrm{ac}}(\mathbb{R}^d)$, then the optimal transport plan is unique and is given by a so-called transport map.

**Theorem 1** (Brenier (1987)). *Let $\mu \in \mathcal{P}_2^{\mathrm{ac}}(\mathbb{R}^d)$ and $\nu \in \mathcal{P}_2(\mathbb{R}^d)$. Then there is a unique transport plan $\boldsymbol{\pi} \in \Gamma^{\mathrm{opt}}(\mu, \nu)$ which is induced by a unique measurable optimal transport map $T \colon \mathbb{R}^d \to \mathbb{R}^d$, i.e., $\boldsymbol{\pi} = (\mathrm{Id}, T)_{\#}\mu$ and*

$$W_2^2(\mu, \nu) = \min_{T \colon \mathbb{R}^d \to \mathbb{R}^d} \int_{\mathbb{R}^d} \|T(x) - x\|_2^2 \, \mathrm{d}\mu(x) \quad subject \quad to \quad T_{\#}\mu = \nu.$$

**Absolutely Continuous Curves** A curve $\gamma \colon I \to \mathcal{P}_2(\mathbb{R}^d)$ on the interval $I \subseteq \mathbb{R}$ is called *absolutely continuous* if there exists a Borel velocity field $v \colon \mathbb{R}^d \times I \to \mathbb{R}^d$ with $\int_I \|v(\cdot, t)\|_{L_2(\gamma(t), \mathbb{R}^d)} \mathrm{d}t < \infty$ such that the continuity equation

$$\partial_t \gamma(t) + \nabla \cdot (v(\cdot, t)\gamma(t)) = 0 \tag{1}$$

is fulfilled on $I \times \mathbb{R}^d$ in a weak sense. Then, any velocity field $v$ solving the continuity equation (1) for fixed $\gamma$ characterizes $\gamma$ as $\gamma(t) = z(\cdot, t)_{\#}\gamma(t_0)$, where $z$ is the solution of the ODE $\dot{z}(x, t) = v(z(x, t), t)$ with $z(x, t_0) = x$ and $t_0 \in I$. It can be shown that for an absolutely continuous curve there exists a unique solution of minimal norm which is equivalently characterized by the so-called regular tangent space $\mathrm{T}_{\gamma(t)}\mathcal{P}_2(\mathbb{R}^d)$, see Appendix A for details. An absolute continuous curve is a geodesic if there exists some $c > 0$ such that $W_2(\gamma(s), \gamma(t)) = c|s - t|$.

The following theorem formulates a dynamic version of the Wasserstein distance based minimal energy curves in the Wasserstein space.

**Theorem 2** (Benamou & Brenier (2000)). *Assume that $\mu, \nu \in \mathcal{P}_2^{\mathrm{ac}}(\mathbb{R}^d)$. Then, it holds*

$$W_2^2(\mu, \nu) = \inf_{\substack{v \colon \mathbb{R}^d \times [0,1] \to \mathbb{R}^d, \\ \dot{z}(x,t) = v(z(x,t),t), \\ z(x,0) = x, \, z(\cdot,1)_{\#}\mu = \nu}} \int_0^1 \int_{\mathbb{R}^d} \|v(z(x,t), t)\|^2 \mathrm{d}\mu(x)\mathrm{d}t.$$

*Moreover, there exists a unique minimizing velocity field $v$ and the curve defined by $\gamma(t) = z(\cdot, t)_{\#}\mu$ with $t \in [0, 1]$ and $\dot{z}(x, t) = v(z(x, t), t)$, $z(x, 0) = x$ is a geodesic which fulfills the continuity equation $\partial_t \gamma(t) + \nabla \cdot (v(\cdot, t)\gamma(t)) = 0$.*

Let $\tau > 0$. Then, by substitution of $t$ by $t/\tau$ and rescaling $v$ in the time variable, this is equal to

$$W_2^2(\mu, \nu) = \inf_{\substack{v \colon \mathbb{R}^d \times [0,\tau] \to \mathbb{R}^d, \\ \dot{z}(x,t) = v(z(x,t),t), \\ z(x,0) = x, \, z(\cdot,\tau)_{\#}\mu = \nu}} \tau \int_0^\tau \int_{\mathbb{R}^d} \|v(z(x,t), t)\|^2 \mathrm{d}\mu(x)\mathrm{d}t.$$

**Wasserstein Gradient Flows** An absolutely continuous curve $\gamma \colon (0, \infty) \to \mathcal{P}_2(\mathbb{R}^d)$ with velocity field $v_t \in \mathrm{T}_{\gamma(t)}\mathcal{P}_2(\mathbb{R}^d)$ is a *Wasserstein gradient flow with respect to* $\mathcal{F} \colon \mathcal{P}_2(\mathbb{R}^d) \to (-\infty, \infty]$ if

$$v_t \in -\partial \mathcal{F}(\gamma(t)), \quad \text{for a.e. } t > 0,$$

where $\partial \mathcal{F}(\mu)$ denotes the reduced Fréchet subdiffential at $\mu$, see Appendix A for a definition.

To compute Wasserstein gradient flows numerically, we can use the generalized minimizing movements or Jordan-Kinderlehrer-Otto (JKO) scheme (Jordan et al., 1998). To this end, we consider the Wasserstein proximal mapping defined as

$$\mathrm{prox}_{\tau\mathcal{F}}(\hat{\mu}) = \operatorname*{arg\,min}_{\mu \in \mathcal{P}_2(\mathbb{R}^d)} \{\frac{1}{2} W_2^2(\mu, \hat{\mu}) + \tau \mathcal{F}(\mu)\}.$$

Then, define as $\mu_\tau^k$ for $k \in \mathbb{N}$ the steps of the minimizing movements scheme, i.e.,

$$\mu_\tau^0 = \mu^0, \qquad \mu_\tau^{k+1} = \mathrm{prox}_{\tau\mathcal{F}}(\mu_\tau^k). \tag{2}$$

We denote the piecewise constant interpolations $\tilde{\gamma}_\tau \colon [0, \infty) \to \mathcal{P}_2(\mathbb{R}^d)$ of the minimizing movement scheme by

$$\tilde{\gamma}_\tau(k\tau + t\tau) = \mu_\tau^k, \quad t \in [0, 1). \tag{3}$$

Then, the following convergence result holds true.

**Theorem 3.** *(Ambrosio et al., 2005, Thm 11.2.1) Let $\mathcal{F}\colon \mathcal{P}_2(\mathbb{R}^d) \to (-\infty, +\infty]$ be proper, lsc, coercive, and $\lambda$-convex along generalized geodesics, and let $\mu^0 \in \overline{\mathrm{dom}\,\mathcal{F}}$. Then the curves $\tilde{\gamma}_\tau$ defined via the minimizing movement scheme (3) converge for $\tau \to 0$ locally uniformly to a locally Lipschitz curve $\gamma\colon (0, +\infty) \to \mathcal{P}_2(\mathbb{R}^d)$ which is the unique Wasserstein gradient flow of $\mathcal{F}$ with $\gamma(0+) = \mu^0$.*

## 2.2 CONTINUOUS NORMALIZING FLOWS AND OT-FLOWS

The concept of normalizing flows first appeared in Rezende & Mohamed (2015). It follows the basic idea to approximate a probability distribution $\nu$ by considering a simple latent distribution $\mu_0$ (usually a standard Gaussian) and to construct a diffeomorphism $\mathcal{T}_\theta\colon \mathbb{R}^d \to \mathbb{R}^d$ depending on some parameters $\theta$ such that $\nu \approx \mathcal{T}_{\theta\#}\mu_0$. In practice, the diffeomorphism can be approximated by coupling-based neural networks (Dinh et al., 2016; Kingma & Dhariwal, 2018), residual architectures (Behrmann et al., 2019; Chen et al., 2019; Hertrich, 2023) or autoregressive flows (De Cao et al., 2020; Durkan et al., 2019; Huang et al., 2018; Papamakarios et al., 2017). In this paper, we mainly focus on *continuous normalizing flows* proposed by Chen et al. (2018); Grathwohl et al. (2019), see also Ruthotto & Haber (2021) for an overview. Here the diffeomorphism $\mathcal{T}_\theta$ is parameterized as neural ODE. To this end let $v_\theta\colon \mathbb{R}^d \times \mathbb{R} \to \mathbb{R}^d$ be a neural network with parameters $\theta$ and let $z_\theta\colon \mathbb{R}^d \times [0, \tau] \to \mathbb{R}^d$ for fixed $\tau > 0$ be the solution of $\dot{z}_\theta = v_\theta$, with initial condition $z_\theta(x, 0) = x$. Then, we define $\mathcal{T}_\theta$ via the solution $z_\theta$ as $\mathcal{T}_\theta(x) = z_\theta(x, \tau)$.

The density $p_\theta$ of $\mathcal{T}_{\theta\#}\mu_0$ can be described by the change-of-variables formula

$$p_\theta(x) = \frac{p_0(x)}{|\det(\nabla \mathcal{T}_\theta(x))|},$$

where $p_0$ is the density of the latent distribution $\mu_0$. In the case of continuous normalizing flows, it can be shown that the denominator can be computed as

$$\log(|\det(\nabla \mathcal{T}_\theta(x))|) = \ell_\theta(x, \tau), \quad \partial_t \ell_\theta(x, t) = \mathrm{trace}(\nabla v_\theta(z_\theta(x, t), t)), \quad \ell_\theta(\cdot, 0) = 0.$$

In order to train a normalizing flow, one usually uses the Kullback-Leibler divergence. If $\nu$ is given by a density $q(x) = Z_g g(x)$, where $Z_g$ is an unknown normalizing constant, then this amounts to the reverse KL loss function

$$\begin{aligned}
\mathcal{L}(\theta) = \mathrm{KL}(\mathcal{T}_{\theta\#}\mu_0, \nu) &= \mathbb{E}_{x \sim \mu_0}[-\log(q(\mathcal{T}_\theta(x))) + \log(p_\theta(x))] \\
&\propto \mathbb{E}_{x \sim \mu_0}[-\log(q(\mathcal{T}_\theta(x))) - \ell_\theta(x, \tau)],
\end{aligned}$$

where $\propto$ indicates equality up to a constant.

In order to stabilize and accelerate the training, Onken et al. (2021) propose to regularize the velocity field $v_\theta$ by its expected squared norm. More precisely, they propose to add the regularizer

$$\mathcal{R}(\theta) = \tau \int_0^\tau \|v_\theta(z_\theta(x, t), t)\|^2 \mathrm{d}t$$

to the loss function. This leads to straight trajectories in the ODE such that adaptive solvers only require very few steps to solve them. Following Theorem 2, the authors of Onken et al. (2021) note that for $\beta > 0$ the functional $\mathcal{L}(\theta) + \alpha \mathcal{R}(\theta)$ has the same minimizer as the functional $\mathcal{L}(\theta) + \beta W_2^2(\mu_0, \mathcal{T}_{\theta\#}\mu_0)$, which relates to the JKO scheme as pointed out by Vidal et al. (2023).

## 3 NEURAL JKO SCHEME

In the following, we learn the steps (2) of the JKO scheme by neural ODEs. While similar schemes were already suggested in several papers (Altekrüger et al., 2023; Alvarez-Melis et al., 2022; Fan et al., 2022; Lambert et al., 2022; Mokrov et al., 2021; Vidal et al., 2023; Xu et al., 2024), we are particularly interested in the convergence properties of the corresponding velocity fields. In Subsection 3.1, we introduce the general scheme and derive its properties. Afterwards, in Subsection 3.2, we describe the corresponding neural network approximation.

Throughout this section, we consider the following assumptions on the objective functional $\mathcal{F}$ and our initialization $\mu_0$.

**Assumption 4.** *Let $\mathcal{F}\colon \mathcal{P}_2(\mathbb{R}^d) \to \mathbb{R}\cup\{\infty\}$ be proper, lower semi-continuous with respect to narrow convergence, coercive, $\lambda$-convex along generalized geodesics and bounded from below. Moreover, assume that $\mathrm{dom}(|\partial\mathcal{F}|) \subseteq \mathcal{P}_2^{\mathrm{ac}}(\mathbb{R}^d)$ and that $\mathcal{F}$ has finite metric derivative $|\partial\mathcal{F}|(\mu_0) < \infty$ at the initialization $\mu_0 \in \mathcal{P}_2^{\mathrm{ac}}(\mathbb{R}^d)$.*

This assumption is fulfilled for many important divergences and loss functions $\mathcal{F}$. We list some examples in Appendix B.1. We will later pay particular attention to the reverse Kullback-Leibler divergence $\mathcal{F}(\mu) = \mathrm{KL}(\mu, \nu) = \int p(x) \log\left(\frac{p(x)}{q(x)}\right)\mathrm{d}x$, where $\nu$ is a fixed target measure and $p$ and $q$ are the densities of $\mu$ and $\nu$ respectively. This functional fulfills Assumption 4 if $q$ is $\lambda$-convex.

Additionally, (Ambrosio et al., 2005, Lem 9.2.7) states that the functional $\mathcal{G}(\mu) = \frac{1}{2\tau}W_2^2(\mu, \mu_\tau^k) + \mathcal{F}(\mu)$ is $(\lambda + \frac{1}{\tau})$-convex along geodesics. In particular, for $\tau < \frac{1}{\lambda}$, the functional $\mathcal{G}$ is strongly convex such that we expect that optimizing it with a generative model is much easier than optimizing $\mathcal{F}$, see also Appendix F.1 for a discussion how this can prevent mode collapse.

### 3.1 PIECEWISE GEODESIC INTERPOLATION

In order to represent the JKO scheme by neural ODEs, we first reformulate it based on Benamou-Brenier (Theorem 2). To this end, we insert the dynamic formulation of the Wasserstein distance in the Wasserstein proximal mapping defining the steps in (2). For any $\mu \in \mathcal{P}(\mathbb{R}^d)$ this leads to

$$\frac{1}{2\tau}W_2^2(\mu, \mu_\tau^k) + \mathcal{F}(\mu) = \min_{\substack{v\colon \mathbb{R}^d\times[0,\tau]\to\mathbb{R}^d \\ \dot{z}(x,t)=v(z(x,t),t), \\ z(x,0)=x,\, z(\cdot,\tau)_\#\mu_\tau^k=\mu}} \frac{1}{2}\int_0^\tau \int_{\mathbb{R}^d} \|v(z(x,t),t)\|^2 \mathrm{d}\mu_\tau^k(x)\mathrm{d}t + \mathcal{F}(\mu).$$

Hence from a minimizer perspective, we obtain that $\mu_\tau^{k+1} = z_\tau^k(\cdot, T)_\#\mu_\tau^k$, where

$$(v_{\tau,k}, z_{\tau,k}) \in \operatorname*{arg\,min}_{\substack{v\colon \mathbb{R}^d\times[0,\tau]\to\mathbb{R}^d \\ \dot{z}(x,t)=v(z(x,t),t),\, z(x,0)=x}} \left\{\frac{1}{2}\int_0^\tau \int_{\mathbb{R}^d} \|v(z(x,t),t)\|^2 \mathrm{d}\mu_\tau^k(x)\mathrm{d}t + \mathcal{F}(z(\cdot,\tau)_\#\mu_\tau^k)\right\}.$$

$$(4)$$

Finally, we concatenate the velocity fields of all steps and obtain the ODE

$$v_\tau|_{(k\tau,(k+1)\tau]} = v_{\tau,k}, \qquad \dot{z}_\tau(x,t) = v_\tau(z_\tau(x,t),t), \qquad z_\tau(x,0) = x.$$

As a straightforward observation, we obtain that the curve defined by the velocity field $v_\tau$ is the geodesic interpolation between the points from JKO scheme (2). We state a proof in Appendix B.2.

**Corollary 5.** *Under Assumption 4 the following holds true.*

(i) *It holds that $W_2^2(\mu_\tau^k, \mu_\tau^{k+1}) = \tau\int_0^\tau \int_{\mathbb{R}^d} \|v_{\tau,k}(z_{\tau,k}(x,t),t)\|^2 \mathrm{d}\mu_\tau^k(x)\mathrm{d}t$, i.e., $v_{\tau,k}$ is the optimal velocity field from the theorem of Benamou-Brenier.*

(ii) *The curve $\gamma_\tau(t) := z(\cdot,t)_\#\mu_0$ fulfills $\gamma_\tau(k\tau + t\tau) = ((1-t)I + t\mathcal{T}_\tau^k)_\#\mu_\tau^k$ for $t \in [0,1]$, where $\mathcal{T}_\tau^k$ is the optimal transport map between $\mu_\tau^k$ and $\mu_\tau^{k+1}$.*

(iii) *$v_\tau$ and $\gamma_\tau$ solve the continuity equation $\partial_t\gamma_\tau(t) + \nabla \cdot (v_\tau(\cdot,t)\gamma(t)) = 0$.*

Analogously to Theorem 3 one can show that also the curves $\gamma_\tau$ are converging locally uniformly to the unique Wasserstein gradient flow (see, e.g., the proof of Ambrosio et al., 2005, Thm 11.1.6). In the next subsection, we will approximate the velocity fields $v_{\tau,k}$ by neural networks. In order to retain the stability of the resulting scheme, the next theorem states that also the velocity fields $v_\tau$ converge strongly towards the velocity field from the Wasserstein gradient flow. The proof is given in Appendix B.3.

**Theorem 6.** *Suppose that Assumption 4 is fulfilled and let $(\tau_l)_l \subseteq (0, \infty)$ with $\tau_l \to 0$. Then, $(v_{\tau_l})_l$ converges strongly to the velocity field $\hat{v} \in L_2(\gamma, \mathbb{R}^d \times [0, T])$ of the Wasserstein gradient flow $\gamma\colon (0, \infty) \to \mathcal{P}_2(\mathbb{R}^d)$ of $\mathcal{F}$ starting in $\mu^0$.*

In some cases the limit velocity field $v$ can be stated explicitly, even though its direct computation is intractable. For details, we refer to Appendix B.1.

## 3.2 NEURAL JKO SAMPLING

In the following, we learn the velocity fields $v_{\tau,k}$ as neural ODEs in order to sample from a target measure $\nu$ given by the density $q(x) = \frac{1}{Z_g} g(x)$ with unknown normalization constant $Z_g = \int_{\mathbb{R}^d} g(x)\mathrm{d}x$. To this end, we consider the Wasserstein gradient flow with respect to the reverse KL loss function $\mathcal{F}(\mu) = \mathrm{KL}(\mu, \nu)$ which has the unique minimizer $\mu = \nu$.

Then, due to (4) the loss function from the JKO steps for the training of the velocity field $v_\theta$ reads as

$$\mathcal{L}(\theta) = \frac{1}{2} \int_0^\tau \int_{\mathbb{R}^d} \|v_\theta(z_\theta(x,t),t)\|^2 \mathrm{d}\mu_\tau^k(x)\mathrm{d}t + \mathcal{F}(z_\theta(\cdot,\tau)_{\#}\mu_\tau^k),$$

where $z_\theta$ is the solution of $\dot{z}_\theta(x,t) = v_\theta(z_\theta(x,t),t)$ with $z_\theta(x,0) = x$. Now, following the derivations of continuous normalizing flows, cf. Section 2.2, the second term of $\mathcal{L}$ can be rewritten (up to an additive constant) as

$$\mathcal{F}(z_\theta(\cdot,\tau)_{\#}\mu_\tau^k) \propto \mathbb{E}_{x \sim \mu_\tau^k}[-\log(g(z_\theta(x,\tau))) - \ell_\theta(x,\tau)],$$

where $\ell_\theta$ is the solution of $\dot{\ell}_\theta(x,t) = \mathrm{trace}(\nabla v_\theta(z_\theta(x,t),t))$ with $\ell_\theta(\cdot,0) = 0$. Moreover, we can rewrite the first term of $\mathcal{L}$ based on

$$\int_0^\tau \int_{\mathbb{R}^d} \|v_\theta(z_\theta(x,t),t)\|^2 \mathrm{d}\mu_\tau^k(x)\mathrm{d}t = \mathbb{E}_{x \sim \mu_\tau^k}\Big[\int_0^\tau \|v_\theta(z_\theta(x,t),t)\|^2 \mathrm{d}t\Big] = \mathbb{E}_{x \sim \mu_\tau^k}[\omega_\theta(x,\tau)],$$

where $\omega_\theta$ is the solution of $\dot{\omega}_\theta(x,t) = \|v_\theta(z_\theta(x,t),t)\|^2$. Hence, we can represent $\mathcal{L}$ up to an additive constant as

$$\mathcal{L}(\theta) = \mathbb{E}_{x \sim \mu_\tau^k}[-\log(g(z_\theta(x,\tau))) - \ell_\theta(x,\tau) + \omega_\theta(x,\tau)], \tag{5}$$

where $(z_\theta, \ell_\theta, \omega_\theta)$ solves the ODE system

$$\begin{pmatrix} \dot{z}_\theta(x,t) \\ \dot{\ell}_\theta(x,t) \\ \dot{\omega}_\theta(x,t) \end{pmatrix} = \begin{pmatrix} v_\theta(x,t) \\ \mathrm{trace}(\nabla v_\theta(z_\theta(x,t),t)) \\ \|v_\theta(z_\theta(x,t),t)\|^2 \end{pmatrix}, \qquad \begin{pmatrix} z_\theta(x,0) \\ \ell_\theta(x,0) \\ \omega_\theta(x,0) \end{pmatrix} = \begin{pmatrix} x \\ 0 \\ 0 \end{pmatrix}. \tag{6}$$

In particular, the loss function $\mathcal{L}$ can be evaluated and differentiated based on samples from $\mu_\tau^k$. Once the parameters $\theta$ are optimized, we can evaluate the JKO steps in the same way as standard continuous normalizing flows. We summarize training and evaluation of the JKO steps in Algorithm 2 and 3 in Appendix D.1. In practice, the density values are computed and stored in log-space for numerical stability. Additionally, note that the continuous normalizing flows can be replaced by other normalizing flow architectures, see Appendix F.2 and Remark 19 for details.

## 4 IMPORTANCE-BASED REJECTION STEPS

While a large number of existing sampling methods rely on Wasserstein gradient flows with respect to some divergences, it is well known that these loss functions are non-convex. This leads to very slow divergence speeds or convergence to suboptimal local minima. In particular, if the target distribution is multimodal the modes often do not have the right mass assigned.

**Annealed Importance Sampling** As a remedy, Neal (2001) proposed to use annealed importance sampling. Here, we assign to each generated sample $x_i$ an importance weight $w_i = \frac{q(x_i)}{p(x_i)}$, where $p$ is some proposal density and $q$ is the density of the target distribution $\nu$. Then, for any $\nu$-integrable function $f\colon \mathbb{R}^d \to \mathbb{R}$ it holds that $\sum_{i=1}^N w_i f(x_i)$ is an unbiased estimator of $\int_{\mathbb{R}^d} f(x)\mathrm{d}\nu(x)$. Note that annealed importance sampling is very sensitive with respect to the proposal $p$ which needs to be designed carefully and problem adapted.

**Rejection Steps** Inspired by annealed importance sampling, we propose to use importance-based rejection steps. More precisely, let $\mu$ be a proposal distribution where we can sample from with density $p(x) = f(x)/Z_f$ and denote by $\nu$ the target distribution with density $q(x) = g(x)/Z_g$. In the following, we assume that we have access to the unnormalized densities $f$ and $g$, but not to the normalization constants $Z_f$ and $Z_g$ Then, for a random variable $X \sim \mu$, we now generate a new random variable $\tilde{X}$ by the following procedure: First, we compute the importance based acceptance

probability $\alpha(X) = \min\left\{1, \frac{q(X)}{\tilde{c}p(X)}\right\} = \min\left\{1, \frac{g(X)}{cf(X)}\right\}$, where $c > 0$ is a positive hyperparameter and $\tilde{c} = cZ_f/Z_g$. Then, we set $\tilde{X} = X$ with probability $\alpha(X)$ and choose $\tilde{X} = X'$ otherwise, where $X' \sim \mu$ and $X$ are independent, see Algorithm 5 for a summary.

**Remark 7.** *This is a one-step approximation of the classical rejection sampling scheme (Von Neumann, 1951), see also (Andrieu et al., 2003) for an overview. More precisely, we arrive at the classical rejection sampling scheme by choosing $\tilde{c} > \sup_x q(x)/p(x)$ and redo the procedure when $X$ is rejected instead of choosing $\tilde{X} = X'$.*

Similarly to importance sampling, the rejection sampling algorithm is highly sensitive towards the proposal distribution $p$. In particular, if $p$ is chosen as a standard normal distribution, it suffers from the curse of dimensionality. We will tackle this problem later in the section by choosing $p$ already close to the target density $q$.

The following proposition describes the density of the distribution $\tilde{\mu}$ of $\tilde{X}$. Moreover, it ensures that the KL divergence to the target distribution decreases. We include the proof in Appendix C.1

**Proposition 8.** *Let $\tilde{\mu}$ be the distribution of $\tilde{X}$. Then, the following holds true.*

*(i) $\tilde{\mu}$ admits the density $\tilde{p}$ given by $\tilde{p}(x) = p(x)(\alpha(x) + 1 - \mathbb{E}[\alpha(X)])$. In particular, we have $\tilde{p}(x) = \tilde{f}(x)/Z_{\tilde{f}}$ with $\tilde{f}(x) = f(x)(\alpha(x) + 1 - \mathbb{E}[\alpha(X)])$.*

*(ii) It holds that $\mathrm{KL}(\tilde{\mu}, \nu) \leq \mathrm{KL}(\mu, \nu)$.*

Note that the value $\mathbb{E}[\alpha(X)]$ can easily be estimated based on samples during the training phase. Indeed, given $N$ iid copies $X_1, ..., X_N$ of $X$, we obtain that $\mathbb{E}[\alpha(X)] \approx \frac{1}{N}\sum_{i=1}^{N}\alpha(X_N)$ is an unbiased estimator fulfilling the error estimate from the following corollary. The proof is a direct consequence of Hoeffding's inequality and given in Appendix C.2.

**Corollary 9.** *Let $X_1, ..., X_N$ be iid copies of $X$. Then, it holds*

$$\mathbb{E}\left[\left|\mathbb{E}[\alpha(X)] - \frac{1}{N}\sum_{i=1}^{N}\alpha(X_N)\right|\right] \leq \frac{\sqrt{2\pi}}{\sqrt{N}} \in O\left(\frac{1}{\sqrt{N}}\right).$$

**Remark 10** (Choice of $c$). *We choose the hyperparameter $c$ such that a constant ration $r > 0$ of the samples will be resampled, i.e., that $\mathbb{E}[\alpha(X)] \approx 1 - r$. To this end, we assume that we are given samples $x_1, ..., x_N$ from $X$ and approximate*

$$\mathbb{E}[\alpha(X)] \approx \frac{1}{N}\sum_{i=1}^{N}\alpha(x_i) = \frac{1}{N}\sum_{i=1}^{N}\min\left\{1, \frac{g(x)}{cp(x)}\right\}.$$

*Note that the right side of this formula depends monotonically on $c$ such that we can find $c > 0$ such that $\mathbb{E}[\alpha(X)] = 1 - r$ by a bisection search. In our numerical experiments, we set $r = 0.2$.*

We summarize the parameter selection and application of a rejection step we in the Algorithms 4 and 5 in Appendix D.1.

**Neural JKO Sampling with Importance Correction** Finally, we combine the neural JKO scheme from the previous section with our rejection steps to obtain a sampling algorithm. More precisely, we start with a simple latent random variable $X_0$ following the probability distribution $\mu_0$ with known density and where we can sample from. In our numerical experiments this will be a standard Gaussian distribution. Now, we iteratively generate random variables $X_k$ following the distribution $\mu_k$, $k = 1, ..., K$ by applying either neural JKO steps as described in Algorithm 2 and 3 or importance-based rejection steps as described in Algorithm 4 and 5. We call the resulting Markov chain $(X_0, ..., X_K)$ an importance corrected neural JKO model. During the sampling process we can maintain the density values $p^k(x)$ of the density $p^k$ of $\mu_k$ for the generated samples by the Algorithms 3 and 5. Vice versa, we can also use Proposition 8 to evaluate the density $p^k$ at some arbitrary point $x \in \mathbb{R}^d$. We outline this density evaluation process in Appendix D.2.

**Remark 11** (Runtime Limitations). *The sampling time of our importance corrected neural JKO sampling depends exponentially on the number of rejection steps since in each rejection step we resample a constant fraction of the samples. However, due to the moderate base of $1 + r$ we will see in the numerical part that we are able to perform a significant number of rejection steps in a tractable time.*

---

**Algorithm 1** Importance corrected neural JKO sampling scheme

**Input:**
- *target measure $\nu = Z_g^{-1} g \mathrm{d}\lambda$ with unnormalized density $g$,*
- *initial measure $\mu^0$ with density $p_0$,*
- *Number $N \in \mathbb{N}$ of samples during for learning phase,*
- *Number $K \in \mathbb{N}$ of total steps.*

**Output:** *Sample generator $\{x^i\}_{i=1}^N \sim \hat{\nu} \approx \nu$.*

  *Let density $p_0$ be the density of $\mu^0$ with samples $x_1^0, \ldots, x_N^0$.*
  **for** $k = 1, \ldots, K$ **do**
    *Define $\mu^k$ with a density $p^k$ either as*
      • *push forward of $\mu^{k-1}$ via a CNF as described in Section 3 with parameters*
        *learned via Algorithm 2 based on samples $x_1^{k-1}, \ldots x_N^{k-1}$ or*
      • *via importance based rejection with $p^k$ determined by Proposition 8 using*
        *Algorithm 4 to learn the parameter $c$ as discussed in Remark 10.*
    *Generate samples $x_1^k, \ldots, x_N^k \sim \mu^k$ and densities $p^k(x_1^k,) \ldots, p^k(x_N^k)$ using*
    *Algorithm 3 or 5 depending on the choice of the propagation layers respectively.*
  **end for**
  *Set $\hat{\nu} = \mu^K$ with density $p^K$.*

---

As discussed in Algorithm 1 the final sampling model structure is determined by the sub-sequential choice of a CNF layer or an importance-based rejection step. In practice, we first utilize CNF layers only, then use several blocks consisting of a single CNF layer composed with 3 rejection steps. In our numerics, we choose an initial step size $\tau_0 > 0$ as a hyper-parameter and then increase the step size exponentially by $\tau_{k+1} = 4\tau_k$. Note that one could alternatively use adaptive step sizes similar to Xu et al. (2024). However, for our setting, we found that the simple step size rule is sufficient.

## 5 NUMERICAL RESULTS

We compare our method with classical Monte Carlo samplers like a Metropolis adjusted Langevin sampling (MALA) and Hamiltonian Monte Carlo (HMC), see e.g., Betancourt (2017); Roberts & Tweedie (1996). Additionally, we compare with two recent deep-learning based sampling algorithms, namely denoising diffusion samplers (DDS, Vargas et al., 2023a) and continual repeated annealed flow transport Monte Carlo (CRAFT, Matthews et al., 2022). We evaluate all methods on a set of common test distributions which is described in detail in Appendix E.1. Moreover, we report the error measures for our *importance corrected neural JKO* sampler (neural JKO IC), see Appendix E.4 for implementation details. Additionally, we emphasize the importance of the rejection steps by reporting values for the same a neural JKO scheme without rejection steps (neural JKO).

For evaluating the quality of our results, we use two different metrics. First, we evaluate the energy distance (Székely, 2002). It is a kernel metrics which can be evaluated purely based on two sets of samples from the model and the ground truth. Moreover it encodes the geometry of the space such that a slight perturbation of the samples only leads to a slight change in the energy distance. Second, we estimate the log-normalizing constant which is equivalent to approximating the reverse KL loss of the model. A higher estimate of the log-normalizing constant corresponds to a smaller KL divergence between generated and target distribution and therefore to a higher similarity of the two measures. Since this requires the density of the model, this approach is not applicable for MALA and HMC. The results are given in Table 1 and 2. We can see, that importance corrected neural JKO sampling significantly outperforms the comparison for all test distributions. In particular, we observe that for shifted 8 modes, shifted 8 peaky and GMM-$d$ the energy distance between neural JKO IC and ground truth samples is in the same order of magnitude as the energy distance between to different sets of ground truth samples. This implies that the distribution generated by neural JKO IC is indistinguishable from the target distribution in the energy distance. For these examples, the $\log(Z)$ esitmate is sometimes slightly larger than the ground truth, which can be explained by numerical effects, see Remark 16 for a detailed discussion. We point out to a precise description of the metrics, the implementation details and additional experiments and figures in Appendix E.

Table 1: Energy distance. We run each method 5 times and state the average value and corresponding standard deviations. The rightmost column shows the reference sampling error, i.e., the lower bound magnitude of the average energy distance between two different sets of samples drawn from the ground truth. A smaller energy distance indicates a better result.

| | Sampler | | | | | | |
|---|---|---|---|---|---|---|---|
| Distribution | MALA | HMC | DDS | CRAFT | Neural JKO | Neural JKO IC (**ours**) | Sampling Error |
| Mustache | $4.6 \times 10^{-2} \pm 1.6 \times 10^{-3}$ | $1.7 \times 10^{-2} \pm 4.3 \times 10^{-4}$ | $6.9 \times 10^{-2} \pm 1.8 \times 10^{-3}$ | $9.2 \times 10^{-2} \pm 9.9 \times 10^{-3}$ | $1.8 \times 10^{-2} \pm 2.0 \times 10^{-3}$ | $2.9 \times 10^{-3} \pm 4.4 \times 10^{-4}$ | $8.6 \times 10^{-5}$ |
| shifted 8 Modes | $5.3 \times 10^{-3} \pm 4.9 \times 10^{-4}$ | $4.1 \times 10^{-5} \pm 3.3 \times 10^{-5}$ | $1.2 \times 10^{-2} \pm 4.1 \times 10^{-3}$ | $5.2 \times 10^{-2} \pm 1.1 \times 10^{-2}$ | $1.3 \times 10^{-1} \pm 3.8 \times 10^{-3}$ | $1.2 \times 10^{-5} \pm 5.1 \times 10^{-6}$ | $2.6 \times 10^{-5}$ |
| shifted 8 Peaky | $1.3 \times 10^{-1} \pm 3.2 \times 10^{-3}$ | $1.2 \times 10^{-1} \pm 2.4 \times 10^{-3}$ | $1.1 \times 10^{-2} \pm 3.4 \times 10^{-3}$ | $5.2 \times 10^{-2} \pm 2.2 \times 10^{-2}$ | $1.3 \times 10^{-1} \pm 2.2 \times 10^{-3}$ | $3.4 \times 10^{-5} \pm 8.2 \times 10^{-6}$ | $2.4 \times 10^{-5}$ |
| Funnel | $1.2 \times 10^{-1} \pm 3.1 \times 10^{-3}$ | $3.1 \times 10^{-3} \pm 3.2 \times 10^{-4}$ | $2.6 \times 10^{-1} \pm 2.6 \times 10^{-2}$ | $7.4 \times 10^{-2} \pm 2.8 \times 10^{-3}$ | $4.6 \times 10^{-2} \pm 1.6 \times 10^{-3}$ | $1.4 \times 10^{-2} \pm 8.2 \times 10^{-4}$ | $3.4 \times 10^{-4}$ |
| GMM-10 | $1.2 \times 10^{-2} \pm 5.5 \times 10^{-3}$ | $1.2 \times 10^{-2} \pm 5.2 \times 10^{-3}$ | $3.7 \times 10^{-3} \pm 1.6 \times 10^{-3}$ | $1.8 \times 10^{-1} \pm 6.6 \times 10^{-2}$ | $1.1 \times 10^{-2} \pm 5.6 \times 10^{-3}$ | $5.3 \times 10^{-5} \pm 1.7 \times 10^{-5}$ | $4.6 \times 10^{-5}$ |
| GMM-20 | $9.1 \times 10^{-3} \pm 2.8 \times 10^{-3}$ | $9.1 \times 10^{-3} \pm 2.7 \times 10^{-3}$ | $5.0 \times 10^{-3} \pm 1.5 \times 10^{-3}$ | $5.4 \times 10^{-1} \pm 1.4 \times 10^{-1}$ | $1.0 \times 10^{-2} \pm 2.8 \times 10^{-3}$ | $1.1 \times 10^{-4} \pm 3.4 \times 10^{-5}$ | $6.4 \times 10^{-5}$ |
| GMM-50 | $2.4 \times 10^{-2} \pm 7.5 \times 10^{-3}$ | $2.4 \times 10^{-2} \pm 7.5 \times 10^{-3}$ | $2.3 \times 10^{-2} \pm 1.1 \times 10^{-2}$ | $1.8 \times 10^{0} \pm 1.7 \times 10^{-1}$ | $2.7 \times 10^{-2} \pm 7.8 \times 10^{-3}$ | $1.0 \times 10^{-4} \pm 4.6 \times 10^{-5}$ | $1.1 \times 10^{-4}$ |
| GMM-100 | $3.6 \times 10^{-2} \pm 1.6 \times 10^{-2}$ | $3.7 \times 10^{-2} \pm 1.7 \times 10^{-2}$ | $3.9 \times 10^{-2} \pm 2.1 \times 10^{-2}$ | $2.8 \times 10^{+1} \pm 1.0 \times 10^{-1}$ | $4.7 \times 10^{-2} \pm 2.2 \times 10^{-2}$ | $6.0 \times 10^{-4} \pm 3.4 \times 10^{-4}$ | $1.5 \times 10^{-4}$ |
| GMM-200 | $6.4 \times 10^{-2} \pm 2.1 \times 10^{-2}$ | $6.6 \times 10^{-2} \pm 1.9 \times 10^{-2}$ | $9.8 \times 10^{-2} \pm 3.1 \times 10^{-2}$ | $3.9 \times 10^{0} \pm 1.6 \times 10^{-1}$ | $8.9 \times 10^{-2} \pm 2.7 \times 10^{-2}$ | $3.3 \times 10^{-3} \pm 1.9 \times 10^{-3}$ | $2.0 \times 10^{-4}$ |

Table 2: Estimated $\log(Z)$. We run each method 5 times and state the average value and corresponding standard deviations. The DDS values for LGCP are taken from Vargas et al. (2023a). Higher values of $\log(Z)$ estimates correspond to better results.

| | Sampler | | | | |
|---|---|---|---|---|---|
| Distribution | DDS | CRAFT | Neural JKO | Neural JKO IC (**ours**) | Ground Truth |
| Mustache | $-1.5 \times 10^{-1} \pm 2.7 \times 10^{-2}$ | $-6.5 \times 10^{-2} \pm 5.5 \times 10^{-2}$ | $-3.0 \times 10^{-2} \pm 2.6 \times 10^{-3}$ | $-7.3 \times 10^{-3} \pm 8.2 \times 10^{-4}$ | $0$ |
| shifted 8 Modes | $-5.7 \times 10^{-2} \pm 2.0 \times 10^{-2}$ | $-1.2 \times 10^{-2} \pm 1.4 \times 10^{-3}$ | $-3.4 \times 10^{-1} \pm 3.1 \times 10^{-3}$ | $+5.1 \times 10^{-6} \pm 2.4 \times 10^{-3}$ | $0$ |
| shifted 8 Peaky | $-1.2 \times 10^{-1} \pm 2.2 \times 10^{-2}$ | $-1.8 \times 10^{-3} \pm 2.6 \times 10^{-3}$ | $-3.5 \times 10^{-1} \pm 3.1 \times 10^{-3}$ | $-2.1 \times 10^{-3} \pm 3.2 \times 10^{-3}$ | $0$ |
| Funnel | $-1.8 \times 10^{-1} \pm 6.8 \times 10^{-2}$ | $-1.2 \times 10^{-1} \pm 7.9 \times 10^{-3}$ | $-1.4 \times 10^{-1} \pm 1.6 \times 10^{-3}$ | $-7.1 \times 10^{-3} \pm 1.9 \times 10^{-3}$ | $0$ |
| GMM-10 | $-2.3 \times 10^{-1} \pm 1.0 \times 10^{-1}$ | $-8.5 \times 10^{-1} \pm 1.7 \times 10^{-1}$ | $-4.3 \times 10^{-1} \pm 5.1 \times 10^{-2}$ | $+3.5 \times 10^{-3} \pm 2.0 \times 10^{-3}$ | $0$ |
| GMM-20 | $-5.1 \times 10^{-1} \pm 6.0 \times 10^{-2}$ | $-1.5 \times 10^{0} \pm 1.7 \times 10^{-1}$ | $-6.3 \times 10^{-1} \pm 2.7 \times 10^{-2}$ | $+6.4 \times 10^{-3} \pm 3.8 \times 10^{-3}$ | $0$ |
| GMM-50 | $-1.3 \times 10^{0} \pm 3.3 \times 10^{-1}$ | $-2.3 \times 10^{0} \pm 1.5 \times 10^{-3}$ | $-9.3 \times 10^{-1} \pm 4.6 \times 10^{-2}$ | $+1.1 \times 10^{-2} \pm 3.9 \times 10^{-3}$ | $0$ |
| GMM-100 | $-3.0 \times 10^{0} \pm 7.3 \times 10^{-1}$ | $-2.3 \times 10^{0} \pm 8.6 \times 10^{-2}$ | $-1.8 \times 10^{0} \pm 9.6 \times 10^{-2}$ | $-3.9 \times 10^{-2} \pm 7.8 \times 10^{-3}$ | $0$ |
| GMM-200 | $-9.4 \times 10^{0} \pm 7.2 \times 10^{-1}$ | $-6.3 \times 10^{0} \pm 1.5 \times 10^{-1}$ | $-5.2 \times 10^{0} \pm 2.5 \times 10^{-1}$ | $-5.6 \times 10^{-2} \pm 1.3 \times 10^{-2}$ | $0$ |
| LGCP | $503.0 \pm 7.7 \times 10^{-1}$ | $507.6 \pm 3.2 \times 10^{-1}$ | $499.9 \pm 1.7 \times 10^{-1}$ | $508.2 \pm 1.0 \times 10^{-1}$ | not available |

## 6 CONCLUSIONS

**Methodology** We proposed a novel and expressive generative method that enables the efficient and accurate sampling from a prescribed unnormalized target density which is empirically confirmed in numerical examples. To this end, we combine *local sampling steps*, relying on piecewise geodesic interpolations of the JKO scheme realized by CNFs, and *non-local rejection and resampling steps* based on importance weights. Since the proposal of the rejection step is generated by the model itself, they do not suffer from the curse of dimensionality as opposed to classical variants of rejection sampling. The proposed approach provides the advantage that we can draw *independent* samples while correcting imbalanced mode weights, iteratively refine the current approximation and evaluate the density of the generated distribution. The latter property is a consequence of the *density value propagation through CNFs* and Proposition 8 and enables possible further post-processing steps that require density evaluations of the approximated sample process.

**Outlook** Our method allows for the pointwise access to the approximated target density and the log normalization constant. These quantities can be used for the error monitoring during the training and hence provide guidelines for the adaptive design of the emulator in terms of CNF -or rejection/resampling steps and provide a straightforward stopping criterion.

**Limitations** In the situation, when the emulator is realized through a stack of underlying rejection/resampling steps, the sample generation process time is negatively affected, see Remark 11. In order to resolve the drawback we plan to utilize diffusion models for the sample generation. This is part of ongoing and future work by the authors. Finally, the use of continuous normalizing flows comes with computational challenges, which we discuss in detail in Appendix F.2.

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

## A    Background on Wasserstein Spaces

We give some more backgrounds on Wasserstein gradient flows extending Section 2. In what follows we refer to $L_2(\mu) := L_2(\mu, \mathbb{R}^d)$ as the set of square integrable measurable functions on $\mathbb{R}^d$ with respect to a given measure $\mu \in \mathcal{P}(\mathbb{R}^d)$.

For an absolutely continuous curve $\gamma$, there exists a unique minimal norm solution $v_t$ of the continuity equation (1) in the sense that any solution $\tilde{v}$ of (1) fulfills $\|\tilde{v}(\cdot, t)\|_{L_2(\gamma(t))} \geq \|v(\cdot, t)\|_{L_2(\gamma(t))}$ for almost every $t$. This is the unique solution of (1) such that $v(\cdot, t)$ is contained in the regular tangent space

$$\mathrm{T}_\mu \mathcal{P}_2(\mathbb{R}^d) := \overline{\{\lambda(T - \mathrm{Id}) : (\mathrm{Id}, T)_{\#}\mu \in \Gamma^{\mathrm{opt}}(\mu, T_{\#}\mu), \ \lambda > 0\}}^{L_2(\mu)}.$$

An absolutely continuous curve $\gamma \colon (0, \infty) \to \mathcal{P}_2(\mathbb{R}^d)$ with velocity field $v_t \in \mathrm{T}_{\gamma(t)}\mathcal{P}_2(\mathbb{R}^d)$ is a *Wasserstein gradient flow with respect to $\mathcal{F} \colon \mathcal{P}_2(\mathbb{R}^d) \to (-\infty, \infty]$* if

$$v_t \in -\partial \mathcal{F}(\gamma(t)), \quad \text{for a.e. } t > 0,$$

where $\partial \mathcal{F}(\mu)$ denotes the reduced Fréchet subdiffential at $\mu$ defined as

$$\partial \mathcal{F}(\mu) := \left\{ \xi \in L_2(\mu) : \mathcal{F}(\nu) - \mathcal{F}(\mu) \geq \inf_{\pi \in \Gamma^{\mathrm{opt}}(\mu, \nu)} \int_{\mathbb{R}^d \times \mathbb{R}^d} \langle \xi(x), y - x \rangle \, \mathrm{d}\pi(x, y) + o(W_2(\mu, \nu)) \ \forall \nu \in \mathcal{P}_2(\mathbb{R}^d) \right\}.$$

The norm $\|v_t\|_{L_2(\gamma(t))}$ of the velocity field of a Wasserstein gradient flow coincides for almost every $t$ with the metric derivative

$$|\partial \mathcal{F}|(\mu) = \inf_{\nu \to \mu} \frac{\mathcal{F}(\mu) - \mathcal{F}(\nu)}{W_2(\mu, \nu)}.$$

For the convergence result from Theorem 3, we need two more definitions. First, we need some convexity assumption. For $\lambda \in \mathbb{R}$, $\mathcal{F} \colon \mathcal{P}_2(\mathbb{R}^d) \to \mathbb{R} \cup \{+\infty\}$ is called *$\lambda$-convex along geodesics* if, for every $\mu, \nu \in \mathrm{dom}\, \mathcal{F} := \{\mu \in \mathcal{P}_2(\mathbb{R}^d) : \mathcal{F}(\mu) < \infty\}$, there exists at least one geodesics $\gamma \colon [0, 1] \to \mathcal{P}_2(\mathbb{R}^d)$ between $\mu$ and $\nu$ such that

$$\mathcal{F}(\gamma(t)) \leq (1 - t)\, \mathcal{F}(\mu) + t\, \mathcal{F}(\nu) - \tfrac{\lambda}{2}\, t(1 - t)\, W_2^2(\mu, \nu), \qquad t \in [0, 1].$$

To ensure uniqueness and convergence of the JKO scheme, a slightly stronger condition, namely being *$\lambda$-convex along generalized geodesics* will be in general needed. Based on the set of three-plans with base $\sigma \in \mathcal{P}_2(\mathbb{R}^d)$ given by

$$\Gamma_\sigma(\mu, \nu) := \left\{ \boldsymbol{\alpha} \in \mathcal{P}_2(\mathbb{R}^d \times \mathbb{R}^d \times \mathbb{R}^d) : (\pi_1)_{\#}\boldsymbol{\alpha} = \sigma, (\pi_2)_{\#}\boldsymbol{\alpha} = \mu, (\pi_3)_{\#}\boldsymbol{\alpha} = \nu \right\},$$

the so-called *generalized geodesics* $\gamma \colon [0, \epsilon] \to \mathcal{P}_2(\mathbb{R}^d)$ joining $\mu$ and $\nu$ (with base $\sigma$) is defined as

$$\gamma(t) := \left( (1 - \tfrac{t}{\epsilon})\pi_2 + \tfrac{t}{\epsilon}\pi_3 \right)_{\#} \boldsymbol{\alpha}, \qquad t \in [0, \epsilon], \tag{7}$$

where $\boldsymbol{\alpha} \in \Gamma_\sigma(\mu, \nu)$ with $(\pi_{1,2})_{\#}\boldsymbol{\alpha} \in \Gamma^{\mathrm{opt}}(\sigma, \mu)$ and $(\pi_{1,3})_{\#}\boldsymbol{\alpha} \in \Gamma^{\mathrm{opt}}(\sigma, \nu)$, see Definition 9.2.2 in Ambrosio et al. (2005). The plan $\boldsymbol{\alpha}$ may be interpreted as transport from $\mu$ to $\nu$ via $\sigma$. Then a function $\mathcal{F} \colon \mathcal{P}_2(\mathbb{R}^d) \to (-\infty, \infty]$ is called *$\lambda$-convex along generalized geodesics* (see Ambrosio et al., 2005, Definition 9.2.4), if for every $\sigma, \mu, \nu \in \mathrm{dom}\, \mathcal{F}$, there exists at least one generalized geodesics $\gamma \colon [0, 1] \to \mathcal{P}_2(\mathbb{R}^d)$ related to some $\boldsymbol{\alpha}$ in (7) such that

$$\mathcal{F}(\gamma(t)) \leq (1 - t)\, \mathcal{F}(\mu) + t\, \mathcal{F}(\nu) - \tfrac{\lambda}{2}\, t(1 - t)\, W_{\boldsymbol{\alpha}}^2(\mu, \nu), \qquad t \in [0, 1],$$

where

$$W_{\boldsymbol{\alpha}}^2(\mu, \nu) := \int_{\mathbb{R}^d \times \mathbb{R}^d \times \mathbb{R}^d} \|y - z\|_2^2 \, \mathrm{d}\boldsymbol{\alpha}(x, y, z).$$

Every function being $\lambda$-convex along generalized geodesics is also $\lambda$-convex along geodesics since generalized geodesics with base $\sigma = \mu$ are actual geodesics. Second, a $\lambda$-convex functional $\mathcal{F} \colon \mathcal{P}_2(\mathbb{R}^d) \to \mathbb{R} \cup \{+\infty\}$ is called coercive, if there exists some $r > 0$ such that

$$\inf\{\mathcal{F}(\mu) : \mu \in \mathcal{P}_2(\mathbb{R}^d), \int_{\mathbb{R}^d} \|x\|^2 \mathrm{d}\mu(x) \leq r\} > -\infty,$$

see (Ambrosio et al., 2005, eq. (11.2.1b)). In particular, any functional which is bounded from below is coercive.

If $\mathcal{F}$ is proper, lower semicontinuous, coercive and $\lambda$-convex along generalized geodesics, one can show that the $\mathrm{prox}_{\tau \mathcal{F}}(\mu)$ is non-empty and unique for $\tau$ small enough (see Ambrosio et al., 2005, page 295).

## B PROOFS AND EXAMPLES FROM SECTION 3

### B.1 EXAMPLES FULFILLING ASSUMPTION 4

Assumption 4 is fulfilled for many important divergences and loss functions $\mathcal{F}$. We list some examples below. While it is straightforward to check that they are proper, lower semicontinuous, coercive and bounded from below, the convexity is non-trivial. However, the conditions under which these functionals are $\lambda$-convex along generalized geodesics are well investigated in (Ambrosio et al., 2005, Section 9.3). In the following we denote the Lebesque measure as $\lambda$. For a measure $\mu \in \mathcal{P}^{\mathrm{ac}}(\mathbb{R}^d)$, we denote by $\mathrm{d}\mu/\mathrm{d}\lambda$ the Lebesque density of $\mu$ if it exists.

- Let $\nu \in \mathcal{P}_2^{\mathrm{ac}}(\mathbb{R}^d)$ with Lebesque density $q$ and define the *forward Kullback-Leibler (KL) loss function*

$$\mathcal{F}(\mu) = \mathrm{KL}(\nu, \mu) := \begin{cases} \int_{\mathbb{R}^d} q(x) \log\left(\frac{q(x)}{p(x)}\right) \mathrm{d}x, & \text{if } \exists \mathrm{d}\mu/\mathrm{d}\lambda = p \text{ and } \mathrm{d}\nu/\mathrm{d}\lambda = q, \\ +\infty & \text{otherwise.} \end{cases}$$

By (Ambrosio et al., 2005, Proposition 9.3.9), we obtain that $\mathcal{F}$ fulfills Assumption 4.

- We can also derive a functional, be reversing the arguments in the KL divergence. Then, we arrive at the *reverse KL loss function* given by

$$\mathcal{F}(\mu) = \mathrm{KL}(\mu, \nu) := \begin{cases} \int_{\mathbb{R}^d} p(x) \log\left(\frac{p(x)}{q(x)}\right) \mathrm{d}x, & \text{if } \exists \mathrm{d}\mu/\mathrm{d}\lambda = p \text{ and } \mathrm{d}\nu/\mathrm{d}\lambda = q, \\ +\infty & \text{otherwise.} \end{cases}$$

Given that $q$ is $\lambda$-convex, we obtain that $\mathcal{F}$ fulfills Assumption 4, see (Ambrosio et al., 2005, Proposition 9.3.2).

- Finally, we can define $\mathcal{F}$ based on the Jensen-Shannon divergence. This results into the function

$$\mathcal{F}(\mu) = \mathrm{JS}(\mu, \nu) := \frac{1}{2}\left[\mathrm{KL}\left(\mu, \tfrac{1}{2}(\mu + \nu)\right) + \mathrm{KL}\left(\nu, \tfrac{1}{2}(\mu + \nu)\right)\right].$$

Assume $\mu$ and $\nu$ admit Lebesque densities $p$ and $q$ respectively. Then, combining the two previous statements, this fulfills Assumption 4 whenever $p$ and $q$ are $\lambda$-convex.

All of these functionals are integrals of a smooth Lagrangian functional, i.e., there exists some smooth $F \colon \mathbb{R}^d \times \mathbb{R} \times \mathbb{R}^d \to \mathbb{R}$ such that

$$\mathcal{F}(\mu) = \begin{cases} \int_{\mathbb{R}^d} F(x, p(x), \nabla p(x)) \mathrm{d}x, & \text{if } \exists \mathrm{d}\mu/\mathrm{d}\lambda = p, \\ \infty & \text{otherwise.} \end{cases}$$

In this case, the limit velocity field from Theorem 3 (which appears as a limit in Theorem 6) can be expressed analytically as the gradient of the so-called variational derivative of $\mathcal{F}$, which is given by

$$\frac{\delta \mathcal{F}}{\delta \gamma(t)}(x) = -\nabla \cdot \partial_3 F(x, p(x), \nabla p(x)) + \partial_2 F(x, p(x), \nabla p(x))$$

where $\partial_i F$ is the derivative of $F$ with respect to the $i$-th argument and $\gamma$ is the Wasserstein gradient flow, see (Ambrosio et al., 2005, Example 11.1.2). For the above divergence functionals, computing these terms lead to a (weighted) difference of the Stein scores of the input measure $\gamma(t)$ and the target measure $\nu$ which is a nice link to score-based methods. More precisely, denoting the density of $\gamma(t)$ by $p_t$, we obtain the following limiting velocity fields.

- For the forward KL loss function we have that $F(x, y, z) = q(x) \log\left(\frac{q(x)}{y}\right)$. Thus, we have that $\frac{\delta \mathcal{F}}{\delta \mu}(x) = -\frac{q(x)}{p(x)}$. Hence, the velocity field $v(\cdot, t) = \nabla \frac{\delta \mathcal{F}}{\delta \gamma(t)}$ is given by

$$v(x, t) = \frac{q(x)}{p_t(x)} \frac{\nabla p_t(x)}{p_t(x)} - \frac{q(x)}{p_t(x)} \frac{\nabla q(x)}{q(x)} = \frac{q(x)}{p_t(x)} \left(\nabla \log(p_t(x)) - \nabla \log(q(x))\right).$$

- For the reverse KL loss function the Lagrangian is given by $F(x, y, z) = y \log\left(\frac{y}{q(x)}\right)$. Thus, we have that $\frac{\delta\mathcal{F}}{\delta\mu}(x) = \log\left(\frac{p(x)}{q(x)}\right) + 1 = \log(p(x)) - \log(q(x)) + 1$. Hence, the velocity field $v(\cdot, t) = \nabla \frac{\delta\mathcal{F}}{\delta\gamma(t)}$ is given by

$$v(x, t) = \nabla(\log(p_t)(x)) - \nabla(\log(q)(x)).$$

- For the Jensen-Shannon divergence the Lagrangian is given by $F(x, y, z) = \frac{1}{2}\left(y \log\left(\frac{2y}{y+q(x)}\right) + q(x)\left(\frac{2q(x)}{y+q(x)}\right)\right)$. Thus, we have that $\frac{\delta\mathcal{F}}{\delta\mu}(x) = \frac{1}{2}\log\left(\frac{p(x)}{p(x)+q(x)}\right)$. Hence, the velocity field $v(\cdot, t) = \nabla\frac{\delta\mathcal{F}}{\delta\gamma(t)}$ is given by

$$
\begin{aligned}
v(x, t) &= \frac{1}{2}\frac{p_t(x) + q(x)}{p_t(x)}\frac{\nabla p_t(x)(p_t(x) + q(x)) - p_t(x)(\nabla p_t(x) + \nabla q(x))}{(p_t(x) + q(x))^2} \\
&= \frac{1}{2}\frac{\nabla p_t(x)q(x) - p_t(x)\nabla q(x)}{(p_t(x) + q(x))p_t(x)} \\
&= \frac{q(x)}{2(p_t(x) + q(x))}\left[\frac{\nabla p_t(x)}{p_t(x)} - \frac{\nabla q(x)}{q(x)}\right] \\
&= \frac{q(x)}{2(p_t(x) + q(x))}\left[\nabla(\log(p_t)(x)) - \nabla(\log(q)(x))\right].
\end{aligned}
$$

Note that computing the score $\nabla \log(p_t)$ of the current approximation is usually intractable, such that these limits cannot be inserted into a neural ODE directly.

### B.2 PROOF OF COROLLARY 5

Using similar arguments as in Altekrüger et al. (2023); Mokrov et al. (2021); Onken et al. (2021); Xu et al. (2024), let $v : \mathbb{R}^d \times [0, \tau] \to \mathbb{R}^d$ such that the solution of

$$\dot{z}(x, t) = v(z(x, t), t), \quad z(x, 0) = x,$$

fulfills $z(\cdot, \tau)_{\#}\mu_\tau^k = \mu_\tau^{k+1}$. Since $(v_{\tau,k}, z_{\tau,k})$ is a minimizer of (4), we obtain that

$$
\begin{aligned}
&\frac{1}{2}\int_0^\tau \int_{\mathbb{R}^d} \|v_{\tau,k}(z_{\tau,k}(x, t), t)\|^2 \mathrm{d}\mu_\tau^k(x)\mathrm{d}t + \mathcal{F}(z_{\tau,k}(\cdot, \tau)_{\#}\mu_\tau^k) \\
&\leq \frac{1}{2}\int_0^\tau \int_{\mathbb{R}^d} \|v(z(x, t), t)\|^2 \mathrm{d}\mu_\tau^k(x)\mathrm{d}t + \mathcal{F}(z(\cdot, \tau)_{\#}\mu_\tau^k).
\end{aligned}
$$

Observing that $\mathcal{F}(z_{\tau,k}(\cdot, \tau)_{\#}\mu_\tau^k) = \mathcal{F}(z(\cdot, \tau)_{\#}\mu_\tau^k) = \mathcal{F}(\mu_\tau^{k+1})$, we obtain that

$$\tau\int_0^\tau \int_{\mathbb{R}^d} \|v_{\tau,k}(z_{\tau,k}(x, t), t)\|^2 \mathrm{d}\mu_\tau^k(x) \leq \tau\int_0^\tau \int_{\mathbb{R}^d} \|v(z(x, t), t)\|^2 \mathrm{d}\mu_\tau^k(x)\mathrm{d}t.$$

Since $v$ was chose arbitrary, we obtain that $v_{\tau,k}$ is the optimal velocity field from the theorem of Benamou-Brenier, which now directly implies part (ii) and (iii). □

### B.3 PROOF OF THEOREM 6

In order to prove convergence of the velocity fields $v_\tau$, we first introduce some notations. To this end, let $\mathcal{T}_\tau^k$ be the optimal transport maps between $\mu_\tau^k$ and $\mu_\tau^{k+1}$ and define by $v_\tau^k = (\mathcal{T}_\tau^k - I)/\tau$ the corresponding discrete velocity fields. Then, the velocity fields $v_\tau$ can be expressed as

$$v_\tau(x, k\tau + t\tau) = v_\tau^k(((1 - t)I + t\mathcal{T}_\tau^k)^{-1}(x)). \tag{8}$$

Further, we denote the piece-wise constant concatenation of the discrete velocity fields by

$$\tilde{v}_\tau(x, k\tau + t\tau) = v_\tau^k, \quad t \in (0, 1).$$

Note that for any $\tau$ it holds that $v_\tau \in L_2(\gamma_\tau, \mathbb{R}^d \times [0, T])$ and $\tilde{v}_\tau \in L_2(\tilde{\gamma}_\tau, \mathbb{R}^d \times [0, T])$. To derive limits of these velocity fields, we recall the notion of convergence from (Ambrosio et al., 2005, Definition 5.4.3) allowing that the iterates are not defined on the same space. In this paper, we stick to square integrable measurable functions defined on finite dimensional domains, which slightly simplifies the definition.

**Definition 12.** *Let $\Omega \subseteq \mathbb{R}^d$ be a measurable domain, assume that $\mu_k \in \mathcal{P}_2(\Omega)$ converges weakly to $\mu \in \mathcal{P}_2(\Omega)$ and let $f_k \in L_2(\mu_k, \Omega)$ and $f \in L_2(\mu, \Omega)$. Then, we say that $f_k$ converges weakly to $f$, if*

$$\int_\Omega \langle \phi(x), f_k(x) \rangle \mathrm{d}\mu_k(x) \to \int_\Omega \langle \phi(x), f(x) \rangle \mathrm{d}\mu(x), \quad \text{as } k \to \infty$$

*for all test functions $\phi \in C_c^\infty(\Omega)$. We say that $f_k$ converges strongly to $f$ if*

$$\limsup_{k \to \infty} \|f_k\|_{L_2(\mu_k, \Omega)} \le \|f\|_{L_2(\mu, \Omega)}. \tag{9}$$

Note that (Ambrosio et al., 2005, Theorem 5.4.4 (iii)) implies that formula (9) is fulfilled with equality for any strongly convergent sequence $f_k$ to $f$. Moreover it is known from the literature that subsequences of the piece-wise constant velocity admit weak limits.

**Theorem 13** (Ambrosio et al., 2005, Theorem 11.1.6). *Suppose that Assumption 4 is fulfilled for $\mathcal{F}$. Then, for any $\mu^0 \in \mathrm{dom}(\mathcal{F})$ and any sequence $(\tau_l)_l \subset (0, \infty)$, there exists a subsequence (again denoted by $(\tau_l)_l$) such that*

- *The piece-wise constant curve $\tilde{\gamma}_{\tau_l}(t)$ narrowly converges to some limit curve $\hat{\gamma}(t)$ for all $t \in [0, \infty)$.*

- *The velocity field $\tilde{v}_{\tau_l} \in L_2(\tilde{\gamma}_{\tau_l}(t), \mathbb{R}^d \times [0, T])$ weakly converges to some limit $\hat{v} \in L_2(\hat{\gamma}(t), \mathbb{R}^d \times [0, T])$ according to Definition 12 for any $T > 0$.*

- *The limit $\hat{v}$ fulfills the continuity equation with respect to $\hat{\gamma}$, i.e.,*

$$\partial_t \hat{\gamma}(t) + \nabla \cdot (\hat{v}_{\tau_l}(\cdot, t) \hat{\gamma}(t)) = 0.$$

In order to show the desired result, there remain the following questions, which we answer in the rest of this section:

- Does Theorem 13 also hold for the velocity fields $v_{\tau_l}$ defined in (8) belonging to the geodesic interpolations?

- Can we show strong convergence for the whole sequence $v_{\tau_l}$?

- Does the limit $\hat{v}$ have the norm-minimizing property that $\|\hat{v}(\cdot, t)\|_{L_2(\hat{\gamma}(t))} = |\partial \mathcal{F}|(\hat{\gamma}(t))$?

To address the first of these questions, we show that weak limits of $\tilde{v}_\tau$ and $v_\tau$ coincide. The proof is a straightforward computation. A similar statement in a slightly different setting was proven in (Santambrogio, 2015, Section 8.3).

**Lemma 14.** *Suppose that Assumption 4 is fulfilled and let $(\tau_l)_l \subseteq (0, \infty)$ be a sequence with $\tau_l \to 0$ as $l \to \infty$ such that $\tilde{v}_{\tau_l} \in L_2(\tilde{\gamma}_{\tau_l}, \mathbb{R}^d \times [0, T])$ converges weakly to some $\hat{v} \in L_2(\hat{\gamma}, \mathbb{R}^d \times [0, T])$. Then, also $v_{\tau_l} \in L_2(\gamma_{\tau_l}, \mathbb{R}^d \times [0, T])$ converges weakly to $\hat{v}$.*

*Proof.* Let $\phi \in C_c^\infty(\mathbb{R}^d \times [0, T])$. In particular, $\phi$ is Lipschitz continuous with some Lipschitz constant $L < \infty$. Then, it holds that

$$\left| \int_{\mathbb{R}^d \times [0,T]} \langle \phi, v_{\tau_l} \rangle \mathrm{d}\gamma_{\tau_l} - \int_{\mathbb{R}^d \times [0,T]} \langle \phi, v \rangle \mathrm{d}\hat{\gamma} \right|$$

$$\le \left| \int_{\mathbb{R}^d \times [0,T]} \langle \phi, v_{\tau_l} \rangle \mathrm{d}\gamma_{\tau_l} - \int_{\mathbb{R}^d \times [0,T]} \langle \phi, \tilde{v}_{\tau_l} \rangle \mathrm{d}\tilde{\gamma}_{\tau_l} \right| + \left| \int_{\mathbb{R}^d \times [0,T]} \langle \phi, \tilde{v}_{\tau_l} \rangle \mathrm{d}\tilde{\gamma}_{\tau_l} - \int_{\mathbb{R}^d \times [0,T]} \langle \phi, v \rangle \mathrm{d}\hat{\gamma} \right|.$$

Since $\tilde{v}_{\tau_l}$ converges weakly to $\hat{v}$, the second term converges to zero as $l \to \infty$. Thus in order to show that also $v_{\tau_l}$ converges weakly to $\hat{v}$, it remains to show that also the first term converges to zero. By

using the notations $T_l = \min\{k\tau_l : k\tau_l \geq T, k \in \mathbb{Z}_{\geq 0}\}$ and $K_l = \lceil \frac{T}{\tau_l} \rceil = \frac{T_l}{\tau_l}$, we can estimate

$$\left| \int_{\mathbb{R}^d \times [0,T]} \langle \phi, v_{\tau_l} \rangle \mathrm{d}\gamma_{\tau_l} - \int_{\mathbb{R}^d \times [0,T]} \langle \phi, \tilde{v}_{\tau_l} \rangle \mathrm{d}\tilde{\gamma}_{\tau_l} \right|$$

$$= \left| \int_0^T \left( \int_{\mathbb{R}^d} \langle \phi(x,t), v_{\tau_l}(x,t) \rangle \mathrm{d}\gamma_{\tau_l}(t)(x) - \int_{\mathbb{R}^d} \langle \phi(x,t), \tilde{v}_{\tau_l}(x,t) \rangle \mathrm{d}\tilde{\gamma}_{\tau_l}(t)(x) \right) \mathrm{d}t \right|$$

$$\leq \int_0^{T_l} \left| \int_{\mathbb{R}^d} \langle \phi(x,t), v_{\tau_l}(x,t) \rangle \mathrm{d}\gamma_{\tau_l}(t)(x) - \int_{\mathbb{R}^d} \langle \phi(x,t), \tilde{v}_{\tau_l}(x,t) \rangle \mathrm{d}\tilde{\gamma}_{\tau_l}(t)(x) \right| \mathrm{d}t$$

$$\leq \sum_{k=0}^{K_l-1} \int_{k\tau_l}^{(k+1)\tau_l} \left| \int_{\mathbb{R}^d} \langle \phi(x,t), v_{\tau_l}(x,t) \rangle \mathrm{d}\gamma_{\tau_l}(t)(x) - \int_{\mathbb{R}^d} \langle \phi(x,t), \tilde{v}_{\tau_l}(x,t) \rangle \mathrm{d}\tilde{\gamma}_{\tau_l}(t)(x) \right| \mathrm{d}t$$

$$= \tau_l \sum_{k=0}^{K_l-1} \int_0^1 \left| \int_{\mathbb{R}^d} \langle \phi(x, k\tau_l + t\tau_l), v_{\tau_l}(x, k\tau_l + t\tau_l) \rangle \mathrm{d}\gamma_{\tau_l}(k\tau_l + t\tau_l)(x) \right.$$

$$\left. - \int_{\mathbb{R}^d} \langle \phi(x, k\tau_l + t\tau_l), \tilde{v}_{\tau_l}(x, k\tau_l + t\tau_l) \rangle \mathrm{d}\tilde{\gamma}_{\tau_l}(k\tau_l + t\tau_l)(x) \right| \mathrm{d}t.$$

By denoting with $\mathcal{T}_{\tau_l}^k$ the optimal transport map from $\mu_{\tau_l}^k$ to $\mu_{\tau_l}^{k+1}$, we have for $t \in (0,1)$ that

$$\gamma_{\tau_l}(k\tau_l + t\tau_l) = ((1-t)I + t\mathcal{T}_{\tau_l}^k)_{\#}\mu_{\tau_l}^k, \quad \tilde{\gamma}_{\tau_l}(k\tau_l + t\tau_l) = \mu_{\tau_l}^k,$$

and the velocity fields satisfy

$$v_{\tau_l}(x, k\tau_l + t\tau_l) = v_{\tau_l}^k(((1-t)I + t\mathcal{T}_{\tau_l}^k)^{-1}(x)), \quad \tilde{v}_{\tau_l}(x, k\tau_l + t\tau_l) = v_{\tau_l}^k(x).$$

Then, the above term becomes

$$\tau_l \sum_{k=0}^{K_l-1} \int_0^1 \left| \int_{\mathbb{R}^d} \langle \phi(x, k\tau_l + t\tau_l), v_{\tau_l}^k(((1-t)I + t\mathcal{T}_{\tau_l}^k)^{-1}(x)) \rangle \mathrm{d}((1-t)I + t\mathcal{T}_{\tau_l}^k)_{\#}\mu_{\tau_l}^k(x) \right.$$

$$\left. - \int_{\mathbb{R}^d} \langle \phi(x, k\tau_l + t\tau_l), v_{\tau_l}^k(x) \rangle \mathrm{d}\mu_{\tau_l}^k(x) \right| \mathrm{d}t$$

$$= \tau_l \sum_{k=0}^{K_l-1} \int_0^1 \left| \int_{\mathbb{R}^d} \langle \phi((1-t)x + t\mathcal{T}_{\tau_l}^k(x), k\tau_l + t\tau_l), v_{\tau_l}^k(x) \rangle \mathrm{d}\mu_{\tau_l}^k(x) \right.$$

$$\left. - \int_{\mathbb{R}^d} \langle \phi(x, k\tau_l + t\tau_l), v_{\tau_l}^k(x) \rangle \mathrm{d}\mu_{\tau_l}^k(x) \right| \mathrm{d}t$$

$$\leq \tau_l \sum_{k=0}^{K_l-1} \int_0^1 \int_{\mathbb{R}^d} \left| \langle \phi((1-t)x + t\mathcal{T}_{\tau_l}^k(x), k\tau_l + t\tau_l) - \phi(x, k\tau_l + t\tau_l), v_{\tau_l}^k(x) \rangle \right| \mathrm{d}\mu_{\tau_l}^k(x)\mathrm{d}t.$$

Using Hölders' inequality and we obtain that this is smaller or equal than

$$\tau_l \sum_{k=0}^{K_l-1} \int_0^1 \left( \int_{\mathbb{R}^d} \|\phi((1-t)x + t\mathcal{T}_{\tau_l}^k(x), k\tau_l + t\tau_l) - \phi(x, k\tau_l + t\tau_l)\|^2 \mathrm{d}\mu_{\tau_l}^k(x) \int_{\mathbb{R}^d} \|v_{\tau_l}^k(x)\|^2 \mathrm{d}\mu_{\tau_l}^k(x) \right)^{1/2} \mathrm{d}t$$

By the Lipschitz continuity of $\phi$ and inserting the definition of $v_{\tau_l}^k = \frac{\mathcal{T}_{\tau_l}^k - I}{\tau_l}$ this is smaller or equal than

$$\tau_l \sum_{k=0}^{K_l-1} \int_0^1 \left( t^2 \frac{L^2}{\tau_l^2} \int_{\mathbb{R}^d} \|\mathcal{T}_{\tau_l}^k(x) - x\|^2 \mathrm{d}\mu_{\tau_l}^k(x) \int_{\mathbb{R}^d} \|\mathcal{T}_{\tau_l}^k(x) - x\|^2 \mathrm{d}\mu_{\tau_l}^k(x) \right)^{1/2} \mathrm{d}t$$

$$= L \sum_{k=0}^{K_l-1} \left( \int_0^1 t\mathrm{d}t \right) \left( \int_{\mathbb{R}^d} \|\mathcal{T}_{\tau_l}^k(x) - x\|^2 \mathrm{d}\mu_{\tau_l}^k(x) \right)$$

$$= \frac{L}{2} \sum_{k=0}^{K_l-1} W_2^2(\mu_{\tau_l}^k, \mu_{\tau_l}^{k+1})$$

$$\leq \frac{L}{2} \sum_{k=0}^{\infty} W_2^2(\mu_{\tau_l}^k, \mu_{\tau_l}^{k+1})$$

Finally, we have by the definition of the minimizing movements scheme that

$$\frac{1}{2\tau_l} W_2^2(\mu_{\tau_l}^k, \mu_{\tau_l}^{k+1}) \leq \mathcal{F}(\mu_{\tau_l}^k) - \mathcal{F}(\mu_{\tau_l}^{k+1}).$$

Summing up for $k = 0, 1, \ldots$ we finally arrive at the bound

$$\left| \int_{\mathbb{R}^d \times [0,T]} \langle \phi, v_{\tau_l} \rangle \mathrm{d}\gamma_{\tau_l} - \int_{\mathbb{R}^d \times [0,T]} \langle \phi, v \rangle \mathrm{d}\hat{\gamma} \right| \leq \tau_l L \left( \mathcal{F}(\mu^0) - \inf_{\mu \in \mathcal{P}_2(\mathbb{R}^d)} \mathcal{F}(\mu) \right).$$

Since $\mathcal{F}$ is bounded from below the upper bound converges to zero as $l \to \infty$. This concludes the proof. $\qquad \square$

Finally, we employ the previous results to show Theorem 6 from the main part of the paper. That is, we show that for any $(\tau_l)_l \subseteq (0, \infty)$ with $\tau_l \to 0$ the whole $v_{\tau_l}$ converges strongly to $v$.

**Theorem 15** (Theorem 6). *Suppose that Assumption 4 is fulfilled and let $(\tau_l)_l \subseteq (0, \infty)$ with $\tau_l \to 0$. Then, $(v_{\tau_l})_l$ converges strongly to the velocity field $\hat{v} \in L_2(\gamma, \mathbb{R}^d \times [0,T])$ of the Wasserstein gradient flow $\gamma \colon (0, \infty) \to \mathcal{P}_2(\mathbb{R}^d)$ of $\mathcal{F}$ starting in $\mu^0$.*

*Proof.* We show that any subsequence of $\tau_l$ admits a subsequence converging strongly to $\hat{v}$. Using the sub-subsequence criterion this yields the claim.

By Theorem 13 and Lemma 14 we know that any subsequence of $\tau_l$ admits a weakly convergent subsequence. In an abuse of notations, we denote it again by $\tau_l$ and its limit by $\tilde{v}$. Then, we prove that the convergence is indeed strong and that $\tilde{v} = \hat{v}$.

**Step 1: Bounding** $\limsup_{l \to \infty} \int_0^T \|v_{\tau_l}(\cdot, t)\|_{L_2(\gamma_{\tau_l}(t), \mathbb{R}^d)}^2 \mathrm{d}t$ **from above.** Since $\lambda$-convexity with $\lambda \geq 0$ implies $\lambda$-convexity with $\lambda = -1$, we can assume without loss of generality that $\lambda < 0$. Then, by (Ambrosio et al., 2005, Lemma 9.2.7, Theorem 4.0.9), we know that for any $\tau > 0$ it holds

$$W_2(\tilde{\gamma}_\tau(t), \gamma(t)) \leq \tau C(\tau, t), \qquad C(\tau, t) = \frac{(1 + 2|\lambda| t_\tau)|\partial \mathcal{F}|(\mu_0)}{\sqrt{2}(1 + \lambda\tau)} \exp\left( -\frac{\log(1 + \lambda\tau)}{\tau} t \right),$$

where $t_\tau = \min\{k\tau : k\tau \geq t, k \in \mathbb{N}_0\}$. For simplicity, we use the notations $\tau_{\max} = \max\{\tau_l : l \in \mathbb{N}_0\}$, $K_l = \lceil \frac{T}{\tau_l} \rceil$ and $T_{\max} = T + \tau_{\max}$. Since $\lambda < 0$, we have that $C(\tau, t) \leq C(\tau, T_{\max}) =: C(\tau)$ for all $t \in [0, T_{\max}]$. Moreover, we have that $C(\tau) \to \frac{(1 + 2|\lambda| T_{\max}|\partial \mathcal{F}|(\mu_0)}{\sqrt{2}} \exp(-\lambda T_{\max})$ as $\tau \to 0$, such that the sequence $(C(\tau_l))_l$ is bounded. In particular, there exists a $C > 0$ such that

$$W_2(\tilde{\gamma}_{\tau_l}(t), \gamma(t)) \leq \tau_l C, \quad \text{for all} \quad t \in [0, T_{\max}].$$

Inserting $t = k\tau_l$ for $k = 0, \ldots, K_l$ gives

$$W_2(\mu_{\tau_l}^k, \gamma(k\tau_l)) \leq \tau_l C, \quad \text{for all} \quad k = 0, \ldots, K_l. \tag{10}$$

Now, we can conclude by the theorem of Benamou-Brenier that

$$\int_0^T \|v_{\tau_l}(\cdot, t)\|_{L_2(\gamma_{\tau_l}(t), \mathbb{R}^d)}^2 \mathrm{d}t \leq \int_0^{K_l \tau_l} \|v_{\tau_l}(\cdot, t)\|_{L_2(\gamma_{\tau_l}(t), \mathbb{R}^d)}^2 \mathrm{d}t$$

$$= \sum_{k=0}^{K_l - 1} \int_{k\tau_l}^{(k+1)\tau_l} \|v_{\tau_l}(\cdot, t)\|_{L_2(\gamma_{\tau_l}(t), \mathbb{R}^d)}^2 \mathrm{d}t$$

$$= \sum_{k=1}^{K_l - 1} W_2^2(\mu_{\tau_l}^k, \mu_{\tau_l}^{k+1}).$$

Now applying the triangular inequality for any $k = 1, \ldots, K_l - 1$

$$W_2^2(\mu_{\tau_l}^k, \mu_{\tau_l}^{k+1}) \leq \left( W_2(\mu_{\tau_l}^k, \gamma(\tau_l k)) + W_2(\gamma(\tau_l k), \gamma(\tau_l(k+1)) + W_2(\gamma(\tau_l(k+1)), \mu_{\tau_l}^{k+1}) \right)^2$$

and the estimate from (10) yields

$$\int_0^T \|v_{\tau_l}(\cdot, t)\|_{L_2(\gamma_{\tau_l}(t), \mathbb{R}^d)}^2 \mathrm{d}t \leq \sum_{k=0}^{K_l - 1} \left( 2\tau C + W_2(\gamma(\tau_l k), \gamma(\tau_l(k+1))) \right)^2$$

By Jensens' inequality the right hand side can be bounded from above by

$$4K_l\tau^2 C^2 + 2\tau K_l C \left( \sum_{k=0}^{K_l-1} \frac{1}{K_l} W_2^2(\gamma(\tau_l k), \gamma(\tau_l(k+1))) \right)^{1/2} + \sum_{k=0}^{K_l-1} W_2^2(\gamma(\tau_l k), \gamma(\tau_l(k+1)))$$

Finally, again Benamou-Brenier gives that $W_2^2(\gamma(\tau_l k), \gamma(\tau_l(k+1))) \leq \int_{k\tau_l}^{(k+1)\tau_l} \|\hat{v}(\cdot, t)\|_{L_2(\gamma(t))}^2 dt$ such that the above formula is smaller or equal than

$$4K_l\tau_l^2 C^2 + 2\tau_l \sqrt{K_l} C \left( \int_0^{K_l\tau_l} \|\hat{v}(\cdot, t)\|_{L_2(\gamma(t))}^2 dt \right)^{1/2} + \int_0^{K_l\tau_l} \|\hat{v}(\cdot, t)\|_{L_2(\gamma(t))}^2 dt.$$

Since $K_l\tau_l \to T$ and $\tau_l \to 0$ as $l \to \infty$ this converges to $\int_0^T \|\hat{v}(\cdot, t)\|_{L_2(\gamma(t))}^2 dt$ such that we can conclude

$$\limsup_{l\to\infty} \int_0^T \|v_{\tau_l}(\cdot, t)\|_{L_2(\gamma_{\tau_l}(t), \mathbb{R}^d)}^2 dt \leq \int_0^T \|\hat{v}(\cdot, t)\|_{L_2(\gamma(t))}^2 dt.$$

**Step 2: Strong convergence.** By (Ambrosio et al., 2005, Theorem 8.3.1, Proposition 8.4.5) we know that for any $v$ fulfilling the continuity equation

$$\partial_t \gamma(t) + \nabla \cdot (v(\cdot, t)\gamma(t)) = 0,$$

it holds that $\|v(\cdot, t)\|_{L_2(\gamma(t), \mathbb{R}^d} \geq \|\hat{v}(\cdot, t)\|_{L_2(\gamma(t), \mathbb{R}^d}$. Since $\tilde{v}$ fulfills the continuity equation by Theorem 13, this implies that

$$\int_0^T \|\tilde{v}(\cdot, t)\|_{L_2(\gamma(t), \mathbb{R}^d}^2 dt \geq \int_0^T \|\hat{v}(\cdot, t)\|_{L_2(\gamma(t), \mathbb{R}^d}^2 dt = \limsup_{l\to\infty} \int_0^T \|v_{\tau_l}(\cdot, t)\|_{L_2(\gamma_{\tau_l}(t), \mathbb{R}^d)}^2 dt.$$

In particular, $v_{\tau_l} \in L_2(\gamma_{\tau_l}, \mathbb{R}^d \times [0, T])$ converges strongly to $\tilde{v}$ such that by (Ambrosio et al., 2005, Theorem 5.4.4 (iii)) it holds equality in the above equation, i.e.,

$$\int_0^T \|\tilde{v}(\cdot, t)\|_{L_2(\gamma(t), \mathbb{R}^d}^2 dt = \int_0^T \|\hat{v}(\cdot, t)\|_{L_2(\gamma(t), \mathbb{R}^d}^2 dt$$

Using again (Ambrosio et al., 2005, Theorem 8.3.1, Proposition 8.4.5) this implies that $\tilde{v} = \hat{v}$. $\quad\square$

## C   PROOFS FROM SECTION 4

### C.1   PROOF OF PROPOSITION 8

(i)  By the law of total probability, we have

$$\tilde{p}(x) = P(\tilde{X} = x) = P(\tilde{X} = x \text{ and } X \text{ was accepted}) + P(\tilde{X} = x \text{ and } X \text{ was rejected}).$$

Since it holds $\tilde{X} = X$ if $X$ and $\tilde{X} = X'$ if $X$ is rejected this can be reformulated as

$$P(X = x \text{ and } X \text{ was accepted}) + P(X' = x \text{ and } X \text{ was rejected})$$
$$= P(X = x)P(X \text{ was accepted}|X = x) + P(X' = x)P(X \text{ was rejected}), \quad (11)$$

where we used the definition of conditional probabilities and the fact that $X$ and $X'$ are independent. Because $X, X' \sim \mu$, we now have $P(X = x) = P(X' = x) = p(x)$ and, by definition, that $P(X \text{ was accepted}|X = x) = \alpha(x)$. Finally, it holds that

$$P(X \text{ was rejected}) = \int_{\mathbb{R}^d} P(X \text{ was rejected}|X = x)p(x)dx$$

$$= \int_{\mathbb{R}^d} (1 - \alpha(x))p(x)dx = 1 - \mathbb{E}[\alpha(X)].$$

Inserting these terms in (11) yields the claim.

(ii) Note, since $X \sim p$, we have

$$\int_{\mathbb{R}^d} \frac{p(x)\alpha(x)}{\mathbb{E}[\alpha(X)]} \mathrm{d}x = \int_{\mathbb{R}^d} \frac{p(x)\alpha(x)}{\int_{\mathbb{R}^d} p(y)\alpha(y)\mathrm{d}y} \mathrm{d}x = 1,$$

in particular $\frac{p(x)\alpha(x)}{\mathbb{E}[\alpha(X)]}$ defines a density. Now let $\eta \in \mathcal{P}_2(\mathbb{R}^d)$ be the corresponding probability measure. Then, it holds by part (i) that $\tilde{\mu} = \mathbb{E}[\alpha(X)]\eta + (1 - \mathbb{E}[\alpha(X)])\mu$. We will show that $\mathrm{KL}(\eta, \nu) \leq \mathrm{KL}(\mu, \nu)$. Due to the convexity of the KL divergence in the linear space of measures this implies the claim.

We denote $Z = \mathbb{E}[\alpha(X)] = \int_{\mathbb{R}^d} \min(p(x), \frac{q(x)}{\tilde{c}})\mathrm{d}x$. Then it holds

$$\mathrm{KL}(\eta, \nu) = \int_{\mathbb{R}^d} \frac{p(x)\alpha(x)}{Z} \log\left(\frac{p(x)\alpha(x)}{Zq(x)}\right) \mathrm{d}x$$

$$= \int_{\mathbb{R}^d} \frac{\min(p(x), \frac{q(x)}{\tilde{c}})}{Z} \log\left(\frac{\tilde{c}p(x)\alpha(x)}{q(x)}\right) \mathrm{d}x - \log(Z\tilde{c}) \int_{\mathbb{R}^d} \frac{\min(p(x), \frac{q(x)}{\tilde{c}})}{Z} \mathrm{d}x$$

$$= \int_{\mathbb{R}^d} \frac{\min(p(x), \frac{q(x)}{\tilde{c}})}{Z} \log\left(\min\left(\frac{\tilde{c}p(x)}{q(x)}, 1\right)\right) \mathrm{d}x - \log(Z) - \log(\tilde{c})$$

$$= \int_{\mathbb{R}^d} \min\left(\frac{\min(p(x), \frac{q(x)}{\tilde{c}})}{Z} \log\left(\frac{\tilde{c}p(x)}{q(x)}\right), 0\right) \mathrm{d}x - \log(Z) - \log(\tilde{c}).$$

Since $\log\left(\frac{\tilde{c}p(x)}{q(x)}\right) \leq 0$ if and only if $p(x) \leq q(x)/\tilde{c}$ this can be reformulated as

$$\int_{\mathbb{R}^d} \min\left(\frac{p(x)}{Z} \log\left(\frac{\tilde{c}p(x)}{q(x)}\right), 0\right) \mathrm{d}x - \log(Z) - \log(\tilde{c})$$

$$\leq \int_{\mathbb{R}^d} \min\left(p(x) \log\left(\frac{\tilde{c}p(x)}{q(x)}\right), 0\right) \mathrm{d}x - \log(Z) - \log(\tilde{c}), \tag{12}$$

where the inequality comes from the fact that $Z = \mathbb{E}[\alpha(X)] \in [0, 1]$. Moreover, it holds by Jensen's inequality that

$$-\log(Z) = -\log(\mathbb{E}[\alpha(X)]) = -\log\left(\int_{\mathbb{R}^d} p(x) \min\left(\frac{q(x)}{\tilde{c}p(x)}, 1\right) \mathrm{d}x\right)$$

$$\leq -\int_{\mathbb{R}^d} p(x) \log\left(\min\left(\frac{q(x)}{\tilde{c}p(x)}, 1\right)\right) \mathrm{d}x$$

$$= -\int_{\mathbb{R}^d} p(x) \min\left(\log\left(\frac{q(x)}{\tilde{c}p(x)}\right), 0\right) \mathrm{d}x$$

$$= \int_{\mathbb{R}^d} p(x) \max\left(\log\left(\frac{\tilde{c}p(x)}{q(x)}\right), 0\right) \mathrm{d}x$$

$$= \int_{\mathbb{R}^d} \max\left(p(x) \log\left(\frac{\tilde{c}p(x)}{q(x)}\right), 0\right) \mathrm{d}x.$$

Thus, we obtain that (12) can be bounded from above by

$$\int_{\mathbb{R}^d} \min\left(p(x) \log\left(\frac{\tilde{c}p(x)}{q(x)}\right), 0\right) \mathrm{d}x + \int_{\mathbb{R}^d} \max\left(p(x) \log\left(\frac{\tilde{c}p(x)}{q(x)}\right), 0\right) \mathrm{d}x - \log(\tilde{c})$$

which equals

$$\int_{\mathbb{R}^d} p(x) \log\left(\frac{\tilde{c}p(x)}{q(x)}\right) \mathrm{d}x - \log(\tilde{c}) = \int_{\mathbb{R}^d} p(x) \log\left(\frac{p(x)}{q(x)}\right) \mathrm{d}x = \mathrm{KL}(\mu, \nu).$$

In summary, we have $\mathrm{KL}(\eta, \nu) \leq \mathrm{KL}(\mu, \nu)$, which implies the assertion.

$\square$

## C.2 Proof of Corollary 9

Let $Y = \left| \mathbb{E}[\alpha(X)] - \frac{1}{N} \sum_{i=1}^{N} \alpha(X_N) \right|$ denote the random variable representing the error. Since $\alpha(X_N) \in [0, 1]$, Hoeffding's inequality Hoeffding (1963) yields that $P(Y > t) \le 2 \exp(-\frac{Nt^2}{2})$. Consequently, we have that

$$\mathbb{E}[Y] = \int_0^\infty P(Y > t) \mathrm{d}t \le 2 \int_0^\infty \exp\left(-\frac{Nt^2}{2}\right) \mathrm{d}t = \frac{\sqrt{2\pi}}{\sqrt{N}}.$$

□

# D  Algorithms

## D.1  Training and Evaluation Algorithms

We summarize the training and evaluation procedures for the neural JKO steps in Algorithm 2 and 3 and for the importance-based rejection steps in Algorithm 4 and 5.

## D.2  Density Evaluation of Importance Corrected Neural JKO Models

Let $(X_0, ..., X_K)$ be an importance corrected neural JKO model. We aim to evaluate the density $p^K$ of $X_K$ at some given point $x \in \mathbb{R}^d$. Using Proposition 8, this can be done recursively by Algorithm 6. Note that the algorithm always terminates since $K$ is reduced by one in each call.

# E  Additional Numerical Results and Implementation Details

## E.1  Test Distributions

We evaluate our method on the following test distributions.

- **Mustache:** The two-dimensional log-density is given as $\log \mathcal{N}(0, \Sigma) \circ T$ with $\Sigma = [1, \sigma; \sigma, 1]$ and $T(x_1, x_2) = (x_1, (x_2 - (x_1^2 - 1)^2))$. Note that $\det(\nabla T(x)) = 1$ for all $x$. In particular, we obtain directly by the transformation formula that the normalization constant is one. Depending on $\sigma \in [0, 1)$ close to 1, this probability distribution has very long and narrow tails making it hard for classical MCMC methods to sample them. In our experiments we use $\sigma = 0.9$.

- **Shifted 8 Modes:** A two-dimensional Gaussian mixture model with 8 equal weighted modes and covariance matrix $1 \times 10^{-2} I$. The modes are placed in a circle with radius 1 and center $(-1, 0)$. Due to the shifted center classical MCMC methods have difficulties to distribute the mass correctly onto the modes.

- **Shifted 8 Peaky:** This is the same distribution as the shifted 8 Modes with the difference that we reduce the width of the modes to the covariance matrix $5 \times 10^{-3} I$. Since the modes are disconnected, it becomes harder to sample from them.

- **Funnel:** We consider the (normalized) probability density function given by $q(x) = \mathcal{N}(x_1 | 0, \sigma_f^2) \mathcal{N}(x_{2:10} | 0, \exp(x_1) I)$, where $\sigma_f^2 = 9$. This example was introduced by Neal (2003). Similarly to the mustache example, this distribution has a narrow funnel for small values $x_1$ which can be hard to sample.

- **GMM-$d$:** A $d$-dimensional Gaussian mixture model with 10 equal weighted modes with covariance matrix $1 \times 10^{-2} I$ and means drawn randomly from a uniform distribution on $[-1, 1]^d$. This leads to a peaky high-dimensional and multimodal probability distribution which is hard to sample from.

- **LGCP:** This is a high dimensional standard example taken from Arbel et al. (2021); Matthews et al. (2022); Vargas et al. (2023a). It describes a Log-Gaussian Cox process on a $40 \times 40$ grid as arising from spatial statistics Møller et al. (1998). This leads to a 1600-dimensional probability distribution with the unnormalized density function $q(x) \propto \mathcal{N}(x | \mu, K) \prod_{i \in \{1, ..., 40\}^2} \exp(x_i y_i - a \exp(x_i))$, where $\mu$ and $K$ are a fixed mean and

---

**Algorithm 2** Training of neural JKO steps

---

**Input:** Samples $x_1^k, ..., x_N^k$ of $\mu^k$.
   Minimize the loss function $\theta \mapsto \mathcal{L}(\theta)$ from (5) using the Adam optimizer.
**Output:** Parameters $\theta$.

---

---

**Algorithm 3** Sampling and density propagation for neural JKO steps

---

**Input:** $\begin{cases} \text{- Samples } x_1^k, ..., x_N^k \text{ of } \mu^k, \\ \text{- Density values } p^k(x_1^k), ..., p^k(x_N^k). \end{cases}$

  **for** $i = 1, ..., N$ **do**
     1. Solve the ODE (6) for $x = x_i^k$.
     2. Set $x_i^{k+1} = z_\theta(x_i^k, \tau)$.
     3. Set $p^{k+1}(x_i^{k+1}) = \frac{p^k(x_i^k)}{\exp(\ell_\theta(x_i^k, \tau))}$.
  **end for**

**Output:** $\begin{cases} \text{- Samples } x_1^k, ..., x_N^k \text{ of } \mu^{k+1}, \\ \text{- Density values } p^{k+1}(x_1^{k+1}), ..., p^{k+1}(x_N^{k+1}). \end{cases}$

---

---

**Algorithm 4** Parameter selection for important-based rejection steps

---

**Input:** $\begin{cases} \text{- Samples } x_1^k, ..., x_N^k \text{ of } \mu^k \text{ with corresponding densities} \\ \text{- desired rejection rate } r, \text{ unnormalized target density } g \end{cases}$

  Choose $c$ by bisection search such that

$$1 - r = \frac{1}{N} \sum_{i=1}^{N} \alpha_k(x_i^k), \qquad \alpha_k(x) = \min\left\{1, \frac{g(x)}{cp^k(x)}\right\}.$$

**Output:** Rejection parameter $c$ and estimate of $\mathbb{E}[\alpha(X_\tau^k)] \approx 1 - r$.

---

---

**Algorithm 5** Sampling and density propagation for important-based rejection steps

---

**Input:** - Samples $x_1^k, ..., x_N^k$ of $\mu^k$

**Assume:** $\begin{cases} \text{- we are able to generate samples from } \mu^k \text{ with corresponding density } p^k \\ \text{- we can evaluate the unnormalized target density } g \end{cases}$

  **for** $i = 1, ..., N$ **do**
     1. Compute acceptance probability $\alpha_k(x_i^k) = \min\left\{1, \frac{g(x_i^k)}{cp^k(x_i^k)}\right\}$.
     2. Draw $u$ uniformly from $[0, 1]$ and $x'$ from $\mu_\tau^k$.
     3. Set $x_i^{k+1} = \begin{cases} x_i^k, & \text{if } u \leq \alpha(x_i^k), \\ x' & \text{if } u > \alpha(x_i^k). \end{cases}$
     4. Compute $\alpha_k(x_i^{k+1}) = \min\left\{1, \frac{g(x_i^{k+1})}{cp^k(x_i^{k+1})}\right\}$.
     5. Set $p^{k+1}(x_i^{k+1}) = p^k(x_i^{k+1})(\alpha_k(x_i^{k+1}) + 1 - \mathbb{E}[\alpha_k(X_k)])$.
  **end for**

**Output:** $\begin{cases} \text{- Samples } x_1^{k+1}, ..., x_N^{k+1} \text{ of } \mu_\tau^{k+1}. \\ \text{- Density values } p^{k+1}(x_1^{k+1}), ..., p_\tau^{k+1}(x_N^{k+1}). \end{cases}$

---

---

**Algorithm 6** Density Evaluation of Importance Corrected Neural JKO Models

---

**Input:** $x \in \mathbb{R}^d$, model $(X_0, ..., X_K)$
    **if** $K = 0$ **then**
        **Return** latent density $p^0(x)$.
    **else if** the last step is a rejection step **then**
        1. Evaluate $p^{K-1}(x)$ by applying this algorithm for $(X_0, ..., X_{K-1})$.
        2. **Return** $p^K(x) = p^{K-1}(\alpha_k(x) + 1 - \mathbb{E}[\alpha_k(X_{k-1})])$
    **else**                                             ▷ *last step is a neural JKO step*
        1. Solve the ODE system (with $v_\theta$ from the neural JKO step)

$$\left( \begin{array}{c} \dot{z}_\theta(x, t) \\ \dot{\ell}_\theta(x, t) \end{array} \right) = \left( \begin{array}{c} v_\theta(x, t) \\ \mathrm{trace}(\nabla v_\theta(z_\theta(x, t), t)) \end{array} \right), \qquad \left( \begin{array}{c} z_\theta(x, \tau) \\ \ell_\theta(x, \tau) \end{array} \right) = \left( \begin{array}{c} x \\ 0 \end{array} \right).$$

        2. Set $\tilde{x} = z_\theta(x, 0)$.
        3. Evaluate $p^{K-1}(\tilde{x})$ by applying this algorithm for $(X_0, ..., X_{K-1})$.
        4. **Return** $p^K(x) = p^{K-1}(\tilde{x}) \exp(\ell_\theta(x, 0))$.
    **end if**
**Output:** Density $p^K(x)$

---

covariance kernel, $y_i$ is some data and $a$ is a hyperparameter. For details on this example and the specific parameter choice we refer to Matthews et al. (2022).

### E.2 ERROR MEASURES

We use the following error measures.

- The **energy distance** was proposed by Székely (2002) and is defined by

$$D(\mu, \nu) = -\frac{1}{2} \int_{\mathbb{R}^d} \int_{\mathbb{R}^d} \|x - y\| \mathrm{d}(\mu - \nu)(x) \mathrm{d}(\mu - \nu)(y).$$

It is the maximum mean discrepancy with the negative distance kernel $K(x, y) = -\|x - y\|$ (see Sejdinovic et al., 2013) and can be estimated from below and above by the Wasserstein-1 distance (see Hertrich et al., 2024). It is a metric on the space of probability measures. Consequently, a smaller energy distance indicates a higher similarity of the input distributions. By discretizing the integrals it can be easily evaluated based on $N \in \mathbb{N}$ samples $\boldsymbol{x} = (x_i)_i \sim \mu^{\otimes N}, \boldsymbol{y} = (y_i)_i \sim \nu^{\otimes N}$ as

$$D(\boldsymbol{x}, \boldsymbol{y}) = \sum_{i,j=1}^{N} \|x_i - y_j\| - \frac{1}{2} \sum_{i,j=1}^{N} \|x_i - x_j\| - \frac{1}{2} \sum_{i,j=1}^{N} \|y_i - y_j\|.$$

We use $N = 50000$ samples in Table 1.

- We also evaluate the squared Wasserstein-2 distance which is defined in Section 2. To this end, we use the Python Optimal Transport package (POT, Flamary et al., 2021). Note that computing the Wasserstein distance has complexity $O(n^3)$ where $n$ is the number of points. Hence, we evaluate the Wasserstein distance based on less samples compared to the case of other metrics. We use $N = 5000$ samples in Table 3. In addition, we want to highlight that the expected Wasserstein distance evaluated on empirical measures instead of its continuous counterpart suffers from the curse of dimensionality. In particular, its sample complexity scales as $O(n^{-1/d})$ (Peyré & Cuturi, 2019, Chapter 8.4.1). Consequently, the sample-based Wasserstein distance in high dimensions can differ significantly from the true Wasserstein distance of the continuous distributions. Indeed, we can see in Table 3 that the sampling error has often the same order of magnitude as the reported errors. Overall we can draw similar conclusions from this evaluation as for the energy distance in Table 1.

- We **estimate the log normalizing constant** (short $\log(Z)$ estimation) which is used as a benchmark standard in various references (e.g., in Arbel et al., 2021; Matthews et al., 2022;

Table 3: We report the expected squared empirical Wasserstein-2 distance ($W_2^2$) and its standard deviation between generated and ground truth samples of size $N = 5000$ for the different methods and for all examples where the ground truth model is known. A smaller value of $W_2^2$ indicates a better result. Note that the curse of dimensionality present in the sample complexity, might limit the reliability of the results for the high-dimensional examples. In particular for the funnel distribution, we observe that the expected empirical Wasserstein-2 distance between two independent sets of ground truth samples is higher than the observed $W_2^2$ values.

| Distribution | Sampler | | | | | | Sampling Error |
|---|---|---|---|---|---|---|---|
| | MALA | HMC | DDS | CRAFT | Neural JKO | Neural JKO IC (**ours**) | |
| Mustache | $4.7 \times 10^{+1} \pm 1.3 \times 10^{+1}$ | $2.8 \times 10^{+1} \pm 4.2 \times 10^{0}$ | $5.2 \times 10^{+1} \pm 2.0 \times 10^{0}$ | $5.4 \times 10^{+1} \pm 1.2 \times 10^{+1}$ | $3.0 \times 10^{+1} \pm 1.3 \times 10^{+1}$ | $1.7 \times 10^{+1} \pm 6.0 \times 10^{0}$ | $1.2 \times 10^{+1}$ |
| shifted 8 Modes | $5.5 \times 10^{-2} \pm 6.9 \times 10^{-3}$ | $4.7 \times 10^{-3} \pm 3.6 \times 10^{-3}$ | $8.7 \times 10^{-2} \pm 3.1 \times 10^{-2}$ | $2.4 \times 10^{-1} \pm 2.4 \times 10^{-3}$ | $5.6 \times 10^{-1} \pm 1.6 \times 10^{-2}$ | $6.5 \times 10^{-3} \pm 2.1 \times 10^{-3}$ | $7.4 \times 10^{-3}$ |
| shifted 8 Peaky | $5.8 \times 10^{-2} \pm 2.4 \times 10^{-2}$ | $5.6 \times 10^{-1} \pm 1.4 \times 10^{-2}$ | $1.0 \times 10^{-1} \pm 2.5 \times 10^{-2}$ | $2.5 \times 10^{-1} \pm 1.1 \times 10^{-1}$ | $5.9 \times 10^{-1} \pm 1.7 \times 10^{-2}$ | $7.2 \times 10^{-3} \pm 1.3 \times 10^{-3}$ | $5.6 \times 10^{-3}$ |
| Funnel | $5.5 \times 10^{+2} \pm 1.2 \times 10^{+2}$ | $7.4 \times 10^{+2} \pm 5.6 \times 10^{+2}$ | $5.3 \times 10^{+2} \pm 7.5 \times 10^{+1}$ | $7.9 \times 10^{+2} \pm 4.4 \times 10^{+2}$ | $9.4 \times 10^{+2} \pm 9.3 \times 10^{+2}$ | $8.5 \times 10^{+2} \pm 3.9 \times 10^{+2}$ | $1.0 \times 10^{+3}$ |
| GMM-10 | $3.8 \times 10^{0} \pm 4.2 \times 10^{-1}$ | $3.8 \times 10^{0} \pm 3.9 \times 10^{-1}$ | $3.8 \times 10^{-1} \pm 1.1 \times 10^{-1}$ | $2.4 \times 10^{0} \pm 1.0 \times 10^{0}$ | $6.3 \times 10^{-1} \pm 1.4 \times 10^{-1}$ | $1.4 \times 10^{-1} \pm 2.3 \times 10^{-2}$ | $1.4 \times 10^{-1}$ |
| GMM-20 | $8.9 \times 10^{0} \pm 4.0 \times 10^{-1}$ | $9.0 \times 10^{0} \pm 3.9 \times 10^{-1}$ | $8.8 \times 10^{-1} \pm 1.0 \times 10^{-1}$ | $7.9 \times 10^{0} \pm 1.5 \times 10^{0}$ | $1.2 \times 10^{0} \pm 2.0 \times 10^{-1}$ | $3.7 \times 10^{-1} \pm 5.0 \times 10^{-2}$ | $3.7 \times 10^{-1}$ |
| GMM-50 | $2.7 \times 10^{+1} \pm 1.0 \times 10^{0}$ | $2.7 \times 10^{+1} \pm 9.8 \times 10^{-1}$ | $3.6 \times 10^{0} \pm 8.9 \times 10^{-1}$ | $2.6 \times 10^{+1} \pm 2.6 \times 10^{0}$ | $4.4 \times 10^{0} \pm 7.9 \times 10^{-1}$ | $1.2 \times 10^{0} \pm 1.3 \times 10^{-1}$ | $1.2 \times 10^{0}$ |
| GMM-100 | $5.7 \times 10^{+1} \pm 1.2 \times 10^{0}$ | $5.7 \times 10^{+1} \pm 1.3 \times 10^{0}$ | $9.9 \times 10^{0} \pm 2.9 \times 10^{0}$ | $5.6 \times 10^{+1} \pm 1.1 \times 10^{0}$ | $1.1 \times 10^{+1} \pm 2.7 \times 10^{0}$ | $2.9 \times 10^{0} \pm 4.7 \times 10^{-1}$ | $2.8 \times 10^{0}$ |
| GMM-200 | $1.2 \times 10^{+2} \pm 2.8 \times 10^{0}$ | $1.2 \times 10^{+2} \pm 2.8 \times 10^{0}$ | $2.4 \times 10^{+1} \pm 4.0 \times 10^{+1}$ | $1.1 \times 10^{+2} \pm 3.0 \times 10^{0}$ | $2.4 \times 10^{+1} \pm 3.9 \times 10^{0}$ | $7.8 \times 10^{0} \pm 6.9 \times 10^{-1}$ | $5.9 \times 10^{0}$ |

Table 4: We report the mode MSEs for the different methods for all examples which can be represented as mixture model. A smaller mode MSE indicates a better result.

| Distribution | Sampler | | | | | |
|---|---|---|---|---|---|---|
| | MALA | HMC | DDS | CRAFT | Neural JKO | Neural JKO IC (**ours**) |
| shifted 8 Modes | $3.2 \times 10^{-3} \pm 1.3 \times 10^{-4}$ | $1.9 \times 10^{-5} \pm 1.1 \times 10^{-5}$ | $8.8 \times 10^{-3} \pm 2.2 \times 10^{-3}$ | $3.2 \times 10^{-2} \pm 6.1 \times 10^{-3}$ | $8.3 \times 10^{-2} \pm 1.0 \times 10^{-3}$ | $1.3 \times 10^{-5} \pm 4.4 \times 10^{-6}$ |
| shifted 8 Peaky | $8.3 \times 10^{-2} \pm 3.7 \times 10^{-4}$ | $7.8 \times 10^{-2} \pm 9.9 \times 10^{-4}$ | $7.9 \times 10^{-3} \pm 2.2 \times 10^{-3}$ | $3.3 \times 10^{-2} \pm 1.3 \times 10^{-2}$ | $8.4 \times 10^{-2} \pm 6.7 \times 10^{-4}$ | $1.5 \times 10^{-5} \pm 3.2 \times 10^{-6}$ |
| GMM-10 | $1.2 \times 10^{-2} \pm 5.5 \times 10^{-3}$ | $1.0 \times 10^{-2} \pm 5.8 \times 10^{-3}$ | $3.6 \times 10^{-3} \pm 1.9 \times 10^{-3}$ | $1.4 \times 10^{-1} \pm 4.7 \times 10^{-2}$ | $1.2 \times 10^{-2} \pm 5.6 \times 10^{-3}$ | $2.1 \times 10^{-5} \pm 3.0 \times 10^{-6}$ |
| GMM-20 | $6.6 \times 10^{-3} \pm 2.2 \times 10^{-3}$ | $6.6 \times 10^{-3} \pm 2.4 \times 10^{-3}$ | $3.6 \times 10^{-3} \pm 1.5 \times 10^{-3}$ | $3.8 \times 10^{-1} \pm 3.9 \times 10^{-2}$ | $7.3 \times 10^{-3} \pm 2.8 \times 10^{-3}$ | $2.4 \times 10^{-5} \pm 9.8 \times 10^{-6}$ |
| GMM-50 | $1.0 \times 10^{-2} \pm 3.8 \times 10^{-3}$ | $9.8 \times 10^{-3} \pm 3.6 \times 10^{-3}$ | $9.6 \times 10^{-3} \pm 4.9 \times 10^{-3}$ | $9.0 \times 10^{-1} \pm 6.4 \times 10^{-5}$ | $1.2 \times 10^{-2} \pm 4.4 \times 10^{-3}$ | $2.6 \times 10^{-5} \pm 6.0 \times 10^{-6}$ |
| GMM-100 | $1.1 \times 10^{-2} \pm 5.4 \times 10^{-3}$ | $1.1 \times 10^{-2} \pm 5.4 \times 10^{-3}$ | $1.1 \times 10^{-2} \pm 5.8 \times 10^{-3}$ | $9.0 \times 10^{-1} \pm 4.8 \times 10^{-8}$ | $1.4 \times 10^{-2} \pm 7.0 \times 10^{-3}$ | $1.5 \times 10^{-4} \pm 8.5 \times 10^{-5}$ |
| GMM-200 | $1.4 \times 10^{-2} \pm 5.1 \times 10^{-3}$ | $1.4 \times 10^{-2} \pm 4.6 \times 10^{-3}$ | $2.2 \times 10^{-2} \pm 8.6 \times 10^{-3}$ | $9.0 \times 10^{-1} \pm 5.8 \times 10^{-8}$ | $1.9 \times 10^{-2} \pm 6.3 \times 10^{-3}$ | $6.8 \times 10^{-4} \pm 4.0 \times 10^{-4}$ |

Phillips et al., 2024; Vargas et al., 2023a). More precisely, for the generated distribution $\mu$ with normalized density $p$ and target measure $\nu$ with density $q(x) = g(x)/Z_g$ we evaluate the term

$$\mathbb{E}_{x \sim \mu} \left[ \log \left( \frac{g(x)}{p(x)} \right) \right] = \log(Z_g) - \mathbb{E}_{x \sim \mu} \left[ \log \left( \frac{p(x)}{q(x)} \right) \right] = \log(Z_g) - \mathrm{KL}(\mu, \nu).$$

Due to the properties of the KL divergence, a higher $\log(Z)$ estimate implies a lower KL divergence between $\mu$ and $\nu$ and therefore a higher similarity of generated and target distribution. In our experiments we compute the $\log(Z)$ estimate based on $N = 50000$ samples. The results are given in Table 2.

- To quantify how well the mass is distributed on different modes for the mixture model examples (shifted 8 Modes, shifted 8 Peaky, GMM-$d$), we compute the mode weights. That is, we generate $N = 50000$ samples and assign each generated samples to the closest mode of the GMM. Afterwards, we compute for each mode the fraction of samples which is assigned to each mode. To evaluate this distribution quantitatively, we compute the mean square error (MSE) between the mode weights of the generated samples and the ground truth weights from the GMM. We call this error metric the **mode MSE**, give the results are in Table 4.

**Remark 16** (Bias in $\log(Z)$ Computation). *In the cases, where the importance corrected neural JKO sampling fits the target distribution almost perfectly, we sometimes report in Table 2 $\log(Z)$ estimates which are slightly larger than the ground truth. This can be explained by the fact that the density evaluation of the continuous normalizing flows uses the Hutchinson trace estimator for evaluating the divergence and a numerical method for solving the ODE. Therefore, we have a small error in the density propagation of the neural JKO steps. This error is amplified by the rejection steps since samples with underestimated density are more likely to be rejected than samples with overestimated density.*

*We would like to emphasize that this effect only appears for examples, where the energy distance between generated and ground truth samples is in the same order of magnitude like the average distance between two different sets of ground truth samples (see Table 1). This means that in the terms of the energy distance the generated and ground truth distribution are indistinguishable. At the same time the bias in the $\log(Z)$ estimate appears at the third or fourth relevant digit meaning that it is likely to be negligible.*

### E.3 Additional Figures and Evaluations

Additionally to the results from the main part of the paper, we provide the following evaluations.

**Visualization of the Rejection Steps**   In Figure 1 and with involved steps in Figure 2, we visualize the different steps of our importance corrected neural JKO model on the shifted 8 Peaky example. Due to the shift of the modes, the modes on the right attract initially more mass than the ones on the left. In the rejection layers, we can see that samples are mainly rejected in oversampled regions such that the mode weights are equalized over time. This can also be seen in Figure 3, where we plot the weights of the different modes over time. We observe, that these weights are quite imbalanced for the latent distribution but are equalized over the rejection steps until they reach the ground truth value.

**(Marginal) Plots of Generated Samples**   Despite the quantitative comparison of the methods in the Tables 1, 2 and 4, we also plot the first to coordinates of generated samples for the different test distributions for all methods for a visual comparison.

In Figure 4, we plot samples of the shifted 8 Modes example. We can see that all methods roughly cover the ground truth distribution even though CRAFT and DDS have slight and the uncorrected neural JKO scheme has a severe imbalance in the assigned mass for the different modes.

For the shifted 8 Peaky example in Figure 5, we see that this imbalance increases drastically for CRAFT, HMC, MALA and the uncorrected neural JKO. Also DDS has has a slight imbalance, while the importance corrected neural JKO scheme fits the ground truth almost perfectly.

The Funnel distribution in Figure 6 has two difficult parts, namely the narrow but high-density funnel on the left and the wide moderate density fan on the right. We can see that DDS does cover none of them very well. Also MALA, CRAFT and the uncorrected neural JKO do not cover the end of the funnel correctly and have also difficulties to cover the fan. HMC covers the fan well, but not the funnel. Only our importance corrected neural JKO scheme covers both parts in a reasonable way.

For the Mustache distribution in Figure 7, the critical parts are the two heavy but narrow tails. We observe that CRAFT and DDS are not able to cover them at all, while HMC and MALA and uncorrected neural JKO only cover them only partially. The importance corrected neural JKO covers them well.

Finally, for the GMM-$d$ example we consider the dimensions $d = 10$ and $d = 200$ in the Figures 8 and 9. We can see that CRAFT mode collapses, i.e., for $d = 10$ it already finds some of the modes and for $d = 200$ it only finds one mode. DDS, HMC, MALA and the uncorrected neural JKO find all modes but do not distribute the mass correctly onto all modes. While this already appears for $d = 10$ it is more severe for $d = 200$. The importance corrected neural JKO sampler finds all modes and distributes the mass accurately.

**Development of Error Measures over the Steps**   We plot how the quantities of interest decrease over the application of the steps of our model. The results are given in the Figure 10. It can be observed that the different steps may optimize different metrics. While the rejection steps improve the $\log(Z)$ estimate more significantly, the JKO steps focus more on the minimization of the energy distance. Overall, we see that in all figures the errors decrease over time.

### E.4 Implementation Details

To build our importance corrected neural JKO model, we first apply $n_1 \in \mathbb{N}$ JKO steps followed by $n_2 \in \mathbb{N}$ blocks consisting out of a JKO step and three importance-based rejection steps. The velocity fields of the normalizing flows are parameterized by a dense three-layer feed-forward neural network. For the JKO steps, we choose an initial step size $\tau_0 > 0$ and then increase the step size exponentially by $\tau_{k+1} = 4\tau_k$. The choices of $n_1$, $n_2$, $\tau_0$ and the number of hidden neurons from the networks is given in Table 5 together with the execution times for training and sampling. For evaluating the density propagation through the CNFs, we use the Hutchinson trace estimator with 5 Rademacher distributed random vectors whenever $d > 5$ and the exact trace evaluation otherwise. For implementing the CNFs, we use the code from Ffjord (Grathwohl et al., 2019) and the `torchdiffeq` library by Chen (2018). We provide the code in the supplementary material.

Table 5: Parameters, training and sampling times for the different examples. For the sampling time we draw $N = 50000$ samples once the method is trained. The execution times are measured on a single NVIDIA RTX 4090 GPU with 24 GB memory.

| Parameter | Distribution | | | | | | | | | |
|---|---|---|---|---|---|---|---|---|---|---|
| | Mustache | shifted 8 Modes | shifted 8 Peaky | Funnel | GMM-10 | GMM-20 | GMM-50 | GMM-100 | GMM-200 | LGCP |
| Dimension | 2 | 2 | 2 | 10 | 10 | 20 | 50 | 100 | 200 | 1600 |
| Number $n_1$ of flow steps | 6 | 2 | 2 | 6 | 4 | 4 | 4 | 4 | 5 | 3 |
| Number $n_2$ of rejection blocks | 6 | 4 | 4 | 6 | 6 | 6 | 7 | 8 | 8 | 6 |
| Initial step size $\tau_0$ | 0.05 | 0.01 | 0.01 | 5 | 0.0025 | 0.0025 | 0.0025 | 0.0025 | 0.001 | 5 |
| Hidden neurons | 54 | 54 | 54 | 256 | 70 | 90 | 150 | 250 | 512 | 1024 |
| batch size | 5000 | 5000 | 5000 | 5000 | 5000 | 5000 | 5000 | 5000 | 2000 | 500 |
| Required GPU memory | 5 GB | 5 GB | 5 GB | 5 GB | 5 GB | 5 GB | 5 GB | 5 GB | 6 GB | 11 GB |
| Training time (min) | 38 | 20 | 21 | 107 | 33 | 34 | 44 | 53 | 80 | 163 |
| Sampling time (sec) | 15 | 2 | 3 | 79 | 22 | 23 | 72 | 129 | 473 | 535 |

For the MALA and HMC we run an independent chain for each generated samples and perform 50000 steps of the algorithm. For HMC we use 5 momentum steps and set the step size to 0.1 for 8Modes and Funnel and step size 0.01 for mustache. To stabilize the first iterations of MALA and HMC we run the first iterations with smaller step sizes (0.01 times the final step size in the first 1000 iterations and 0.1 times the final step size for the second 1000 iterations). For MALA we use step size 0.001 for all examples. For DDS and CRAFT we use the official implementations. In particular for the test distributions such as Funnel and LGCP we use the hyperparameters suggested by the original authors. For the other examples we optimized them via grid search.

## F  COMPUTATIONAL ASPECTS OF NORMALIZING FLOWS

In this appendix, we give some more details about the computational aspects of the normalizing flows in our model. First, we discuss the relation between multimodal target distributions, mode collapse and non-convex loss functions of normalizing flows. Afterwards, we discuss some computational aspects of continuous normalizing flows like ODE discretizations and density evaluations with trace estimators. Finally, we run some standard normalizing flow networks on our numerical example distributions and compare them with our importance corrected neural JKO sampling.

### F.1  MULTIMODALITIES, MODE COLLAPSE AND NON-CONVEX LOSS FUNCTIONS

For the sampling application, normalizing flows are usually trained with the reverse KL loss function $\mathcal{F}(\mu) = \mathrm{KL}(\mu, \nu)$, see, e.g., Marzouk et al. (2016). More precisely, a normalizing flow aims to learn the parameters $\theta$ of a family of diffeomorphisms $\mathcal{T}_\theta$ such that $\mathcal{T}_{\theta\#}\mu_0 \approx \nu$ by minimizing $\mathcal{F}(\mathcal{T}_{\theta\#}\mu_0)$. In the case that $\nu$ is multimodal it can be observed that this training mode collapses. That is, the approximation $\mathcal{T}_{\#}\mu_0$ covers not all of the modes of $\nu$ but instead neglects some of them. Examples of this phenomenon can be seen in the numerical comparison in Appendix F.3. One reason for this effect is that the functional $\mathcal{F}$ is convex if and only if $\nu$ is log-concave and therefore unimodal, see (Ambrosio et al., 2005, Prop. 9.3.2). In particular, for multimodal $\nu$, the functional $\mathcal{F}$ is non-convex. In the latter case of, then the mode collapses can correspond to the convergence to a local minimum, as the following example shows.

**Example 17.** *We consider the case $d = 1$ and the target distribution*

$$\nu = \tfrac{1}{2}\mathcal{N}(-\tfrac{1}{2}, 0.05^2) + \tfrac{1}{2}\mathcal{N}(\tfrac{1}{2}, 0.05^2).$$

*As latent distribution $\mu_0$ we choose a standard Gaussian. Then, we parametrize the normalizing flow $\mathcal{T}_\theta = (1-\theta)T_0 + \theta\mathcal{T}_1$ for $\theta \in [0, 1]$, where $\mathcal{T}_0$ is the optimal transport map between $\mu_0$ and $\nu$ and $\mathcal{T}_1$ is the optimal transport map between $\mu_0$ and $\mathcal{N}(\tfrac{1}{2}, 0.05^2)$. In particular, $\mathcal{T}_{\theta\#}\mu_0$ perfectly recovers the target distribution for $\theta = 0$ and produces a mode collapsed version of it for $\theta = 1$. Now, we plot the reverse KL loss function $\mathcal{L}(\theta) = \mathrm{KL}(\mathcal{T}_{\theta\#}\mu_0, \nu)$ and the densities of the generated distributions $\mathcal{T}_{\theta\#}\mu_0$ in Figure 11. We observe that it has two local minima for $\theta = 0$ and $\theta = 1$ (for $\theta$ outside of $[0, 1]$, $\mathcal{T}$ is no longer invertible), where $\theta = 0$ is the perfectly learned parameter and $\theta = 1$ is the mode collapsed version. At the same time, we note that the curve $\theta \mapsto \mathcal{T}_{\theta\#}\mu_0$ is a geodesic in the Wasserstein space. In particular, the non-convexity of $\mathcal{L}$ is a direct consequence of the fact that $\mathcal{F}(\mu) = \mathrm{KL}(\mu, \nu)$ is geodesically non-convex.*

**Remark 18** (Motivation of Wasserstein Regularization). *This connection between the non-convexity of the loss function $\mathcal{F}$ and mode collapse also motivates the Wasserstein regularization from Section 3.*

*By considering the loss function $\mathcal{G}(\mu) = \mathcal{F}(\mu) + \frac{1}{2\tau}W_2^2(\mu, \mu_\tau^k)$ for $\tau > 0$ small enough instead of $\mathcal{F}$, we obtain a loss function which is convex in the Wasserstein space, see (Ambrosio et al., 2005, Lem. 9.2.7). Even though this does not imply that the map $\theta \mapsto \mathcal{G}(\mathcal{T}_{\theta\#}\mu_0)$ is convex, we can expect that the training does not get stuck in local minima as long as the architecture of $\mathcal{T}_\theta$ is expressive enough.*

*Theoretically, we need for $\lambda$-convex $\mathcal{F}$ that $\tau \leq \frac{1}{\lambda}$ to ensure that $\mathcal{G}$ is geodesically convex. However, if $\mu^k$ is already close to a minimum of $\mathcal{F}$, then it can be sufficient that the functional $\mathcal{G}$ is convex locally around $\mu^k$. In this case, the distribution generated by the CNF will stay in this neighborhood. Note that in practice the constant $\lambda$ is unknown such that we start with a small step size $\tau_0$ and increase it over time as outlined at the end of Section 4 and in Appendix E.4.*

### F.2 COMPUTATIONAL LIMITATIONS OF (CONTINUOUS) NORMALIZING FLOWS

Similar to the literature on neural JKO schemes (Vidal et al., 2023; Xu et al., 2024), our implementation of the neural JKO scheme relies on continuous normalizing flows (Chen et al., 2018). This comes with some limitations and challenges, which were extensively discussed in (Chen et al., 2018). Since they are relevant for our method, we give a synopsis below.

**Derivatives of ODE Solutions**   For the training phase we need the derivative of the solution of an ODE. For this, we use the `torchdiffeq` package (Chen, 2018). In particular, this package does not rely backpropagation through the steps from the forward solver. Instead, it overwrites the backward pass of the ODE by solving an *adjoint ODE*. This avoids expensive tracing in the automatic differentiation process within the forward pass and keeps the memory consumption low. Moreover, the quadratic regularization of the velocity field leads to straight paths such that the adaptive solvers only require a few steps for solving the ODE, see (Onken et al., 2021) for a detailed discussion. Indeed, we use in our numerical examples the `dopri5` solver from `torchdiffeq`, which is an adaptive Runge-Kutta method. However, we still have to solve two ODEs during the training time. This still can be computational costly, in particular when the underlying model is large.

**Trace Estimation for Density Computations**   In order to evaluate the (log-)density of our model, we have to compute the divergence $\text{div } v_\theta = \text{trace}(\nabla v_\theta(z_\theta(x,t), t))$ of the learned vector field $v_\theta$ (and integrate it over time), see (6). Computing the trace of the Jacobian of $v_\theta$ becomes computational costly in high dimensions. As a remedy, we consider the Hutchinson trace estimator (Hutchinson, 1989), which states that for any matrix $A$ and a random vector $z$ with mean zero and identity covariance matrix it holds that $\mathbb{E}[z^\mathrm{T}Az] = \text{trace}(A)$. Applying this estimator to the divergence, we obtain that the divergence coincides with $\mathbb{E}[z^\mathrm{T}\nabla v_\theta(z_\theta(x,t),t)z]$. The integrand is now again a Jacobian-vector product which can be computed efficiently. Finally, we estimate the trace by empirically discretizing the expectation by finitely many realizations of $z$.

In our numerics, we choose $z$ to be Rademacher random vectors, i.e. each entry is with probability $\frac{1}{2}$ either $-1$ or $1$. For the training of the continuous normalizing flow, an unbiased low-precision estimator is sufficient, such that we discretize the expectation with one realization of $z$. During evaluation, we require a higher precision and use $5$ realizations instead.

**Initialization**   In order to find a stable initialization of the model, we initialize the last layer of the velocity field $v_\theta$ with zeros such that it holds $v_\theta(x, t) = 0$ for all $x \in \mathbb{R}^d$, $t \in [0, \tau]$. In this case, the solution $z_\theta$ of the ODE $\dot{z}_\theta = v_\theta$ with initial condition $z_\theta(x, 0) = x$ is given by $z_\theta(x, t) = x$ for all $t$. In particular, we have that the transport map $\mathcal{T}_\theta = z_\theta(x, \tau)$ is the identity such that the generated distribution $\mathcal{T}_{\theta\#}\mu_0$ coincides with the latent distribution for the initial parameters.

**Remark 19** (Other Normalizing Flow Architectures)**.** *In general any architecture $\mathcal{T}_\theta$ of normalizing flows can be considered in the neural JKO scheme. To this end, we can replace the velocity field regularization in (4) by the penalizer $\int_{\mathbb{R}^d} \|\mathcal{T}_\theta(x) - x\|^2 \mathrm{d}\mu_\tau^k$. By Breniers' theorem (Brenier, 1987) this is again equivalent to minimizing the Wasserstein distance. However, while discrete-time normalizing flows like coupling-based networks (Dinh et al., 2016; Kingma & Dhariwal, 2018) and autoregressive flows (De Cao et al., 2020; Durkan et al., 2019; Huang et al., 2018; Papamakarios et al., 2017) are often faster than CNFs, this is not generally true in our setting, since their evaluation time does not benefit from the OT-regularization. Additionally, we observed numerically that the expressiveness of discrete-time architectures scale much worse to high dimensions and are less stable to train. Nevertheless, the evaluation can be cheaper and the density evaluation since these architectures*

Table 6: Comparison of neural JKO IC with normalizing flows trained with the reverse KL loss function. We evaluate the energy distance (smaller values are better), the $\log(Z)$-estimation (larger values are better) and the expected squared empirical Wasserstein distance (smaller values are better).

| Distribution | Energy distance | | | $\log(Z)$-estimation | | | squared Wasserstein-2 | | |
|---|---|---|---|---|---|---|---|---|---|
| | continuous NF | coupling NF | neural JKO IC | continuous NF | coupling NF | neural JKO IC | continuous NF | coupling NF | neural JKO IC |
| Mustache | $2.1 \times 10^{-2}$ | $4.9 \times 10^{-3}$ | $2.9 \times 10^{-3}$ | $-7.1 \times 10^{-2}$ | $-2.2 \times 10^{-2}$ | $-7.3 \times 10^{-3}$ | $3.8 \times 10^{+1}$ | $3.1 \times 10^{+1}$ | $1.7 \times 10^{+1}$ |
| shifted 8 Modes | $4.1 \times 10^{-1}$ | $4.1 \times 10^{-2}$ | $1.2 \times 10^{-5}$ | $-1.4 \times 10^{0}$ | $-4.3 \times 10^{-1}$ | $+5.1 \times 10^{-6}$ | $1.4 \times 10^{0}$ | $2.2 \times 10^{-1}$ | $6.5 \times 10^{-3}$ |
| shifted 8 Peaky | $5.7 \times 10^{-1}$ | $5.8 \times 10^{-1}$ | $3.4 \times 10^{-5}$ | $-2.1 \times 10^{0}$ | $-2.1 \times 10^{0}$ | $-2.1 \times 10^{-3}$ | $1.8 \times 10^{0}$ | $1.8 \times 10^{0}$ | $7.2 \times 10^{-3}$ |
| Funnel | $1.9 \times 10^{-1}$ | $2.2 \times 10^{-3}$ | $1.4 \times 10^{-2}$ | $-9.4 \times 10^{-1}$ | $-2.8 \times 10^{-2}$ | $-7.1 \times 10^{-3}$ | $3.8 \times 10^{+2}$ | $8.1 \times 10^{+2}$ | $8.5 \times 10^{+2}$ |
| GMM-10 | $5.4 \times 10^{-1}$ | $5.3 \times 10^{-1}$ | $5.2 \times 10^{-5}$ | $-2.3 \times 10^{0}$ | $-2.3 \times 10^{0}$ | $+3.5 \times 10^{-3}$ | $3.4 \times 10^{0}$ | $3.5 \times 10^{0}$ | $1.4 \times 10^{-1}$ |
| GMM-20 | $1.1 \times 10^{0}$ | $1.1 \times 10^{0}$ | $1.1 \times 10^{-4}$ | $-2.4 \times 10^{0}$ | $-2.3 \times 10^{0}$ | $+6.4 \times 10^{-3}$ | $9.5 \times 10^{0}$ | $9.4 \times 10^{0}$ | $3.7 \times 10^{-1}$ |

*do not have to deal with trace estimators. Residual architectures (Behrmann et al., 2019; Chen et al., 2019) is at least similar to continuous normalizing flows, but they are very expansive to train and evaluate, and additionally rely on trace estimators for computing the density of the generated samples.*

### F.3 NUMERICAL COMPARISON

Finally, we compare our neural JKO IC scheme with standard normalizing flows trained with the reverse KL loss function as proposed in Marzouk et al. (2016). In particular, we aim to demonstrate the benefits of our neural JKO IC scheme to avoid mode collapse.

To this end, we train two architectures of normalizing flows for our examples using the reverse KL loss function as proposed in Marzouk et al. (2016). First, we use a continuous normalizing flow with the same architecture as used in the neural JKO (IC). That is, we parameterize the velocity field $v_\theta$ by a dense neural network with three hidden layers and the same hidden dimensions as in Table 5. Second, we compare with a coupling network with 5 Glow-coupling blocks (Kingma & Dhariwal, 2018), where the coupling blocks have two hidden layers with three times the hidden dimension as in Table 5. We found numerically that choosing larger architectures does not bring significant advantages.

We plot some generated distributions of the continuous and coupling normalizing flows as well as the neural JKO IC scheme in Figure 12. As we have already seen in the previous examples, the neural JKO IC scheme is able to recover multimodal distributions almost perfectly. On the other side, the normalizing flow architectures always collapse to one or a small number of modes. We additionally report the error measures in Table 6. We can see that neural JKO IC performs always better than the normalizing flow architectures (note that for the Funnel distribution all $W_2^2$-values are below the sampling error reported in Table 3). For multimodal distributions, the normalizing flow approximations are by several orders of magnitude worse than neural JKO IC, while they work quite well for unimodal distributions (Mustache and Funnel).

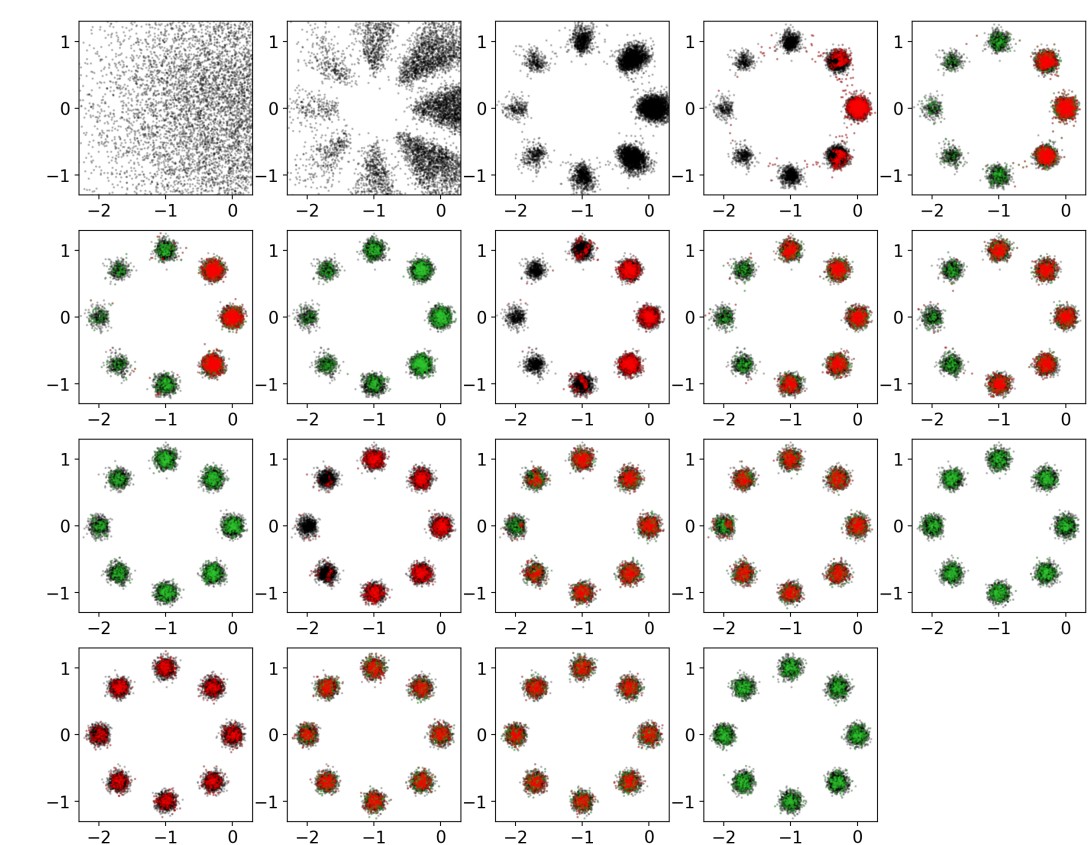

Figure 2: Visualization of the steps of the importance corrected neural JKO model for the shifted 8 Peaky example. We start at the top left with the latent distribution and apply in each image one step from the model. The red color indicates samples which are rejected in the next step and the green color marks the resampled points.

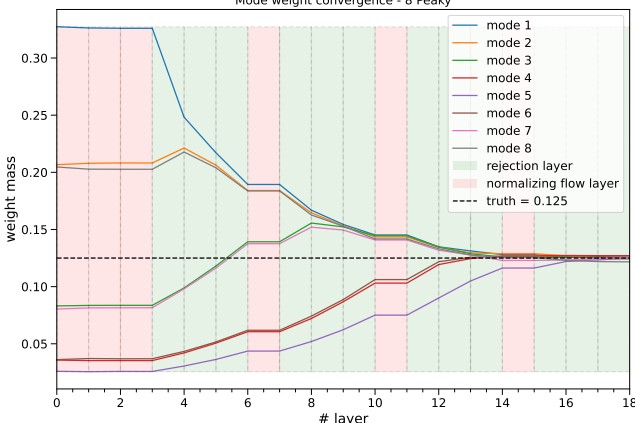

Figure 3: Plot of the mode weights for the 8 Peaky example over the different layers of the importance corrected neural JKO model. We observe that the weights are mainly changed by the rejection steps and not by the neural JKO steps.

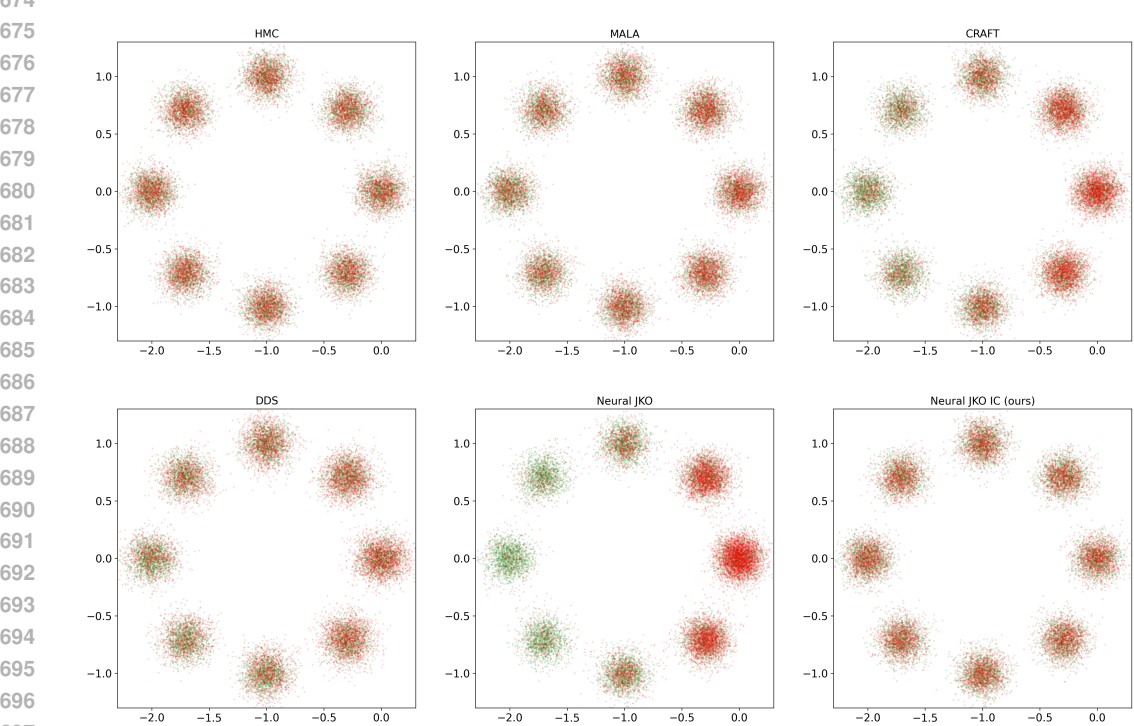

Figure 4: Sample generation with various methods for shifted 8 Modes example with ground truth samples and generated samples for each associated method. While most methods recover the distribution well, we can see a imbalance in the modes for the uncorrected neural JKO, CRAFT and DDS.

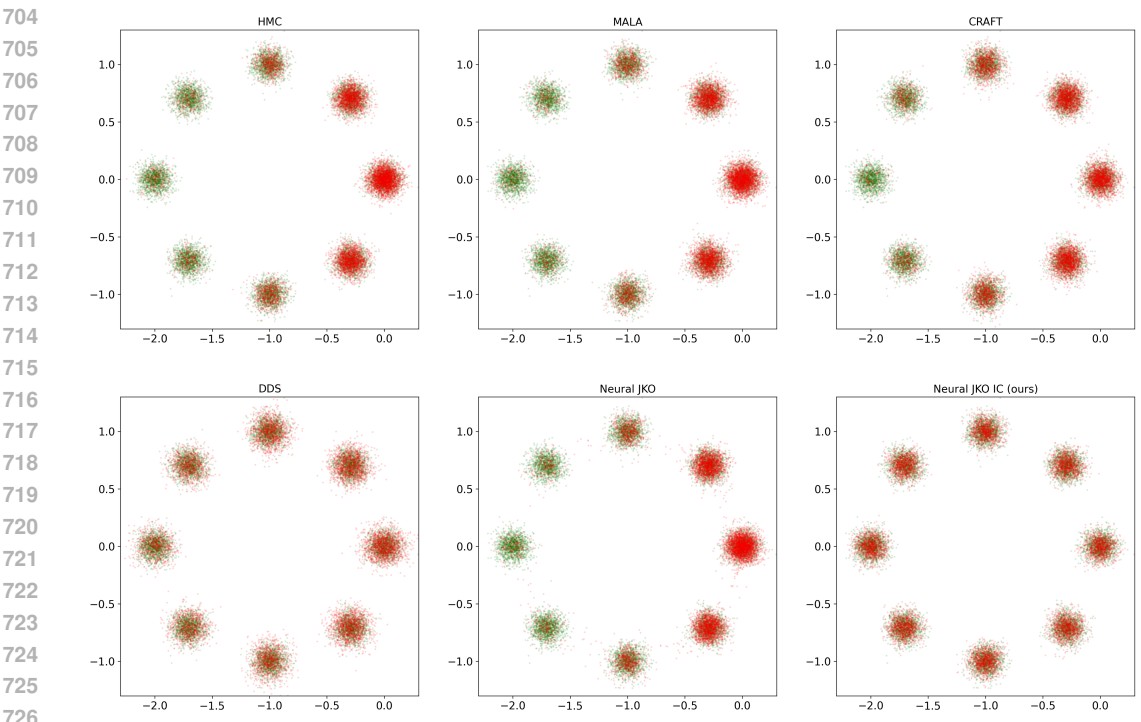

Figure 5: Sample generation with various methods for shifted 8 Peaky mixtures with ground truth samples and generated samples for each associated method. We can see a severe imbalance among the modes for HMC, MALA, CRAFT and neural JKO. Also DDS has a slight imbalance between the modes.

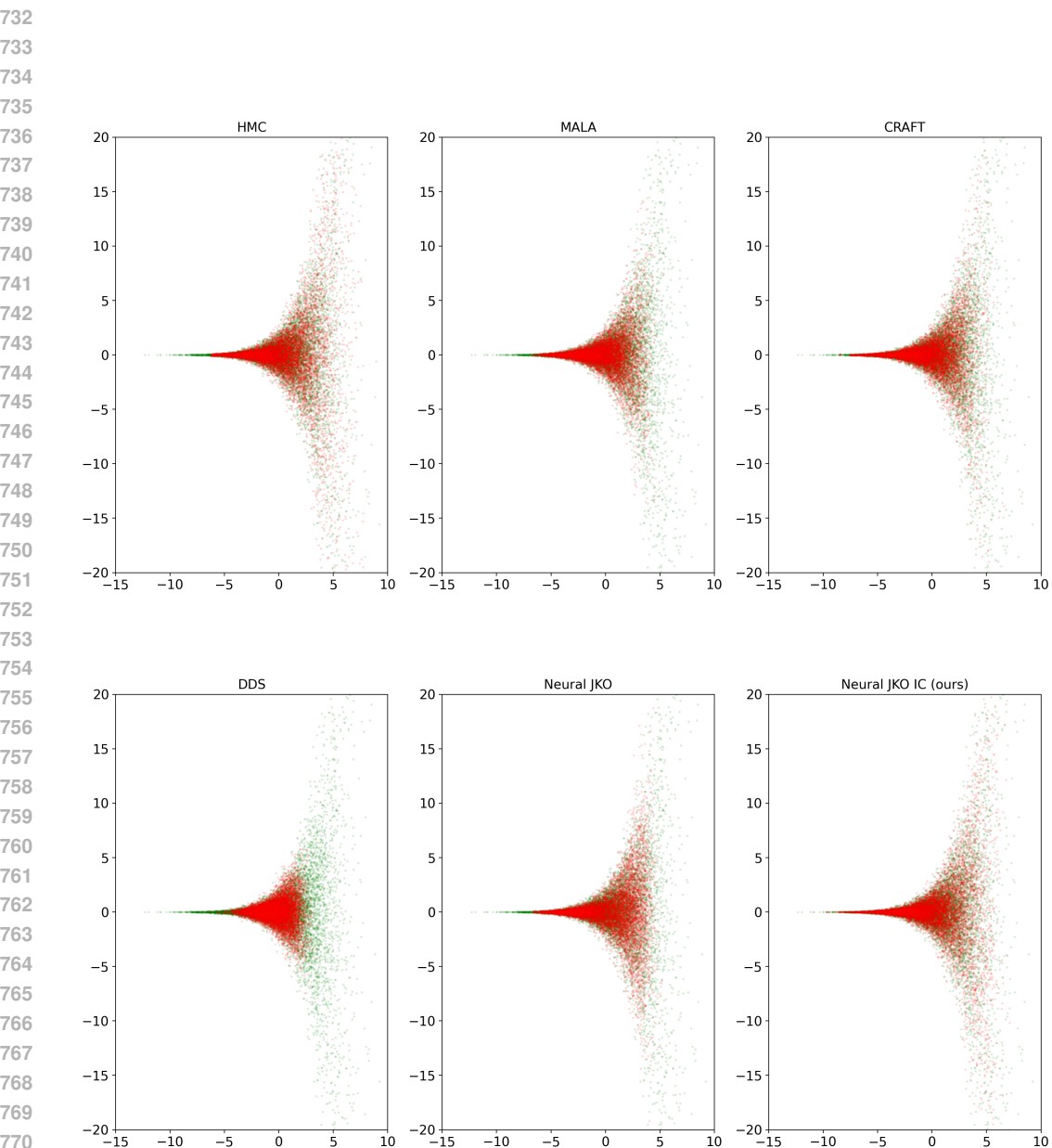

Figure 6: Marginalized sample generation with various methods for the $d = 10$ funnel distribution with ground truth samples and generated samples for each associated method. We observe that only the importance corrected neural JKO covers the thin part of the funnel well.

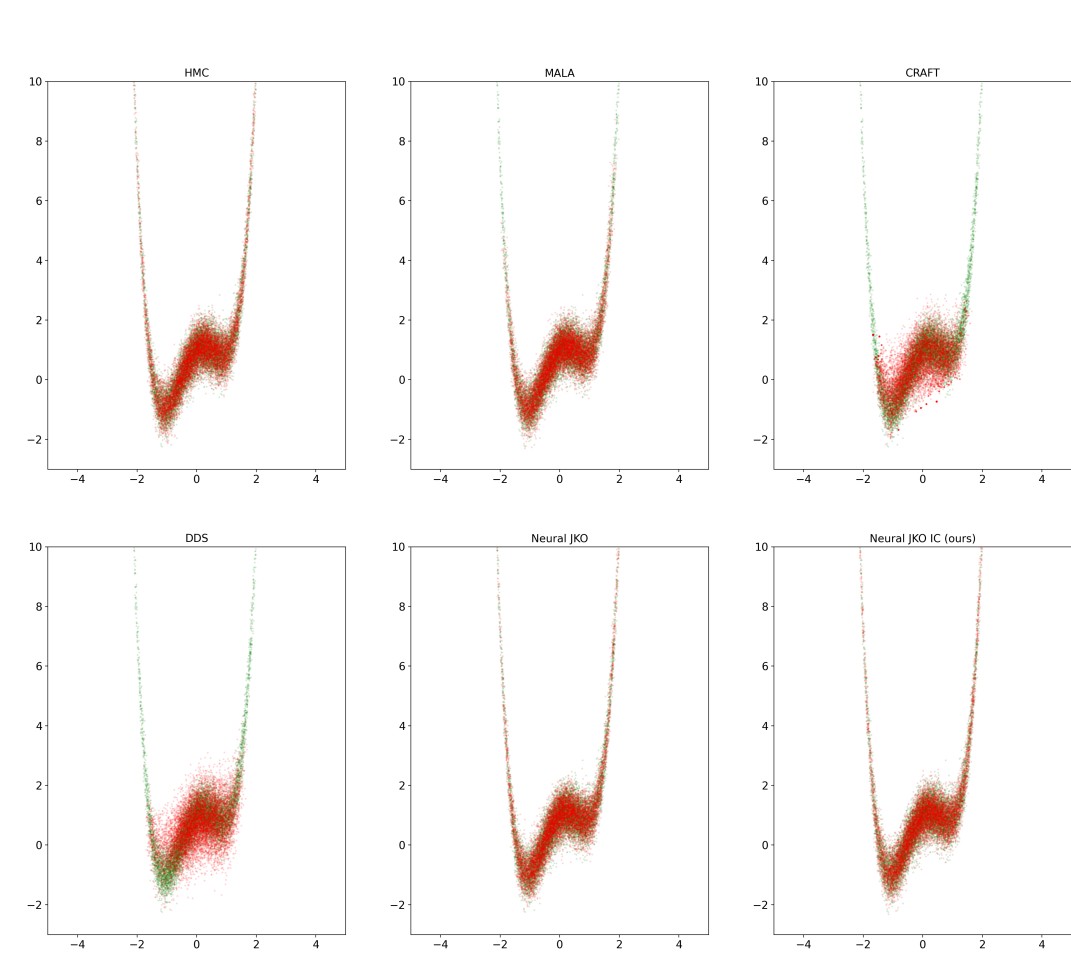

Figure 7: Sample generation with various methods for the $d = 2$ mustache distribution with ground truth samples and generated samples for each associated method. We can see that MALA, CRAFT and DDS have difficulties to model the long tails of the distribution properly.

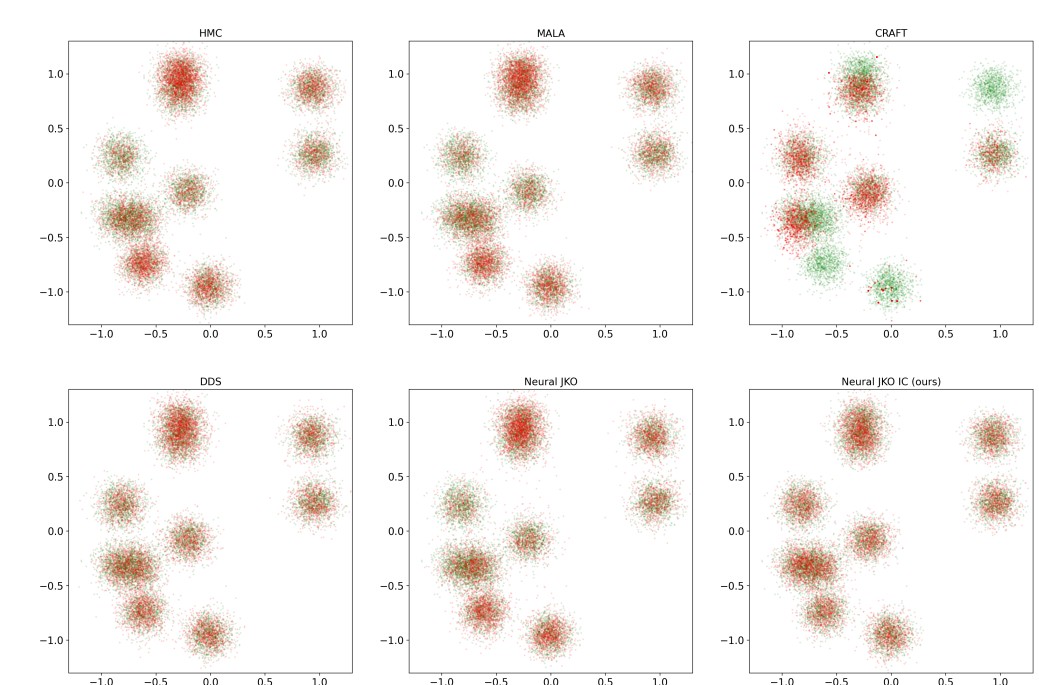

Figure 8: Marginalized sample generation with various methods for the GMM-10 distribution with ground truth samples and generated samples for each associated method. We observe that CRAFT mode collapses and that only the importance corrected neural JKO model distributes the mass correctly onto the modes.

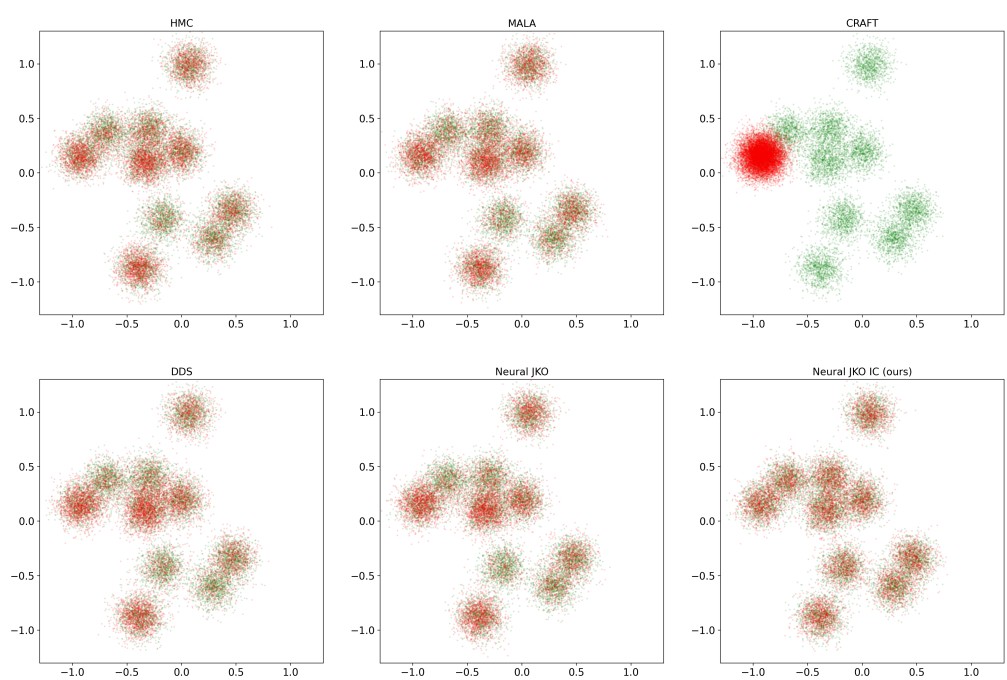

Figure 9: Marginalized sample generation with various methods for the GMM-200 distribution with ground truth samples and generated samples for each associated method. We observe that CRAFT mode collapses and that only the importance corrected neural JKO model distributes the mass correctly onto the modes.

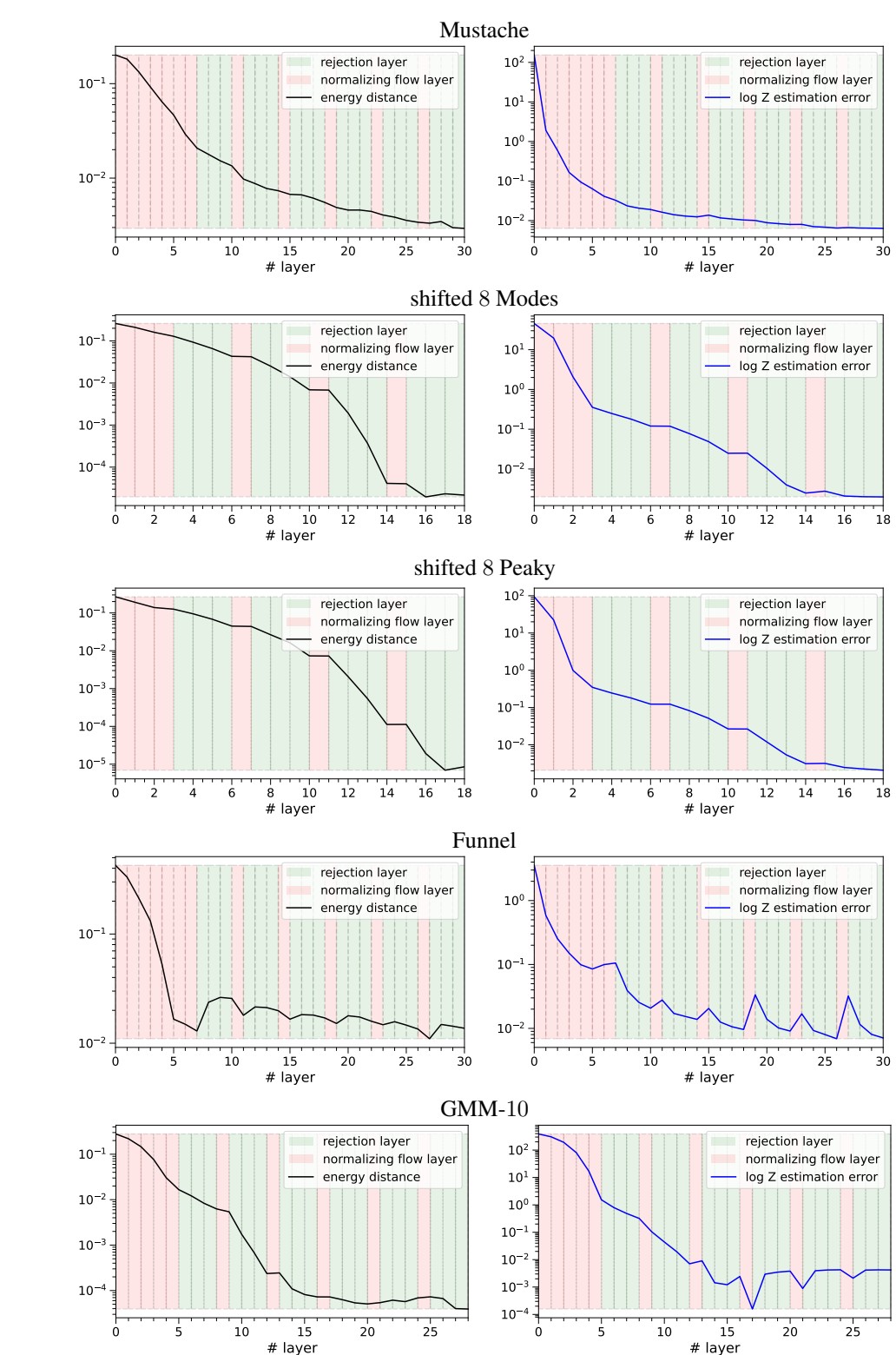

Figure 10: We plot the energy distance (**left**) and $\log(Z)$ estimate (**right**) over the steps of our importance corrected neural JKO method for different examples. We observe that the error measures decrease in the beginning and then saturate at some value.

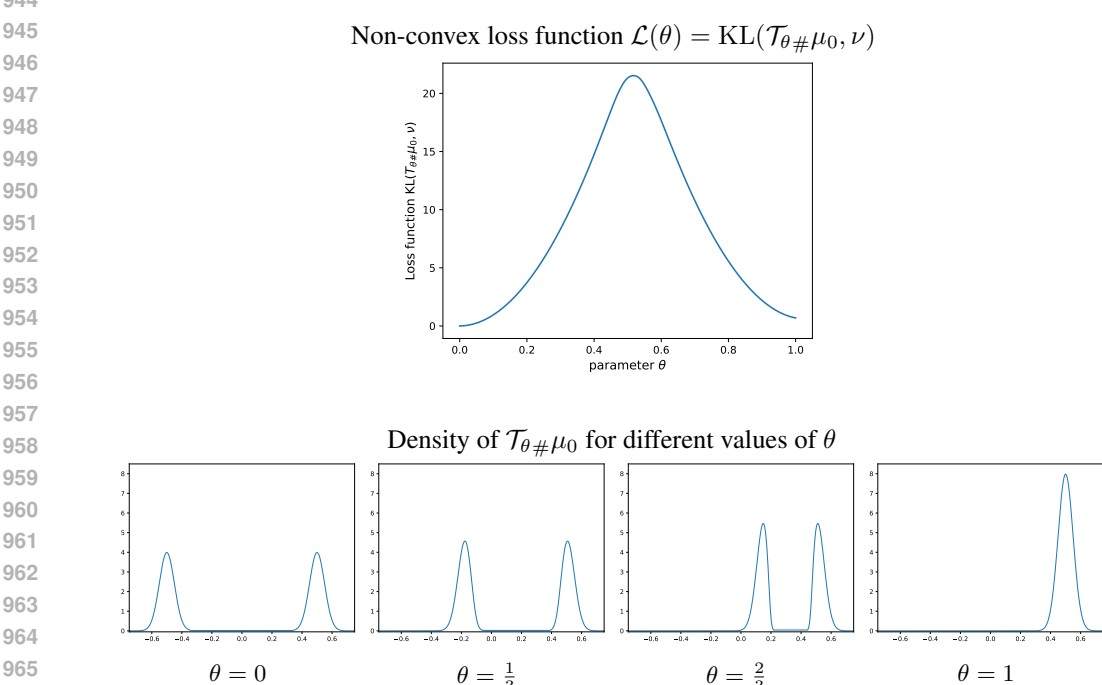

Figure 11: Illustration of the loss function $[0,1] \ni \theta \mapsto \mathcal{L}(\theta) = \mathrm{KL}(\mathcal{T}_{\theta\#}\mu_0, \nu)$ and the densities of the generated distributions $\mathcal{T}_{\theta\#}\mu_0$ from Example 17. Both values $\theta = 0$ and $\theta = 1$ correspond to local minima (note that $\mathcal{L}(1) > 0 = \mathcal{L}(0)$). In particular $\theta = 1$ corresponds to the case of mode collapse.

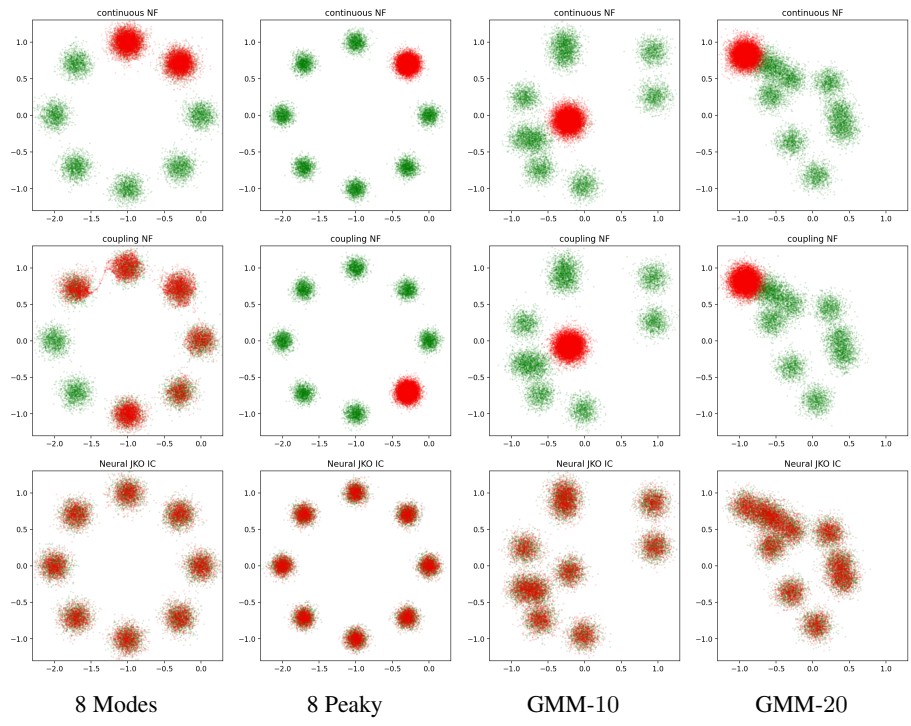

Figure 12: Marginalized sample generation for a single normalizing flow compared with neural JKO IC for different example distributions with ground truth samples and generated samples for each associated method. We observe that the standard normalizing flow architectures always collapse to one or few modes while neural JKO IC recovers all modes correctly.

