# OpenReview forum: "Importance Corrected Neural JKO Sampling"
_ICLR.cc/2025/Conference — Submitted to ICLR 2025_

### Official Review · Reviewer_wcjw · 2024-11-03

**Soundness:** 3
**Presentation:** 3
**Contribution:** 3
**Rating:** 6
**Confidence:** 3

**Summary:**

In this article, the author(s) consider the problem of sampling from unnormalized density functions, and the basic idea is to implicitly construct a diffeomorphism to transport a simple distribution to the target distribution, based on continuous normalizing flows (CNFs). To further improve the quality of sampling, a rejection-resampling step is also used.

**Strengths:**

Sampling is a fundamental task in machine learning, and this article tackles this problem by utilizing some recent advances in deep generative modeling. The use of JKO scheme to iteratively optimizing the model parameters is also interesting.

**Weaknesses:**

1. Overall I like the idea of this article, but one of my majors concerns on this article is the motivation of using a continuous normalizing flow (CNF) rather than the classical discrete-time normalizing flow. If my understanding is correct, CNF uses neural networks to model the velocity field $v_\theta(\cdot)$, and then the transport map $\mathcal{T}\_\theta(\cdot)$ is implicitly defined by $v_\theta(\cdot)$. However, when minimizing the (reverse) KL loss function, it would be easier to work on $\mathcal{T}\_\theta(\cdot)$, which is what classical normalizing flows do. Also, the sampling of classical normalizing flows should be faster than CNFs, since they do not need to solve an ODE.

2. Moreover, for classical normalizing flows, the evaluation of the (log-)density function is *exact*, which is important for rejection sampling. CNF needs to approximate the Jacobian term in evaluating the density.

3. There are some notable existing works on sampling using diffeomorphisms/normalizing flows, for example [1][2][3]. When comparing deep-learning-based sampling algorithms for the experiments, I would suggest at least adding one of them, since they are closer to the method used in this article.

4. In the last paragraph of the introduction section, I think it is helpful to distinguish *modeling* methods (learning from data) with *sampling* methods (learning from a density function). The latter is closer to the topic of this article.

5. For the evaluation metric in the experiments, isn't Wasserstein distance a more natural choice? It has the benefit of avoiding selecting the kernel as a hyperparameter.

[1] Marzouk, Y., Moselhy, T., Parno, M., & Spantini, A. (2016). An introduction to sampling via measure transport. arXiv preprint arXiv:1602.05023.

[2] Hoffman, M., Sountsov, P., Dillon, J. V., Langmore, I., Tran, D., & Vasudevan, S. (2019). Neutra-lizing bad geometry in hamiltonian monte carlo using neural transport. arXiv preprint arXiv:1903.03704.

[3] Qiu, Y., & Wang, X. (2024). Efficient multimodal sampling via tempered distribution flow. Journal of the American Statistical Association.

**Questions:**

Besides the questions in the "Weaknesses" section, I have a few other minor ones that I hope the author(s) can clarify.

1. In Theorem 3, the curve is denoted as $\gamma_\tau$, but I only see $\tilde{\gamma}_\tau$ in equation (3). Are they the same?

2. In Section 4, the "Annealed Importance Sampling" paragraph, I would like to point out that the self-normalized estimator $\sum_{i=1}^N w_i f(x_i)/(\sum_{i=1}^N w_i)$ is in general not unbiased, since the denominator is also a random variable. The numerator itself $\sum_{i=1}^N w_i f(x_i)$ is actually unbiased. The self-normalized version is used when $w_i$ is computed from unnormalized densities.

3. In the introduction section, I see "this regularization converts the objective functional into a convex one". Is there any theoretical guarantee for this?

---

> ### Author Response · Authors · 2024-11-17
> **Response to the Reviewer**
>
> Thank you very much for the review. We carefully revised the paper and inserted the requested changes.
>
> ## Regarding Weaknesses
> 1. and 2.
>  During writing the paper, we also experimented with discrete-time architectures for normalizing flows, mainly with coupling flows similar to the Glow model (Kingma and Dhariwal 2018). Compared to the continuous normalizing flows we made the following observations, which also align with the paper of Onken et al. 2021 ("OT-flow: Fast and accurate continuous normalizing flows via optimal transport"):
>
>     - Since the regularization of the velocity field in the continuous normalizing flows (CNFs) leads to geodesic interpolations, the particle trajectories become straight paths. Hence, adaptive solvers only require very few steps to solve the involved ODE. Therefore, training and sampling with OT-regularized CNFs is much faster than with standard CNFs, this is discussed in detail by Onken et al. 2021. In particular, we observed that the training does not speed up significantly with the coupling flows, in high demensions, when large subnetworks are required, discrete-time architectures can be even slower than the ODE formulation.
>
>     - Moreover, we observed in our experiments that the expressiveness of discrete-time architectures scales worse to high dimensions than the expressiveness of continuous normalizing flows.
>
>     - On the other hand, we agree that discrete-time normalizing flows can improve the evaluation time and accuracy due to the exact density evaluations. However, we can also achieve sufficient accuracy for the density evaluation with trace estimators in the CNFs to meet the requirements for the rejection steps.
>
> We added a discussion on other architectures (Remark 19) and point to it at the end of Section 3.2. Moreover, also considering the comments of the other reviewers, we added some more details on the computational cost of OT-regularized CNFs and trace estimators (Appendix F.2) and point to these remarks in the limitations section.
>
> 3. We added a comparison with two kinds of normalizing flows with the loss function from Marzouk et al. 2016 in Appendix F.3. First, we use a continuous normalizing flow architecture (since this is what we have used for our method). Second, (also related to point 1. and 2.) we used an explicit coupling architecture based on triangular blocks. We can observe that this works mostly fine for the unimodal examples (good results for funnel and mustache), but often mode collapses for multimodal distributions. We added Figure 12 for an experimental illustration of the prescribed situation.
>
> 4. Thanks for the comment. We agree and rewrote the related work section such that we now clearly indicate which references consider sampling and which consider generative modeling.
>
> 5. The Wasserstein distance has several drawbacks as evaluation metric. Most important, its sample complexity depends exponentially on the dimension such that the empirical version of the Wasserstein distance differs significantly from continuous version. Despite that it is not independent of our model (which is based on the Wasserstein geometry) and suffers from heavy computation times for large number of samples (cubic scaling). Nevertheless, we added a table evaluating the empirical Wasserstein distance between generated and ground-truth samples in the appendix (Table 3). The conclusions are similar as for the energy distance.
>
> ## Regarding Questions
>
> 1. This was a typo, which we have now corrected. Thanks for spotting it!
>
> 2. You are absolutely right. We corrected the error. Many thanks!
>
> 3. This comes from the fact that we are adding up a $\lambda$-convex function and $1/\tau$ times the Wasserstein distance (which is $1$-convex). Hence we obtain a function which is $(\lambda+\frac1\tau)$-convex, so in particular strongly convex for $\tau>-\lambda$. A proof of this statement can be found in Lemma 9.2.7 of Ambrosio et al. 2005. We added the statement before Section 3.1. Additionally, we added a Section on the connection between multimodal target distributions, non-convexity of the KL divergence and mode collapse in the Appendix F.1. The new appendix F.3 shows in numerical examples that normalizing flows mode-collapse on the multimodal examples, for which our neural JKO IC works.

---

> ### Comment · Reviewer_wcjw · 2024-11-26
>
> I would like to thank the author(s) for the detailed response, which answers most of my previous questions. After reading the revised manuscript, now I have one more question that puzzles me. In the theory, one needs the step size $\tau$ to be sufficiently small, but at the end of Section 4, the author(s) show that an exponentially increasing step size scheduling $\tau_{k+1}=4\tau_k$ is used. Is there any contradiction on this aspect?

---

> > ### Author Response · Authors · 2024-11-26
> >
> > Thank you for your evaluation of our revised paper.
> >
> > The main intuition behind the exponential step size rule is that as soon as our approximation $\mu^k$ is close to a minimum of the functional $\mathcal F$, it is sufficient that the functional $\mathcal G(\mu)=\frac1{2\tau}W_2^2(\mu,\mu^k)+\mathcal F(\mu^k)$ is convex locally around $\mu^k$ (see also (*) below). In this case, the distribution of our CNF will stay in this neighborhood, where $\mathcal G$ is convex.
> >
> > In practice, the convexity constant $\lambda$ from the functional $\mathcal F$ is unknown, such that the step size choide remains a hyper parameter to tune. Generally, the choice of the step size is a trade-off between stability and run time. Since we start with a very small step size, the global convexity condition for the steps will most likely be fulfilled in the beginning which prevents the mode collapsing behavior of CNFs. However, we can see that the (estimated) distance $W_2(\mu^{k+1},\mu^k)$ decreases very soon such that the step size can be increased.
> >
> > We did also experiment with constant step sizes (since it matches the theoretical results). While this generally works, this approach is very sensitive towards the choice of this step size and requires more steps to converge (such that training and evaluation are slower). In contrast, the exponential step size rule turned out to be more efficient. In addition, it is also comparable robust, since it fastly compensates for initial step sizes which are chosen smaller than necessary. Alternatively, adaptive step size rules could be considered (see, e.g., the paper of Xu et al., which is referenced at the end of Section 4).
> >
> > We extended Remark 18 to with some additional details.
> >
> > &nbsp;
> >
> > &nbsp;
> >
> > (*) In this context it might be worth to mention that the velocity fields in our CNFs are initialized with zero in the last layer. While this is a common trick for stabilizing the training of ResNets or NeuralODEs, it particularly implies that the transport map $\mathcal T$ of the normalizing flow is initialized as the identity such that $\mathcal T_\\#\mu^k=\mu^k$ in the initilization. We added this comment to the computational details of CNFs in Appendix F.2, i.e. the "Initialization"-paragraph.

---

> > > ### Comment · Reviewer_wcjw · 2024-11-26
> > >
> > > Thanks for the prompt response. Most of my questions have been addressed, and I would like to raise my score.

---

### Official Review · Reviewer_9UKK · 2024-11-03

**Soundness:** 3
**Presentation:** 4
**Contribution:** 2
**Rating:** 6
**Confidence:** 4

**Summary:**

This paper describes a sampling algorithm using proposals from a continuous normalizing flow with an additional reweighting / accept-reject step, which the authors claim enables the use of a data-free variational inference objective that reduces the reverse KL divergence. The methodology is analyzed and studied numerically on simple examples, where it outperforms a number of standard methods and more recent ML-based strategies like CRAFT and DDS.

**Strengths:**

- The paper is technically strong throughout and provides a clear assessment of the algorithm on a suite of relatively standard examples for the MCMC literature.
	- The clear connections to optimal transport provide a nice framework to enable connections to other algorithms.

**Weaknesses:**

- It is somewhat difficult to identify the contribution / novelty of this work. The basic idea of using reweighting and or metropolization has been used heavily with normalizing flows. E.g., https://arxiv.org/abs/2105.12603 or https://arxiv.org/abs/1812.01729. Much of the paper is standard background material.
- The connection between the objectives written in section 3.2 and the extremely widely used flow-matching / rectified flow objectives is not explored at all. While both of the aforementioned algorithms use (or can be directly formulated with) a Benamou-Brenier OT objective, the authors choose to instead use the more tedious and less efficient adjoint state framework for CNFs.
- The importance sampling algorithm is not very clearly explained. Because it seems to be the primary novel content in the paper, it would be beneficial to focus more heavily on this.
- Not at all clear from the paper how this scheme avoids mode collapse.

**Questions:**

- How does the neural JKO approach differ from the rectified flow / flow-matching objective?
- Is it possible to avoid the adjoint formulation, which is widely known to be inefficient?
- Some of the initial motivation focuses on mode collapse. Do any of your examples clearly indicate that this method avoids mode collapse where standard CNFs fail? A simple example in 2d with well-separated modes is typically enough to observe mode collapse when optimizing the reverse KL.

---

> ### Author Response · Authors · 2024-11-17
> **Response to Reviewer**
>
> Thank you very much for the review. We carefully revised the paper. Please find below our answers to your comments and questions.
>
> ## Contributions
>
> We want to highlight the following points:
>
> - To the best of our knowledge our importance-based rejection steps are the first resampling/rejection steps which allow it explicitly to access of the density throughout their application (see Proposition 8). This property is crucial for the iteration of several importance steps, in particular their concatination.
>
> - In contrast to classical MCMC steps (most Metropolis Hastings implementations, Langevin sampling, HMC, etc.), the importance-based rejection steps act non-local, i.e., the outcome of the rejection step does not depend on the input sample but on the whole input distribution. This is a very important property, which enables our algorithm to reweight the mass between disconnected modes, which is impossible for classical MCMC steps relying on local adjustments.
>
> - In addition to the rejection steps, we prove in Theorem 6 that the velocity fields of the JKO scheme converge to the velocity field of the Wasserstein gradient flow. In the context of the approximation with normalizing flows this is an important stability results, because the velocity fields are the object of interest that we aim to approximate.
>
> - Regarding the two papers, you have reference: The reweighting by Noe et al. considers a different setting, where both the (unnormalized) target density and some samples from the target distribution are given. In our setting (without samples from the target distribution), the reweighting is not applicable. In contrast to the paper of Gabrié et al., our neural JKO IC scheme returns a generative model, which can easily generate additional samples and evaluate the density of our approximation. Nevertheless, we agree that, from a high-level viewpoint, these papers consider the same problem as our paper with different tools. Therefore, we added them to the related work section.
>
> ## Rectified Flow / Flow Matching
>
> Rectified flows and flow matching consider the generative modeling setting. That is, they already assume given samples from the target distribution (which is not the case in our setting). Since the training is based on the construction of explicit couplings between training and latent samples, it is not applicable to sampling (or minimizing our Wasserstein-regularized KL functional). Therefore, we do not see a close relation to our loss function from Section 3.2.
>
> As a side note: There is a preprint (appeared less than a month before ICLR submission deadline), which extends flow matching to sampling by iteratively sampling from the current model and creating the couplings this approximation (https://www.arxiv.org/2408.16249). We have now added this preprint to the related works part. However, it is not clear if (and if yes how) this approach could be applied in our setting, i.e. for solving the steps of the neural JKO scheme with Wasserstein regularization.
>
> We added flow matching as an example of a generative model to the related work section and mention its sampling variant.
>
> ## Adjoint formulation of CNFs
>
> We now discuss some computational aspects of CNFs in Appendix F.2. In particular, we explain that the regularization of the velocity fields leads to straight paths such that the involved ODE can be solved by very few steps, see also the paper by Onken et al. 2021 for a detailed discussion.
>
> ## Presentation of Section 4
>
> We extended Section 4 with additional explanations. In particular, we added a summary of the importance corrected neural JKO scheme in Algorithm 1, point directly to the summary of the rejection step (Algorithm 5) and discuss more in detail how we build the final model.
>
> ## Avoiding Mode Collapse
>
> We added an related discussion in Appendix F.3, where we apply standard NFs (one coupling NFs and continuous NF) to our examples. In Figure 12 we plot the corresponding results and compare them with our neural JKO IC results. That standard (C)NFs mode collapse, while our neural JKO IC scheme covers all modes. Additionally, we added in Appendix F.1 a discussion on the relation between non-convex loss functions, local minima and mode collapse for multimodal target distributions. Before Section 3.1 we added a comment that the neural JKO scheme leads to a convex loss objective (as function on the space of probability measures), such that local minima cannot appear.

---

### Official Review · Reviewer_ff9a · 2024-11-03

**Soundness:** 3
**Presentation:** 3
**Contribution:** 3
**Rating:** 6
**Confidence:** 4

**Summary:**

This paper combines ideas from JKO sampling and annealed importance sampling to established an unbiased algorithm for implementing the Neural JKO protocol. They establish that using a learned velocity field to perform the transport monotonically decreases the KL divergence between the model density at time $t$ to the target density (as we should expect from a transport with an implied interpolating density). As the author understands it, the main contribution of this paper is the establishment of importance weights that allow to correct errors arising from the approximate velocity field in the neural JKO scheme.

**Strengths:**

- The authors exposit their method very nicely, and in addition make useful connections to various facets of the transport problem in the literature that are necessary for understanding the context of their method -- e.g. the motivation for the JKO scheme and its relation to regularized-OT solutions.

- The experiments are comprehensive as with those in the literature.

- Having a correctable JKO scheme is important. Given the reviewer's current knowledge, this was absent from previously proposed solutions, and for sampling algorithms, being able to show that your method is unbiased (or has controlled bias) is essential.

**Weaknesses:**

- As the reviewer understands it, the proposed method requires backpropagation through the solution of the ODE for training. Could the authors verify that this is true? This has been a perennial issue in using KL-like optimization for training samplers (it does not scale well with the dimensionality of the problem), and it's not clear to the reviewer that this primary obstacle is avoided in the OT regularized version.

- Moreover, this scales even worse when the ODE solution requires the computation of the trace of the divergence of a vector field. A measure of computational cost compared to the relevant literature / experiments seems important. In high dimensional sampling problems, I imagine the memory footprint of training probably has to be quite large.

- These issues are left out of the limitation discussion, which seems important, in addition to what they already remark on that the number of discretization steps (and therefore opportunity for rejection sampling) may be large if the flow is not learned well. Adding these reflections would be useful (and does not retract from the contribution of the authors!), and would allow the reviewer to bump their score a bit!

**Questions:**

To the reviewer, the rejection sampling step seems more like sequential monte-carlo than it does annealed importance sampling. These methods share a lot of the same conceptual reasoning, and the reviewer thinks in fact that the resampling procedure in SMC could be interpreted in this context too. Could the authors comment on that? Do they think it would be useful to make this connection?

---

> ### Author Response · Authors · 2024-11-17
> **Response to Reviewer**
>
> Thank you very much for the review.
>
> ## Answer to Questions: Relation to SMC
>
> Thank you very much for this question. There are some relations to SMC (alternate the use of importance weights with local adjustments), but there are also some major differnces. In particular we see the following advantages of our importance corrected JKO sampling over SMC:
>
> - After the first step, one cannot compute the density of the distribution of SMC any more. Instead SMC relies on replacing the density by a quotiont of a forward and backward Markov kernel. The quality of this approximation heavily relies on how exactly the backward kernel indeed approximates the inverse of the forward kernel. It is not directly clear how errors in the approximation propagate and add up. In contrast, we can compute the density of our approximation explicitly (up to the Radamacher trace estimation in the CNF).
>
> - Once, the SMC algorithm was running, generating additional samples is not possible in SMC. If additional samples are required, one has to redo the whole SMC procedure. In contrast, we obtain a generative model out of our algorithm, where we can easily generate additional samples after intermediate trainings steps of the intermediate measures.
>
> - Due to the SMC-resampling step the generated samples of SMC are not independent. In constrast, our samples are independent.
>
> We added some of these explanations to the section of related work.
>
>
> ## Discussion on mentioned Weaknesses: Computational cost of CNFs
>
> We added in Appendix F.2 a discussion of the computational limitations of CNFs which extends the limitations discussion in the conclusion section 6.
>
> Some synopsis:
>
> For solving the ODEs and differentiate its solutions, we use the package torchdiffeq (Ref.: Chen, 2018). In particular, this package does not backpropagate through the steps from the forward solver. Instead, it overwrites the backward pass of the ODE by sovlving an adjoint ODE. This avoids tracing within the forward pass and keeps the memory consumption low. The largest model (for the 1600 dimensional LGCP example) requires about 11 GB memory on a single GPU while still using a rather large batch size of 500. Hence, memory consumption is not the limiting factor in our model.
> We added related required GPU memory to table 5 in the implementation details section E.4 (note that we did not optimize our implementation wrt to memory, since this never was an issue throughout our experiments).
>
> In terms of computation time, the OT regularizations improves the ODE solutions significantly, since the minimum of the problems is a geodesic in the Wasserstein space. In particular, the sample trajectories through the ODE are straight paths such that adaptive ODE solvers only require very few steps to solve the ODE. These advantages of OT regularization are discussed in detail in the paper of Onken et al. 2021 ("OT-flow: Fast and accurate continuous normalizing flows via optimal transport"). We added this comment also to Section 2.2.
>
> On the other side, we agree that the evaluation of the divergence of the vector field remains a computational drawback, which is required since we have to evaluate the density of the model. However, this mainly effects the evaluation phase since during training a low-precision estimator (Hutchinson with one slice) is enough, as long as it is unbiased. Nevertheless, for the evaluation, we need a higher precision in order to be able to perform the importance-based rejection steps, which we realize by taking more slices (5 realizations of a Radamacher random vector) in the Hutchinson estimator. But we do not need to differentiate its computation at evaluation time, such that we can run it without gradient tracing.
>
> We added these discussions in Appendix F.2 and point to them in the limitations section.

---

> > ### Comment · Reviewer_ff9a · 2024-12-01
> >
> > Thanks to the authors for their thorough replies. I can perhaps follow up a little bit on my question about the relation to SMC. I did not mean explicitly the sequential Monte Carlo procedure as it is originally pitched, but rather the principle of performing resampling along the trajectory, which you could do at any point during your integration because you have importance weights.
> >
> > One important clarification is that for SMC, the stochastic update step does de-correlate the samples after they've been resampled, though of course this takes some time.
> >
> > Thanks for the information about the cost of the divergence! I think this is an open question in general about how to find a good network architecture that also has efficient divergence calculation, which would be useful to understand in this context.
> >
> > I'll keep my score as it is, given that it is already a weak accept. Thanks!

---

> ### Author Response · Authors · 2024-12-02
>
> Thank you for following up in the discussion. After reading your additional comments, we think that we misinterpreted your original comment regarding SMC. We agree that the motivation for our rejection steps is rather (sequential) importance sampling than annealed importance sampling. As already written before, we also agree that, from an abstract viewpoint, the importance-based rejection scheme shares some conceptional motivations with SMC.
>
> In order to address this comment in the manuscript, we rename the "Annealed Importance Sampling" paragraph in the beginning of Section 4 to "Importance Sampling" and reformulated it (see new version below). Moreover, we add a new remark (see below) in the  Neural JKO Sampling with Importance Correction paragraph of Section 4 to outline the relation to the resampling procedure in SMC.
>
> We would like to point out, that the resampling procedure from SMC cannot be applied in our setting, since it generates multiple samples at the same position. Since (despite the rejection steps) we solely use deterministig transformations, they would remain at the same position such that the effective number of samples would decrease over time.
>
> Regarding the de-correlation in SMC: Thanks for pointing this out. We wanted to express that in contrast to SMC our update step generates iid samples by construction. We reformulate the last sentence before the contributions paragraph in the introduction accordingly:
>
> > However, the corresponding importance sampling step generates non-iid samples, that de-correlate over time by construction.
>
> --------------------------------
>
> Reformulated version of the Importance Sampling paragraph:
>
> > **Importance Sampling**&nbsp;&nbsp;&nbsp; As a remedy, many sampling algorithms from the literature are based on importance weights, see, e.g., sequential Monte Carlo samplers (Del Moral et al., 2006) or annealed importance sampling (Neal, 2001). That is, we assign to each generated sample $x_i$ a weight $w_i=\frac{q(x_i)}{p(x_i)}$, where $p$ is some proposal density and $q$ is the density of the target distribution $\nu$. Then, for any $\nu$-integrable function $f\colon\mathbb{R}^d\to\mathbb{R}$ it holds that $\sum_{i=1}^N w_i f(x_i)$ is an unbiased estimator of $\int_{\mathbb{R}^d} f(x) d\nu(x)$.  Note that importance sampling is very sensitive with respect to the proposal $p$ which needs to be designed carefully and problem adapted.
>
> > **Rejection Steps**&nbsp;&nbsp;&nbsp; Inspired by importance sampling [...].
>
> --------------------------------
>
> New remark regarding relation to SMC:
>
> > **Remark** (Relation to SMC).&nbsp;&nbsp;&nbsp; While the steps of the importance-corrected neural JKO scheme are completely different to those in sequential Monte Carlo (SMC), SMC and neural JKO IC share the motivation of using importance-based resampling throughout the generation process. As a key difference to SMC, our neural JKO IC scheme successively builds a generative model, which allows to efficiently regenerate iid samples at any intermediate step and to evaluate the corresponding density explicitly.

---

### Official Review · Reviewer_wGJ8 · 2024-11-04

**Soundness:** 3
**Presentation:** 2
**Contribution:** 3
**Rating:** 6
**Confidence:** 4

**Summary:**

The paper considers the problem of sampling a probability distribution known up to a normalization constant, a classical question of interest in statistical mechanics and bayesian inference. It is proposed to do so by combining JKO steps using continuous normalizing flows (CNF) based on neural ODE with importance based rejection steps to correct for the bias introduced by errors in the CNF. More specifically, the JKO scheme is used to build a sequence of piecewise geodesic interpolations between densities performing steepest descent in Wasserstein-2 metric using as objective function the reversed KL divergence of the evolving measure from the target as energy. Since this KL divergence can be expressed as an expectation over the evolving measure of the unnormalized target distribution, it can be written explicitly. The optimization can thereby be performed in steps, which is practice can be done by propagating samples from one measure along the sequence to the next while solving some ODE along the way. A key aspect of the method is claimed to be the addition of an importance sampling step to accept or reject these samples along te way, in order to remove statistical errors that could accumulate as the walkers used in the sampling slowly drift from the sequence of measure generated.

**Strengths:**

The problem studied is an important one, and the algorithm proposed may be useful to address it.

**Weaknesses:**

The idea of building JKO schemes using is not new, see in particular the paper by Xu, Cheng, and Tie (cited here),. It would therefore be helpful to better emphasizes what is new here (which seems to be the importance sampling step, and demonstrate via example that it solves the slow convergence and instability issues that otherwise could arise. In this respect, the author could summarize (or relegate to the appendix) the background material on Wasserstein spaces, CNF and OT,  the JKO flow, and AIS as well as Metropolis-Hastings: while this material is explained well here, this exposition takes a lot of space and as a result the details on the actual scheme being proposed are rather sparse (in particular in the way the importance sampling bit of the scheme is implemented), and so are the numerical experiments reported.

**Questions:**

1. Please give details about the importance sampling scheme: the introduction in Sec. 4 could be shortened so that the **Neural JKO Sampling with Importance Correction** is expanded and the scene proposed clarified.

2. The paper blends CNF to perform steepest descent in continuous-time via the solution of ODE, with the JKO scheme which is a discrete-time proximal scheme. Doing so therefore requires discretizing the ODE, which of course is a step that always needs to be done, but the details of this discretization step should be discussed more carefully. In particular, how is the parameter $\tau$ in the JKO scheme chosen: is the same at every step?  Is the discretization step used to solve the CNF (and related ODE) taken to be $\tau$, or are multiple integration steps performed for every $\tau$? Can the time step by adapted along the way?

3. The method seems to require restarting the integration of the CNF from the beginning each time a rejection step is performed. Is this indeed the case? This seems quite costly in practice.

4. Several papers have recently appeared that propose similar schemes, with an importance sampling step also added, see in particular:

https://arxiv.org/abs/2410.02711

https://arxiv.org/abs/2102.07501

https://arxiv.org/abs/2111.15141

https://arxiv.org/abs/2307.01050

These papers should be discussed here, by giving a careful comparison between the methods and their numerical results.

5. More details about the numerics should be given.

---

> ### Author Response · Authors · 2024-11-17
> **Response to the Reviewer**
>
> We would like to thank the reviewer for the evaluation of our paper.
>
> ## Differences to prior work
>
> We agree with the reviewer that the derivation of our importance-based rejection steps is one key contributions of our paper. To the best of our knowledge our importance-based rejection steps are the first resampling/rejection steps which allow it to access the density of the arising model (see Prop 8). This is fundamental for applying them iteratively.
> Additionally, we want to highlight the following two points regarding Section 3:
>
> - We derive a convergence result of the velocity fields from the JKO scheme towards the velocity field of the Wasserstein gradient flows, which took quite some effort to prove. In the context of the approximation with normalizing flows this is an important stability results, because the velocity fields are the object which we aim to approximate.
>
> - The paper of Xu et al. 2024 consider a different setting than our paper, even though they derive a similar simulation of the JKO scheme via CNFs. Since they are given samples from the target distribution (as opposed to our setting given its unnormalized density function), their target functional is given by $\mathcal F(\mu)=KL(\mu,\nu)$, where $\nu$ is a standard normal distribution. Due to the strong log-concavity of $\nu$, this is a strongly convex functional in the Wasserstein space simplifying the solution of these subproblems significantly. In particular, Xu et al. 2024 do not have to deal with common problems from the sampling problem like slow convergence of the underlying flow due to multimodal or narrow target distributions leading to poor constants in related Poincaré or log-Sobolev inequalities.
>
> We have rewritten the related work part where we now clearly indicate which models consider the sampling which consider generative modeling. For the exposition provided, we feel that the backgrounds on the Wasserstein distance and its dynamical formulation and time rescaling are required for presenting Thm 6 properly. Therefore, we decided not to relegate more backgrounds to the appendix.
>
> ## Answer to Questions:
>
> 1. We would like to thank the reviewer for this input. We extended the paragraph about Neural JKO Sampling with Importance Correction and added a summary of the proposed scheme in Alg 1. Additional, information about a suggested realization of layer choices is given in the final part of section 4. An algorithm for the application of a rejection step was already included (Alg 5).
> However, we would like to point out that the design of the rejection steps and their analysis (Prop 8) belong to the main contributions of our paper and cannot be found in the literature (see Rem 7 for the relation to classical rejection sampling).
> For the introductionary paragraph on annealed importance sampling, we followed your suggestion to shorten it.
>
>
> 2. Please note that there are two different kinds of discretization. Regarding the discretization of the ODEs from the CNFs, we added Appendix F.2 with more details. Regarding the discretization (the $\tau_k$) from the JKO scheme:  While, it is very important to choose the $\tau$ small enough in the beginning, the Wasserstein gradient flow slows down fastly, which suggest to increase $\tau$ over time. One possibility for choosing $\tau$ could be to choose $\tau$ adaptively based on the Wasserstein distance between $\mu_k$ and $\mu_{k+1}$ (Xu et al. 2024 have done something like that). However, in our numerics we found that the simpler scheme of choosing $\tau_{k+1}=4\tau_k$ worked fine, such we sticked to it. While the chosen $\tau_k$ were already stated in the implementation details (Appendix E.4), we have now added a comment about that at the end of Sec 4 (and mention the adaptive schedule by Xu et al. 2024).
>
> 3. In each rejection step, we have to evaluate (requires no training and no backpropagation!) the whole model again for only a subset of the samples (not all of them!). However, since we do not need to retrain the models and do not require any gradient computations in these resampling steps, the training and evaluation times remain moderate (they are already included in Tab 5). In addition, Rem 11 and the discussion on limitations comment on the runtime effects of the rejection steps.
>
> 4. We included the additional references in the related work section of the paper. However, we would like to emphasize that the first one appeared after the submission deadline of ICLR. The second one is already included in the related work and we compared our numerical results with its follow up paper (CRAFT). The last two are conceptually similar to DDS (included in our numerics) and only use importance sampling at final time (not as intermediate steps). We added a comment in the related work section.
>
> 5. The detailed numerical setup is specified in Appendix E. If there is some specific point missing, we are happy to add it.

---

> > ### Comment · Reviewer_wGJ8 · 2024-12-01
> >
> > Thank you for this detailed answer. Some of my concern have been partially addressed, and as a result I have raised my score accordingly. However, my overall feeling remains that the paper would benefit from a thorough revision, and as a result may be better suited for a resubmission.

---

### Author Response · Authors · 2024-11-17
**Response to the Reviewers**

We would like to thank all reviewers for the thorough evaluation of our paper. We carefully revised our paper. Changes and additions are indicated in blue. We respond to each reviewer seperately with detailed answers how we addressed the raised comments and questions.

---

### Meta-Review · Area_Chair_NSWp · 2024-12-19

**Metareview:**

This paper proposes a novel method for sampling from unnormalized distributions by combining continuous normalizing flows (CNFs) with rejection sampling based on importance weights to correct biases. The CNF framework enables fast sampling and density evaluation using neural networks, while the biases introduced by CNFs are appropriately corrected through rejection sampling.

All reviewers acknowledged the significant contribution of the JKO scheme in this study. However, concerns were raised regarding the lack of discussion on importance sampling, computational costs, and parameter tuning. As the paper is borderline, I conducted a detailed review and reached a similar conclusion. While the mathematical contributions are substantial, as an algorithmic proposal, the paper requires major revisions, as also highlighted by Reviewer wGJ8.

Specifically, although the authors added a pseudo code (Algorithm 1) in response to reviewer comments, the algorithm calls several additional sub-algorithms (Algorithms 2–4 in the Appendix), which significantly impacts readability. Moreover, the critical aspect of how the JKO scheme is combined with importance sampling is not explicitly detailed in the pseudo code. For instance, the description, "In practice, we first utilize CNF layers only, then use several blocks consisting of a single CNF layer composed with 3 rejection steps", remains vague. While Appendix E.2 provides details about the combination, this should be clearly presented in the main text as an explicit algorithm.

Additionally, there is no discussion on the stability of the proposed method with respect to how the JKO scheme and importance sampling are combined. The pseudo code also lacks details about key hyperparameters, such as step sizes, which are critical for the algorithm’s implementation. Multiple reviewers raised concerns about hyperparameters, and while these were partially addressed during the discussion phase, such discussions should also be included in the main text due to their importance.

Another significant aspect of the study is the ability to calculate densities, which is an important advantage of the proposed method. However, the specific method for this calculation is only described in the pseudo code in the Appendix. This too should be presented in the main text for clarity and completeness.

Addressing these issues requires substantial revisions to ensure the algorithmic aspects are adequately described. While the mathematical contributions of this study are significant, there is a notable discrepancy in quality between the theoretical and algorithmic components. Given the inadequacies in the latter, I recommend rejection at this time, with the hope that the paper will be resubmitted after major revisions.

**Additional Comments On Reviewer Discussion:**

Reviewer wGJ8 and Reviewer ff9a raised concerns regarding the computational cost of the proposed method. These concerns were addressed through supplementary materials provided by the authors, resolving the issue. Reviewer wGJ8 and Reviewer 9UKK highlighted a lack of discussion regarding comparisons with existing studies. This was resolved by incorporating additional discussions into the main text.

Reviewer 9UKK and Reviewer wcjw raised concerns about the motivation for using CNFs and the weak connection between CNFs and the objective function. These concerns were addressed with further discussions provided by the authors. Similarly, Reviewer wGJ8 and Reviewer 9UKK pointed out the lack of detail regarding the importance sampling component of the algorithm, while Reviewer wGJ8 and Reviewer wcjw noted the insufficient discussion regarding discretization. The authors attempted to address these issues by adding pseudo code and supplementary discussions.

However, as highlighted in the above meta-review, the algorithmic component of the paper still suffers from a lack of readability and sufficient discussion. Reviewer wGJ8 also acknowledged that the authors’ responses were inadequate in addressing these concerns. As the Area Chair (AC), I concur with this assessment and recommend rejection at this time.

---

### Decision · Program_Chairs · 2025-01-22

Reject